# CTCF mutation at R567 causes developmental disorders via 3D genome rearrangement and abnormal neurodevelopment

Jie Zhang [1,2,9], Gongcheng Hu [3,9], Yuli Lu [1,4,9], Huawei Ren [5,9], Yin Huang [3], Yulin Wen [1,4], Binrui Ji [1,6], Diyang Wang [7], Haidong Wang [5], Huisheng Liu [3], Ning Ma [3], Lingling Zhang [8], Guangjin Pan [1,4], Yibo Qu [7], Hua Wang [8], Wei Zhang [3], Zhichao Miao [3] & Hongjie Yao [1,2,3,4] ✉

The three-dimensional genome structure organized by CTCF is required for development. Clinically identified mutations in *CTCF* have been linked to adverse developmental outcomes. Nevertheless, the underlying mechanism remains elusive. In this investigation, we explore the regulatory roles of a clinically relevant R567W point mutation, located within the 11th zinc finger of CTCF, by introducing this mutation into both murine models and human embryonic stem cell-derived cortical organoid models. Mice with homozygous CTCF^R567W mutation exhibit growth impediments, resulting in postnatal mortality, and deviations in brain, heart, and lung development at the pathological and single-cell transcriptome levels. This mutation induces premature stem-like cell exhaustion, accelerates the maturation of GABAergic neurons, and disrupts neurodevelopmental and synaptic pathways. Additionally, it specifically hinders CTCF binding to peripheral motifs upstream to the core consensus site, causing alterations in local chromatin structure and gene expression, particularly at the clustered protocadherin locus. Comparative analysis using human cortical organoids mirrors the consequences induced by this mutation. In summary, this study elucidates the influence of the CTCF^R567W mutation on human neurodevelopmental disorders, paving the way for potential therapeutic interventions.

There is a growing body of evidence indicating a close association between the disruption of chromatin's three-dimensional structure and the onset and progression of various diseases, including developmental disorders and cancers[1–5]. Playing a crucial role as the master weaver of the genome[6], CCCTC-binding factor (CTCF) actively participates in the regulation of Topologically Associating Domains (TADs) and chromatin loops[7–9]. Alterations in the expression level or insufficient dosage of CTCF at the cellular level can result in shifts in cell fate,

potentially leading to cellular demise[7,10–12]. Notably, maternal depletion of *Ctcf* halts early embryonic development[13], elevating zygotic lethality[14]. In excitatory neurons, a conditional knockout of *Ctcf* induces neuronal cell death and gliosis[15].

Moreover, *Ctcf* proves indispensable for early forebrain development, contributing to a delicate balance between neural progenitor cell proliferation and differentiation[16]. Recent discoveries in clinical and targeted sequencing studies have unveiled cases of heterozygous

*CTCF* mutations, encompassing deletions, frameshift, and missense mutations. These variations are highly correlated with potential symptoms subsumed under the classification of intellectual developmental disorder, autosomal dominant 21 (OMIM 615502), which can range from global developmental delay, intellectual disability, and short stature to autistic behaviors and symptoms resembling congenital heart disease[17–21]. While mouse models utilizing tissue-specific *Ctcf* conditional knockout have been explored to mirror the pertinent phenotypic traits[16,22–25], the impact of germline-origin *Ctcf* point mutations on individual development and the underlying mechanisms in human diseases remain unclear.

Building on the observation that a clinically heterozygous *CTCF* mutation at c.1699 C > T (p.Arg567 > Trp) induces severe phenotypes, including intellectual disability, microcephaly, hypotonia, growth deficiency, delayed development, short stature, delayed bone age, and feeding difficulties[17,19,20,26,27], we establish a *Ctcf* mutant mouse model and a human embryonic stem cell (hESC)-derived cortical organoid model harboring the CTCF[R567W] mutation to explore its mechanistic involvement. In the mouse model, homozygous CTCF[R567W]-mutant offspring experience early postnatal mortality, while their heterozygous counterparts exhibit a survival rate comparable to wild-type mice, displaying only a lean phenotype during the juvenile stage. Additionally, at embryonic-day-18.5 (E18.5), mice with homozygous CTCF[R567W] mutation manifest abnormal neurological and cardiopulmonary development.

Specifically, within the mouse cortex, the CTCF[R567W] mutation accelerates the depletion of stem-like cells, hastens the maturation of GABAergic neurons, and induces irregularities in both neurodevelopmental and synaptic pathways. The CTCF[R567W] mutation leads to a reduction in CTCF binding to its upstream motifs, causing a partial reorganization of chromatin interactions, and subsequently influencing gene expression, particularly within the clustered protocadherin (*cPcdh*) locus. Consistent with the mouse model, the hESC-derived cortical organoid model reveals that the homozygous CTCF[R567W] mutation hinders self-organization during differentiation, while the heterozygous mutation causes an imbalance in the differentiation of stem-like cells and the maturation of neurons. This imbalance, induced by the CTCF[R567W] mutation, impacts pathways crucial for neural development and affects genes implicated in neurodevelopmental disorders.

## Results
### Construction and phenotypic characterization of CTCF[R567W]-mutant mice
The R567 residue of CTCF, a conserved residue across most known orthologues (Fig. 1a), is located proximal to the splice donor site in exon 9. Notably, this residue, situated within the 11[th] zinc finger (ZF) domain of the CTCF protein (Fig. 1b), was predicted to specifically contact DNA[17,28]. To investigate the impact of the CTCF[R567W] mutation on developmental phenotypes, we generated a mouse model harboring the CTCF[R567W] mutation using CRISPR/Cas9-mediated genome editing (Fig. 1b). Through multiple rounds of breeding and genotyping, we observed that heterozygous mutant mice (*Ctcf*[+/R567W]) exhibited a birth state similar to that of wild-type mice (*Ctcf*[+/+]), while homozygous mutant mice (*Ctcf*[R567W/R567W]) experienced respiratory distress and succumbed within 30 minutes (min) after birth (Fig. 1c and Supplementary Fig. 1a).

Upon analyzing the genotyping results of numerous embryos at E18.5, we noted that most *Ctcf*[R567W/R567W] mice could developed into intact individuals, albeit considerably smaller than their wild-type counterparts (0.815 ± 0.107 g vs 1.157 ± 0.095 g, $p = 1 \times 10^{-15}$; Fig. 1d). Additionally, based on a 1-month recording, *Ctcf*[+/R567W] mice demonstrated a relatively thin physique with no discernible difference in body weight by sex (Fig. 1e–g), consistent with the lean phenotype reported in previous studies on *Ctcf* hemizygous mice[9] and clinical cases of *CTCF*

mutation[11,17,19,29]. The lean phenotype in *Ctcf*[+/R567W] mice might be attributed to reduced levels of growth hormone (GH), as indicated by enzyme-linked immunosorbent assay (ELISA) (16.14 ± 3.78 mg/mL vs 13.1 ± 2.38 mg/mL, $p = 0.043$) and quantitative RT-PCR (RT-qPCR) (Supplementary Fig. 1b, c). Despite this, the lifespan of *Ctcf*[+/R567W] mice was comparable to that of *Ctcf*[+/+] mice (Supplementary Fig. 1d). Furthermore, we noted a statistically significant increase in *Ctcf* expression in neuron and lung tissues, as revealed by RT-qPCR, due to the CTCF[R567W] mutation (Supplementary Fig. 1e). However, this mutation did not significantly alter CTCF splicing or distribution across tissues, as assessed by Western blot and immunohistochemistry (Supplementary Fig. 1f, g).

In investigating pathological phenotypes following *Ctcf* mutation, we focused on the brain, heart, and lung tissues due to the birth lethal phenotype observed in *Ctcf*[R567W/R567W] mice and the prevalence of neurodevelopmental disorders in clinical populations. Our observations revealed normalcy in all three tissues of both *Ctcf*[+/+] and *Ctcf*[+/R567W] mice, with no apparent pathological changes (Fig. 1h, i and Supplementary Fig. 1h and 2a–c). However, *Ctcf*[R567W/R567W] mice exhibited abnormal histopathology in their hearts, characterized by a decreased proportion of ventricles, resembling the noncompaction cardiomyopathy phenotype, including a thin ventricular wall, noncompaction of the ventricular myocardium, and trabecular hyperplasia (Fig. 1i). Additionally, we observed a thickened alveolar interstitium and dense, noninflated alveoli in the lungs of E18.5 and postnatal day 0 (P0) *Ctcf*[R567W/R567W] mice (Supplementary Fig. 1h). While these pathological changes could potentially contribute to respiratory distress, they might also represent secondary effects of the animals' demise rather than direct consequences of *Ctcf* homozygous mutation.

To investigate whether the *Ctcf*[+/R567W]-induced phenotype in mice could replicate clinical neurodevelopmental disorders and autism-like traits, we conducted a series of behavioral experiments, including the open-field test (OFT), novel object recognition (NOR) test, three-chamber test, elevated plus maze (EPM) test, morris water maze test, and rotarod test. No significant abnormalities were observed in *Ctcf*[+/R567W] adult mice compared with wild-type mice (Supplementary Fig. 2d–q), except for a lower resistance to fatigue in the rotarod test (Supplementary Fig. 2r). These findings suggest that, unlike *Ctcf* hemizygous mice which exhibit loss of *Ctcf* expression in the brain[30], *Ctcf*[+/R567W] mice maintain mostly normal brain function, given that *Ctcf* expression is preserved. Discrepancies between mice and humans may be attributed to the complexities of the human nervous system. In summary, our results indicate that the homozygous CTCF[R567W] mutation leads to birth lethality and developmental delay in mice, somewhat mirroring known clinical phenotypes manifested as overall developmental disorders.

### CTCF[R567W] homozygous mutation hampers neural development and electrical activity
To explore the potential impact of the CTCF[R567W] mutation on neurodevelopment, we conducted follow-up experiments using E18.5 fetal mouse brain tissues and primary neurons. As the phenotypes of *Ctcf*[+/R567W] mice were generally consistent with those of *Ctcf*[+/+] mice according to preliminary observations and tests, therefore, we focused on examining the effects of the homozygous mutation on neural development and activity in subsequent studies.

We initially labeled cultured primary neurons with an anti-TUJ1 antibody to evaluate neurite complexity. Both primary neurons and stem cell-derived neural clusters from *Ctcf*[R567W/R567W] mice exhibited significantly less complex neuronal protrusions compared to cells from *Ctcf*[+/+] mice (Fig. 2a and Supplementary Fig. 3a). Despite no differences in soma size across genotypes, morphological analysis revealed fewer primary neurites in neurons from *Ctcf*[R567W/R567W] mice (4.524 ± 1.504 vs 5.864 ± 1.552, $p = 0.0127$; Fig. 2b). Total neurite length was also significantly decreased in *Ctcf*[R567W/R567W] mice (171.2 ± 59.38 vs

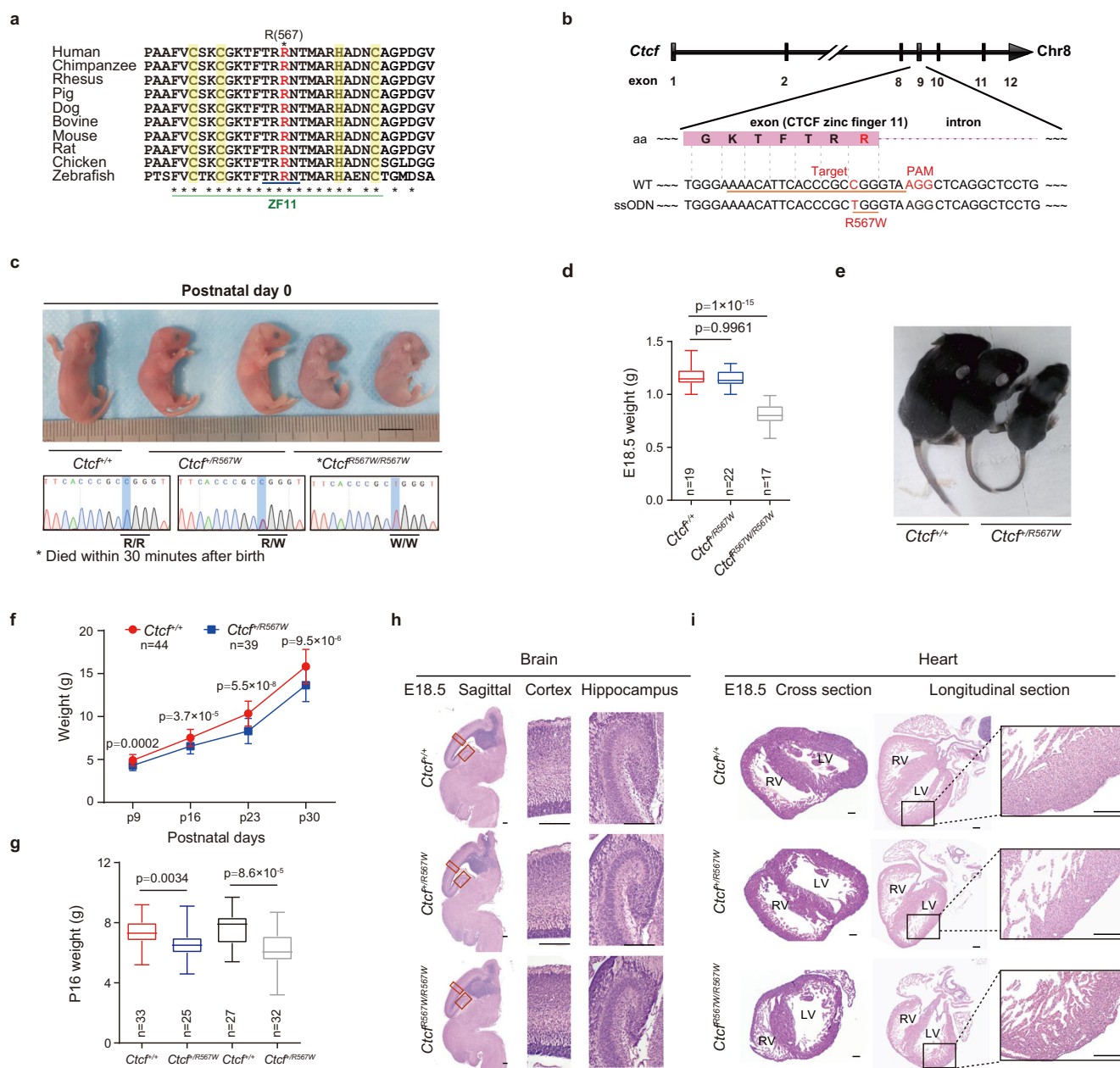

**Fig. 1 | Construction of CTCF$^{R567W}$-mutant mice and phenotypic analysis. a** The amino acid (aa) conservation of CTCF around R567 across species. CTCF ZF11 residues annotated in UniProt are marked with a green line, DNA-binding residues (residues 565–568) with a blue line, and $Zn^{2+}$ coordinating residues with a yellow band. R567 is highlighted in red. **b** CRISPR/Cas9 knock-in of the CTCF$^{R567W}$ mutation diagram. The sgRNA targeting sequence (*Ctcf* exon 9) and the mutation site R567W (R, Arg, CGG; W, Trp, TGG) are underlined. The mutated base and the PAM sequence NGG are in red. The aa sequence around the mutation site is highlighted with a pink box. **c** Representative photograph and Sanger sequencing results of neonatal mice. Scale bar, 10 mm. The mutation base C > T is highlighted in blue. **d** Box and whisker plots showing the body weight (95% confidence interval) of E18.5 mice (*Ctcf$^{+/+}$*: *n* = 19, *Ctcf$^{+/R567W}$*: *n* = 22, *Ctcf$^{R567W/R567W}$*: *n* = 17). **e** Two-week-old *Ctcf$^{+/+}$* and *Ctcf$^{+/R567W}$* mice images. **f** Weight curves of postnatal day 9 (P9), P16, P23, and P30 *Ctcf$^{+/+}$* and *Ctcf$^{+/R567W}$* mice (*Ctcf$^{+/+}$*: *n* = 44, *Ctcf$^{+/R567W}$*: *n* = 39). **g** Box and whisker plots showing the body weight (95% confidence interval) of P16 mice for different sexes of *Ctcf$^{+/+}$* and *Ctcf$^{+/R567W}$* mice (*Ctcf$^{+/+}$*: M, *n* = 33, F, *n* = 25, *Ctcf$^{+/R567W}$*: M, *n* = 27, F, *n* = 32; M: male, F: female). **h, i** Sagittal brain and heart paraffin H&E staining images of *Ctcf$^{+/+}$*, *Ctcf$^{+/R567W}$*, and *Ctcf$^{R567W/R567W}$* mice at E18.5. Enlarged views of the cortex, hippocampus, and left ventricular wall are shown on the right. Scale bars, 200 µm. Quantitative data are presented as mean ± SD. *p*-values were obtained by one-way ANOVA with Dunnett's multiple comparisons test adjustment (**d**), two-way ANOVA with Šídák's multiple comparisons test adjustment (**f**), and two-tailed unpaired *t*-test (**g**) are indicated in the graphs. Experiments in (**h**, **i**) were repeated independently three times with similar results. All Box and whisker plots show lower and upper quartiles (box limits), median (center line) and minimum to maximum values (whiskers). Source data are provided as a Source Data file.

339.3 ± 109.9 µm, $p = 8.4 \times 10^{-7}$; Fig. 2b), indicating that the *Ctcf* homozygous mutation results in neurite abnormalities.

We employed microelectrode array (MEA) to assess E18.5 primary neurons and explore whether the CTCF$^{R567W}$ mutation affects neuro-electric activity and neural network formation. While all three genotypes exhibited spontaneous firing activity, *Ctcf$^{R567W/R567W}$* neurons displayed significantly lower neural network activity, evident through reduced numbers of spikes, bursts and network bursts, along with decreases in mean firing rate and synchronization index (Fig. 2c–h and Supplementary Fig. 3b–e). No significant variation in these metrics was observed between *Ctcf$^{+/+}$* and *Ctcf$^{+/R567W}$* neurons (Fig. 2c and Supplementary Fig. 3f–i).

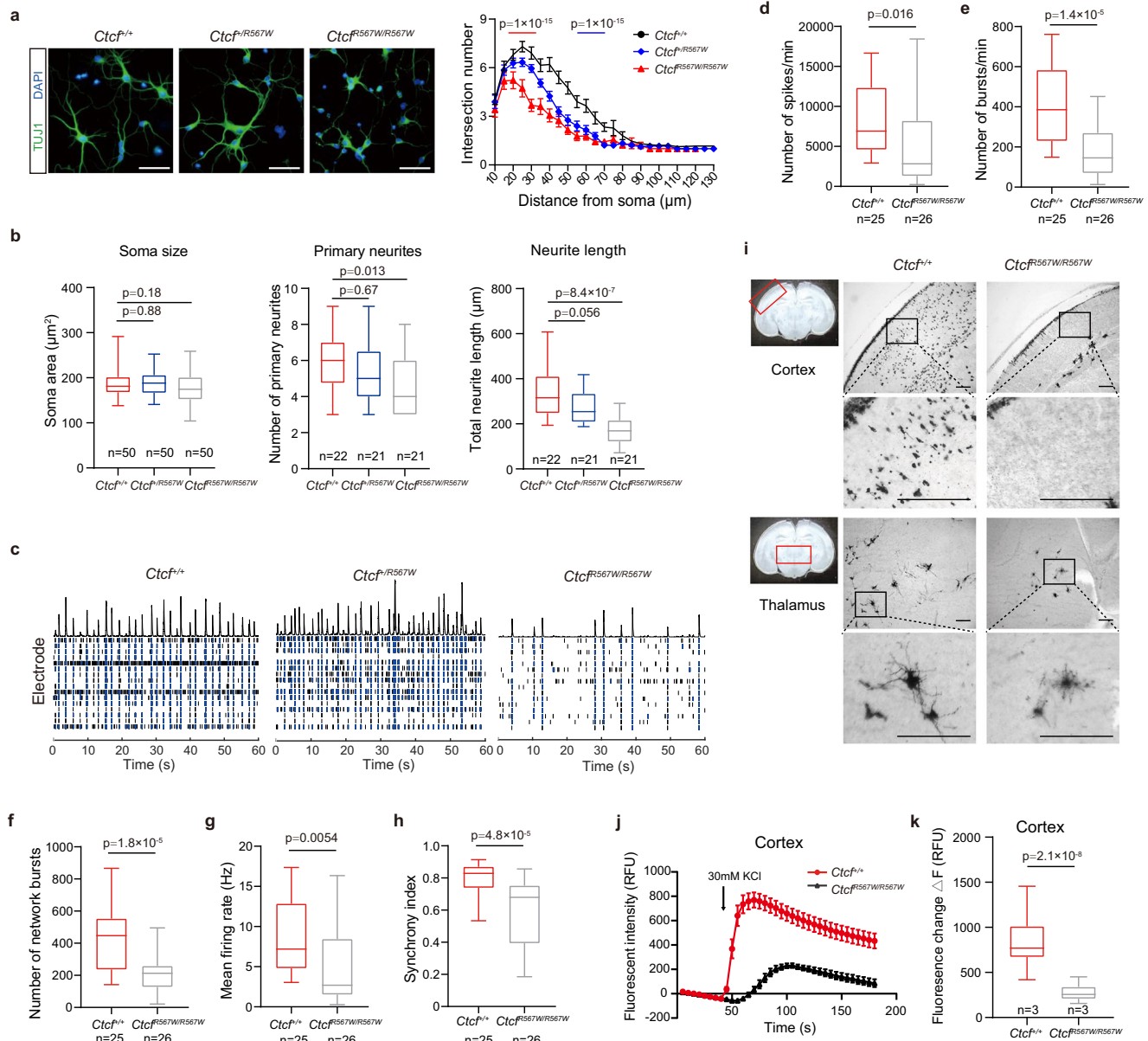

**Fig. 2 | CTCF$^{R567W}$ homozygous mutation impairs neural development and electrical activity. a** TUJ1 (green) immunostaining and Sholl analysis of primary neurons cultured for 5 days from E18.5 mouse forebrains (*n* = 3 for each genotype). Nuclei were stained with DAPI (blue). Scale bars, 50 μm. **b** Quantification of soma sizes (left), the number of primary neurites (middle) and total neurite length (right) of primary neurons from (**a**) (*n* = 3 for each genotype, the n of neurons counted are indicated in the graph). **c** The raster image of MEA experiments showing electrical activity of cultured neurons from three genotypes over a time period (60 s) on day 14. Black and blue bars represent spikes and bursts, respectively. The top of the raster plot shows the spike histogram. **d**–**h** Quantification of MEA metrics of *Ctcf$^{+/+}$* and *Ctcf$^{R567W/R567W}$* cultured neurons. Statistics of electrical activity for 15 min, including the number of spikes per min (**d**) the number of bursts per min (**e**) the number of network bursts (**f**) the mean firing rate (**g**) and the synchrony index (**h**) are provided (*Ctcf$^{+/+}$*: *n* = 4, *Ctcf$^{R567W/R567W}$*: *n* = 2; the n of cultured wells recorded are indicated in the graph). **i** Golgi staining of brain slices at the cortex (top) or thalamus (bottom) from *Ctcf$^{+/+}$* and *Ctcf$^{R567W/R567W}$* mice at E18.5. Scale bars, 100 μm. Areas of interest are indicated by red rectangles. **j**, **k** Calcium influx after depolarization from in vitro-cultured cortical neurons on day 10 for *Ctcf$^{+/+}$* and *Ctcf$^{R567W/R567W}$* mice (*n* = 3 for each genotype) (RFU: relative fluorescence units). Data from the Sholl analysis are presented as mean ± SEM, and other quantitative data are presented as mean ± SD. *p*-values by two-way ANOVA with repeated measures (**a**), one-way ANOVA with Dunnett's multiple comparisons test adjustment (**b**), and two-tailed unpaired *t*-test (**d**–**h** and **k**) are indicated in the graphs. Experiments were repeated three times independently with similar results. All Box and whisker plots show lower and upper quartiles (box limits), median (center line) and minimum to maximum values (whiskers), representing 95% confidence intervals. Source data are provided as a Source Data file.

Golgi staining further revealed fewer stained neurons and neurites in the thalamus and cortex of *Ctcf$^{R567W/R567W}$* mice compared to *Ctcf$^{+/+}$* mice (Fig. 2i). Calcium (Ca$^{2+}$) imaging experiments also demonstrated significantly attenuated Ca$^{2+}$ signals in cultured cortical and hippocampal neurons of the *Ctcf$^{R567W/R567W}$* group (Fig. 2j, k and Supplementary Fig. 3j, k). In conclusion, our results suggest that the *Ctcf* homozygous mutation at R567 impairs neural activity and network function.

## CTCF$^{R567W}$ mutation induces premature depletion of progenitor cells and accelerates the maturation of GABAergic neurons

CTCF plays a crucial role in transcriptional regulation[31–33]. To assess the impact of the homozygous CTCF$^{R567W}$ mutation on gene expression in brain tissue, we conducted RNA sequencing (RNA-seq) on whole brains from E18.5 fetal mice. Principal component analysis (PCA) revealed a subtle deviation in *Ctcf$^{R567W/R567W}$* samples from *Ctcf$^{+/+}$* and *Ctcf$^{+/R567W}$*

samples, suggesting a certain degree of heterogeneity among the samples (Supplementary Fig. 4a). Differentially expressed gene (DEG) analysis showed minimal differences between *Ctcf*[+/+] and *Ctcf*[+/R567W] mice, while a limited number of DEGs were identified in *Ctcf*[R567W/R567W] mice (Supplementary Fig. 4b). Considering the diverse cell types in the brain, we isolated neurons from E18.5 fetal brain cortices, cultured them in vitro, and conducted RNA-seq experiments. The results indicated a downregulation of more DEGs in neurons, most of which were consistent with the RNA-seq data from the whole brain (Fig. 3a, b). Notably, these co-downregulated genes were enriched in interferon pathways and *cPcdh*-mediated cell adhesion molecules (Fig. 3c). The *cPcdh* gene clusters, consisting of three tandem gene clusters (*Pcdhα*, *Pcdhβ*, and *Pcdhγ*), are associated with neuronal identity, synaptic organization, axonal guidance, and neural circuit assembly[34–36].

To assess the impact of the CTCF[R567W] mutation on neurodevelopment at the single-cell level, we performed single-nucleus RNA seq (snRNA-seq) on brain cortices from E18.5 *Ctcf*[+/+] and *Ctcf*[R567W/R567W] mice. Following quality control and data preprocessing, we obtained a total of 13,137 and 15,661 cells from wild-type and mutant samples, respectively. Utilizing the ScType platform for annotation[37], cluster analysis identified nine cell populations, including radial glial cells, neural progenitor cells, neuroblasts, immature neurons, glutamatergic neurons, GABAergic neurons, oligodendrocyte precursor cells, microglia, and endothelial cells (Fig. 3d and Supplementary Fig. 4c). Our data revealed that the CTCF[R567W] mutation had no significant effect on the cellular phenotype or landscape of the mouse cortex but led to noticeable shifts in specific cohorts. Specifically, the mutant mice exhibited a reduction in the proportion of stem-like neural progenitor cells and radial glial cells, as well as immature neurons, coupled with an increase in postmitotic neuroblasts and inhibitory GABAergic neurons (Fig. 3e). Additionally, a marginal decrease in excitatory glutamatergic neurons was observed (Fig. 3e).

The DEGs identified through functional annotation of main stem/progenitor cells and neuronal clusters revealed that the CTCF[R567W] mutation impacts crucial pathways guiding neural development, synaptic structuring, and cell-cell adhesion. However, the major pathways affected were not consistently observed across different cell subgroups (Fig. 3f, g). Notably, GABAergic neurons exhibited enriched differential pathways associated with locomotory behavior, cognition, and synaptic transmission. In contrast, glutamatergic neurons displayed enrichment in pathways related to the regulation of membrane potential, neuron migration, and learning processes (Fig. 3g). Furthermore, an overlap was observed between DEGs and risk genes identified in genome-wide association studies (GWAS) of autism spectrum disorder (ASD) (Supplementary Fig. 4d). Additionally, there was a downregulation of *Pcdhα* and *Pcdhβ* genes and an upward trend in the expression of *Pcdhγ* genes, demonstrating consistent expression changes across all subgroups (Supplementary Fig. 4e). Collectively, the CTCF[R567W] mutation disrupts mouse cortical neurodevelopment, leading to premature depletion of the progenitor pool and accelerated maturation of GABAergic neurons. These findings provide partial elucidation of the phenotypic manifestations observed in previous neuronal experiments (Fig. 2), establishing a tangible link to intellectual disabilities and autism-like phenotypes in clinical developmental disorders associated with *CTCF* mutation.

### CTCF[R567W] mutation has little effect on cell type proportions but leads to noticeable DEGs in E18.5 heart and lung tissues

Further investigation into the effect of the CTCF[R567W] mutation on gene expression in E18.5 heart and lung tissues, guided by snRNA-seq and focusing on cardiopulmonary pathological phenotypes in *Ctcf*[R567W/R567W] mice (Fig. 1i and Supplementary Fig. 1h), revealed distinct patterns. We classified 11 clusters, with 7 associated with cardiomyocytes (CMs), for cardiac tissue (Fig. 3h and Supplementary Fig. 5a), and 14 clusters for lung tissue cell types (Supplementary Fig. 5b, c). The mutation did not

significantly alter the proportions of most cell types (Supplementary Fig. 5d, e). DEG analysis showed that CTCF[R567W] resulted in a limited number of DEGs per cell type (Supplementary Fig. 5f, g). Notably, in cardiac cells, genes linked to energy metabolism and myocardial contraction were significantly downregulated in *Ctcf*[R567W/R567W] mice, while genes associated with muscle tissue development were upregulated (Fig. 3i, j). In the lungs, irregular expression of genes related to vascular structures and cell morphogenesis was observed, with DEGs varying across all clusters (Supplementary Fig. 5h, i). These gene pathways might correlate with the previously detected abnormal heart and lung morphology phenotypes and could contribute to potential respiratory failure at birth in these mutant mice (Fig. 1i and Supplementary Fig. 1h). Joint analysis of DEGs across heart, brain, and lung tissues indicated that the DEGs induced by CTCF[R567W] were specific to both tissues and cell types (Fig. 3k).

### CTCF[R567W] weakens CTCF binding to sites with upstream motifs

To explore the impact of the CTCF[R567W] mutation on the manifestation of the associated disease phenotype through alterations in specific CTCF binding, we scrutinized the chromatin binding characteristics of CTCF[R567W]. This investigation is crucial, considering that CTCF exerts its biological functions by binding to specific DNA motifs via its 11 ZFs. Specifically, ZFs 4–7 engage the M1/core (C) motif, which represents the core consensus, and ZFs 9–11 engage the M2/upstream (U) motif, corresponding to peripheral motifs upstream to the core consensus site[28,38–40]. The crystal structure of CTCF ZFs (6–11) in complex with its DNA motif (PDB: 5YEL) has been resolved, and it is established that amino acids 565–568 within ZF11 of CTCF are suggested to form hydrogen bonds with DNA, corresponding to the CTCF U motif (Fig. 4a, b)[28]. Structural modeling indicates that the R567W mutation causes a shift of the ZF away from the DNA's phosphate backbone, thereby disrupting the critical hydrogen bond between R567 and the DNA backbone. Furthermore, the W567 residue may impede the formation of the essential hydrogen bond between the neighboring R566 residue and DNA bases, further compromising ZF11 binding to the U motif (Fig. 4b, c).

To validate the hypothesis, we synthesized oligonucleotide probes containing the CTCF C motif (M1) and U motif (M2), respectively, based on the previous studies[40]. We then purified recombinant fusion proteins for Glutathione S-Transferase (GST)-tagged wild-type (wt) CTCF ZF domain (ZF1–11), ZF1–11 with R567W mutation (mut), and ZF1–11 with R567W/R566C double mutation (dmut) (Supplementary Fig. 6a). In vitro electrophoretic mobility shift assay (EMSA) results indicated that CTCF mutation diminished migration of both U and C motif probes, as quantified by the bound/free ratios (Supplementary Fig. 6b, c). Additionally, GST pull-down assays followed by dot blot analysis revealed reduced enrichment of both R567W and R567W/R566C mutant proteins for all tested probes, including the U, C, U + C and mutant U + C motif probes (Supplementary Fig. 6d, e). Collectively, these in vitro findings confirm that the CTCF[R567W]-mutant ZFs exhibit a diminished capacity to bind DNA motif probes compared to wild-type ZFs.

To assess the alterations in chromatin binding affinity of the CTCF[R567W]-mutant protein in vivo, we conducted adjusted chromatin immunoprecipitation sequencing (ChIP-seq) experiments on brain, heart, and lung tissues in *Ctcf*[+/+] and *Ctcf*[R567W/R567W] mice using anti-CTCF antibody (Supplementary Fig. 7a). Surprisingly, we observed a reduction in CTCF binding at numerous sites in the CTCF[R567W] mutant group across all tissues (Fig. 4d and Supplementary Fig. 7b, c), suggesting weakened CTCF chromatin binding by CTCF[R567W]. Additionally, a greater reduction in CTCF binding corresponded to an increased enrichment of U CTCF motifs, specifically the ZFs 9–11 binding motifs, while the core consensus sequence remained invariant (Supplementary Fig. 7d).

To scrutinize the detailed binding properties affected by CTCF[R567W], we performed motif enrichment analysis on our ChIP-seq

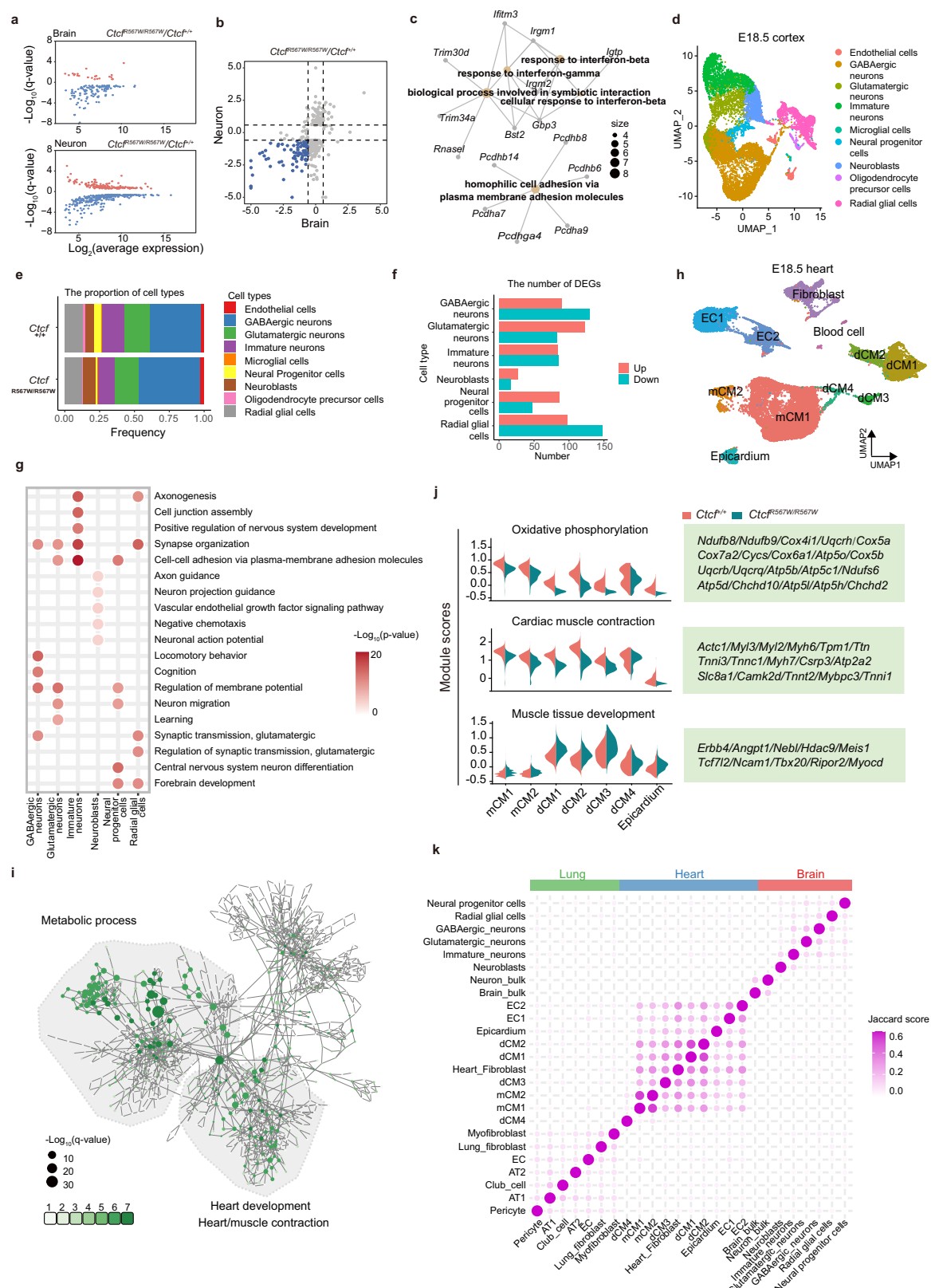

data using a previously described method[41]. Using brain tissues as an example, we categorized CTCF sites into four groups based on alterations in CTCF binding strength following the CTCF^R567W mutation. For each group, we extracted 20 bp upstream sequences of CTCF core consensus binding motif, which were then divided into 10 subclusters (Supplementary Fig. 7e). The sequence logo for each subcluster

indicated that sites with a greater decrease in CTCF binding upon mutation tended to contain a higher percentage of the U motif (group 1: 3.81%, group 2: 8.57%, group 3: 27.79%, group 4: 31.69%) (Supplementary Fig. 7e). In addition to the canonical CTCF U motif, we identified several U motif-like sequence logos, termed noncanonical U motifs, at the downregulated CTCF sites following the mutation. To

**Fig. 3 | Effect of CTCF[R567W] homozygous mutation on neurological and cardio-pulmonary development. a, b** Scatter plot showing DEGs detected by RNA-seq in whole brains and primary cultured neurons of E18.5 *Ctcf*[+/+] and *Ctcf*[R567W/R567W] mice. Significantly upregulated genes are shown in red, and significantly downregulated genes are shown in blue (**a**). Co-downregulated genes are shown in blue (**b**). **c** Network showing the GO analysis results of co-downregulated genes in (**b**). **d** UMAP visualization of snRNA-seq data and cell clustering results of cortex samples from E18.5 *Ctcf*[+/+] and *Ctcf*[R567W/R567W] mice. **e** Proportion of different cell types in the cortex of *Ctcf*[+/+] and *Ctcf*[R567W/R567W] mice. **f** Number of DEGs for each cell type in cortex samples of E18.5 *Ctcf*[+/+] and *Ctcf*[R567W/R567W] mice from snRNA-seq data. **g** GO analysis of DEGs for indicated cell clusters from cortex snRNA-seq in *Ctcf*[R567W/R567W]

mice compared to *Ctcf*[+/+] mice. GO analysis was carried out by clusterProfiler software with two-sided hypergeometric test. **h** UMAP visualization of snRNA-seq data and cell clustering results of heart samples from E18.5 *Ctcf*[+/+] and *Ctcf*[R567W/R567W] mice. Plots are colored by cell type. mCM, mature cardiomyocytes; dCM, developing cardiomyocytes; EC, endothelial cells. **i** GO network of DEGs for 7 myocardial-related cell types (CM1, CM2, dCM1, dCM2, dCM3, dCM4, epicardium). Sizes of the circles represent the level of enrichment of the GO term. A greener color indicates more cell types enriched in that term. **j** Average expression levels of indicated gene sets in different cell types of heart snRNA-seq data. **k** Correlation heatmap of DEGs from the indicated cell types based on the Jaccard score.

comprehensively assess the CTCF U motif, clustering analysis was performed using the upstream sequences of all downregulated CTCF sites in three tissues, revealing seven types of upstream motifs labeled from U1 to U7, with U1 corresponding to the canonical CTCF U motif (Fig. 4e and Supplementary Fig. 7f). Collectively, CTCF sites containing U motifs accounted for ~60% of downregulated CTCF sites (Supplementary Fig. 7g), and about 20–30% of total CTCF sites in each tissue, respectively (Supplementary Fig. 7h). The reduced CTCF sites post-CTCF[R567W] mutation in different tissues contained similar proportions of different types of U motifs (Fig. 4f). These results suggest that CTCF[R567W] partially reduces CTCF binding, potentially achieved by weakening its ability to bind the U motif.

## CTCF[R567W] leads to local reorganization of higher-order chromatin structures

CTCF mediates long-range chromatin interactions[42,43]. To investigate whether CTCF[R567W] disrupts chromatin organization, Bridge Linker-Hi-C (BL-Hi-C) was performed in the brain, heart, and lung tissues of E18.5 *Ctcf*[+/+] and *Ctcf*[R567W/R567W] mice. Quality control analysis revealed nearly identical distance decay curves and good reproducibility between replicates in each tissue (Supplementary Fig. 8a, b). The *Ctcf* mutation had little effect on compartments (Supplementary Fig. 8c) but noticeably altered the insulation score, implying potential changes to TADs (Fig. 4g and Supplementary Fig. 8d). Further analysis classified TADs into stable, fusion, separation, and confusion types (Fig. 4h). While most TADs remained unchanged (Fig. 4i), disturbed TADs were observed only in certain regions, exhibiting a tissue-specific alteration pattern (Fig. 4j). These results indicate that CTCF[R567W] reduces CTCF binding and locally alters chromatin structure, which may be linked to the disruption of gene expression.

## CTCF[R567W] mutation contributes to differential binding and disruption of enhancer-promoter interactions at the *Pcdhβ* locus

In our investigation into the connection between the CTCF[R567W] mutation-mediated chromatin structural variation and gene expression, we identified DEGs within TADs and invariant TAD boundaries. Notably, a majority of DEGs were located within stable TADs (Fig. 5a). Further analysis revealed a significant number of DEGs existed within the same TAD, with a particular focus on the *cPcdh* clusters, encompassing over 30 DEGs on chromosome 18 (Chr18) (Fig. 5b). Visualization of this locus exhibited a subdivision into two subTADs, one corresponding to *Pcdhα* and the other to *Pcdhβ* and *Pcdhγ*. Intriguingly, the subTAD containing *Pcdhβ* and *Pcdhγ* further fragmented into two subTADs post-CTCF[R567W] mutation (Fig. 5c).

A detailed examination of *cPcdhs* expression indicated a prevailing downregulation of genes within this locus, although some remained unchanged or exhibited upregulation (Fig. 5d). The genes at the *cPcdh* locus were characterized by CTCF binding sites in their promoters, interacting with CTCF on the distal enhancer to facilitate long-range interactions[44]. Strikingly, changes in CTCF binding strength at the *cPcdh* promoters precisely corresponded with the changes in expression of *cPcdh* genes, while CTCF binding only slightly reduced at the two enhancer elements (*Pcdhα-E* and *Pcdhβγ-E*) (Fig. 5d and

Supplementary Fig. 8e). Considering the cooperative role of CTCF and cohesin in mediating chromatin interactions[45], our examination of cohesin subunit RAD21 occupancy via ChIP-qPCR at the *cPcdh* locus in the brain tissue of *Ctcf*[+/+] and *Ctcf*[R567W/R567W] mice yielded consistent results with CTCF. This included decreased RAD21 at *Pcdhβ* promoters and enhancers, accompanied by an increased binding at *Pcdhγa5, b4,* and *b5,* respectively, indicating a downstream shift of chromatin loops away from the *Pcdhβ* locus (Supplementary Fig. 8f, g). QHR-4C experiments[46] demonstrated a significant weakening of the connections between the distal enhancer (SE-f) and the promoters of *Pcdhβ* genes in *Ctcf*[R567W/R567W] mice (Fig. 5e), suggesting the loss of multiple long-range interactions resulted in a noticeable alteration in the TAD structure of this locus, complementing the observed TAD splitting and isolation of the *Pcdhβ* locus into a subTAD (Fig. 5c, f).

The CTCF binding changes induced by the CTCF[R567W] mutation in the *cPcdh* locus were further elucidated. Notably, the overall binding strength of CTCF at the *cPcdh* locus was significantly weaker than the genome-wide average (Supplementary Fig. 8h), implying weak CTCF motifs. Motif analysis of CTCF core motifs and flanking 20 bp sequences revealed that all these CTCF sites possessed weak CTCF core motifs, and sites with decreased CTCF binding at the *cPcdh* locus were highly enriched with U motifs, while unchanged or increased CTCF binding sites were not (Supplementary Fig. 8i).

Together, the CTCF[R567W] mutation selectively reduces CTCF binding in the promoters of *Pcdhβ* genes and alters the chromatin structure of the *cPcdh* locus by decreasing EP interactions. This leads to a downregulation of *cPcdh* genes, potentially correlating with impairments of neural function.

## CTCF[R567W] mutation leads to developmental anomalies of hESC-derived cortical organoids

To evaluate the functional consequences of the CTCF[R567W] mutation within the human cell system, we employed CRISPR/Cas9 technology to introduce the mutation into hESCs (Supplementary Fig. 9a). Two clones each for wild-type, heterozygous, and homozygous cells were successfully generated, confirmed through PCR and Sanger sequencing of both genomic DNA and cDNA (Supplementary Fig. 9b–d; cloning and validation details were shown in the Methods section). Notably, our investigation revealed that the CTCF[R567W] mutation had no effect on *CTCF* expression or splicing patterns, aligning with our observations in the mouse model across diverse tissues (Supplementary Fig. 9e, f).

Utilizing wild-type and CTCF[R567W] hESCs, we initiated the generation of cerebral cortical organoids following the established protocols[47,48]. In contrast to wild-type organoids, the homozygous (CTCF[R567W/R567W]) organoids displayed significant developmental defects, manifesting as poor organization, reduced survival rates beyond 2 months, and substantially smaller sizes during differentiation, as evidenced by images captured at days 20, 30, and 60, respectively (Fig. 6a, c, d). Conversely, the heterozygous (CTCF[+/R567W]) organoids initially exhibited smaller sizes but reached dimensions comparable to those of wild-type organoids by days 60 and 90 (Fig. 6a–d).

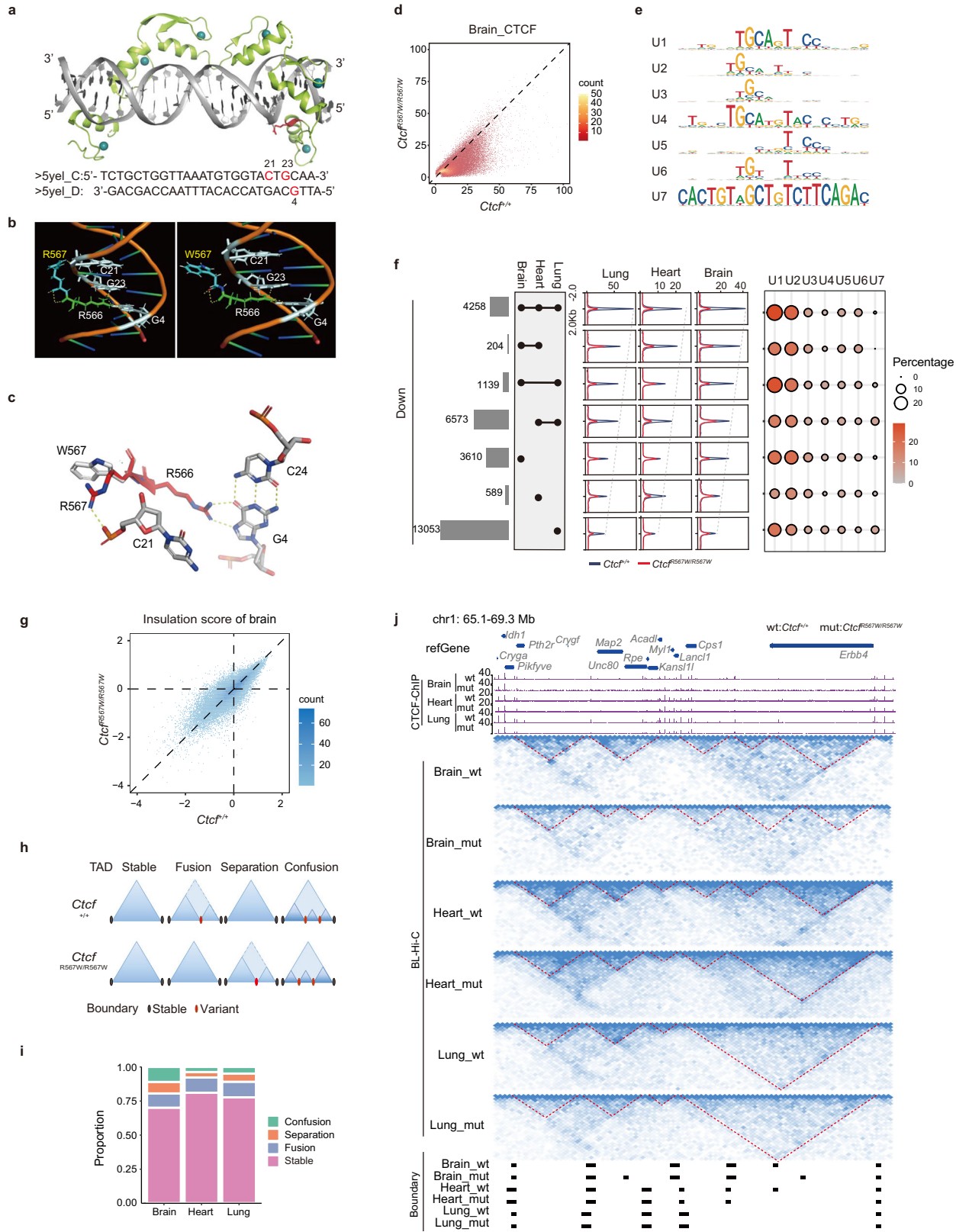

Furthermore, *CTCF^{R567W/R567W}* organoids exhibited reductions in the levels of both the neural progenitor marker SOX2 after 1 month of differentiation and the projected neuronal marker SATB2 at 3 months, indicative of early depletion of stem cells/progenitor cells and abnormalities in neurodevelopment (Supplementary Fig. 9g). Notably, excitatory neuronal marker genes (e.g. *VGLUT1, VGLUT2, and TBR1*) were abnormally downregulated (Supplementary Fig. 9h), while

inhibitory neuronal marker genes (including *GAD1, GAD2, VGAT*, etc.) were abnormally upregulated in *CTCF^{+/R567W}* organoids compared with wild-type organoids (Supplementary Fig. 9i), mirroring findings observed in single-cell data from the mouse cortex.

Concurrent with the observed developmental anomalies, we performed RNA-seq experiments on *CTCF^{+/+}, CTCF^{+/R567W},* and *CTCF^{R567W/ R567W}* hESC-derived organoids on day 35 (Supplementary Fig. 10a).

**Fig. 4 | CTCF^R567W weakens CTCF binding to sites containing upstream motifs and locally reorganizes higher-order chromatin structures. a** Crystal structure of the CTCF ZFs6–11-gb7CSE (CTCF-binding conserved sequence element within *Pcdhyb7* promoter) complex obtained from PDB: 5YEL[28] showing Zn$^{2+}$ ions in cyan, DNA in gray, ZFs6–11 in green, and the R567 residue in red. Corresponding DNA sequences are shown at the bottom. **b** Molecular dynamic simulations of the CTCF-DNA complex. Wild-type and R567W mutant structures with R567/W567 residues are shown in blue and with R566 residues in green. Structures also show hydrogen bonding (yellow dashed lines) between R566/R567 and DNA bases C21, G23, and G4 (gray). C: cytosine; G: guanine. **c** Overlay of wild-type (red) and R567W mutant (white) CTCF-DNA structures showing changes in hydrogen bonding (yellow dashed lines) between the R567/W567 residue and surrounding DNA bases and phosphate backbone upon mutation. Structures of (**a**–**c**) were visualized using the PyMoL Molecular Graphics System[91]. Hydrogen bonds are shown as dashed lines.

**d** Comparison of normalized CTCF binding strength for brain tissues between *Ctcf*^+/+ and *Ctcf*^R567W/R567W mice. **e** CTCF U motifs of downregulated CTCF sites derived from Supplementary Fig. 7f. **f** Overlap analysis of downregulated CTCF binding sites (log$_2$(fold change of *Ctcf*^+/+/*Ctcf*^R567W/R567W) > 1) in brain, heart and lung tissues (left). Average CTCF binding strengths (middle) and ratios of different types of U motifs (right) were analyzed for each set of overlapping peaks. **g** Comparison of genome-wide insulation scores in brain tissues between *Ctcf*^+/+ and *Ctcf*^R567W/R567W mice. The color in the plot represents the density of points. **h** Models depicting different types of TADs caused by the CTCF^R567W mutation. **i** Bar plot showing proportions of different types of TADs in brain, heart, and lung tissues. **j** Hi-C and CTCF ChIP-seq tracks illustrating the stable and variant TADs among brain, heart and lung tissues between *Ctcf*^+/+ and *Ctcf*^R567W/R567W mice. TAD boundaries identified for each tissue are shown at the bottom.

Unlike the limited number of DEGs observed in mouse models, both *CTCF*^+/R567W and *CTCF*^R567W/R567W cortical organoids exhibited hundreds of DEGs compared with the wild-type organoids (Supplementary Fig. 10b–d). Co-upregulated genes in both mutant organoids were associated with mitochondrial metabolism-related pathway, while co-downregulated genes primarily participated in developmental processes, including forebrain development, axonogenesis, and stem cell population maintenance (Supplementary Fig. 10e). These down-regulated pathways elucidate the phenotypes of small size, stem cell depletion, and abnormal neurodevelopment in the mutant organoids, consistent with the clinical manifestations of growth retardation and intellectual impairment in heterozygous *CTCF* mutant cases.

Moreover, in line with the data presented in the mouse model, we observed extensive downregulation of *cPCDH* genes in both *CTCF*^+/R567W and *CTCF*^R567W/R567W organoids (Supplementary Fig. 10f), underscoring the conservation of the effects induced by the CTCF^R567W mutation at this locus. These findings indicate that hESC-derived CTCF^R567W heterozygous cerebral cortical organoids may also exhibit developmental irregularities, supporting the involvement of the R567W mutation in *CTCF*-related clinical developmental disorders.

### CTCF^R567W heterozygous mutation expedites depletion of stem-like cells and prematurely drives development of hESC-derived cortical organoids

To assess if organoids with solely CTCF^R567W heterozygous mutation manifested comparable functional consequences to those observed in the snRNA-seq data of the mouse cortex, we conducted scRNA-seq on *CTCF*^+/+ and *CTCF*^+/R567W cortical organoids at 1- and 2-month differentiation intervals, encompassing a total of 71,955 individual cells. In order to contextualize our organoid data, we integrated the scRNA-seq data information with the human fetal developmental brain scRNA dataset from the Allen Brain Atlas. Notably, our organoid data aligned well with the data from the fetal brain cortex (Supplementary Fig. 10g, h). Subsequently, we categorized the cells in the integrated organoid dataset into distinct subgroups such as erythrocytes, fibroblasts, glioblasts, immune cells, neural crest cells, neuroblasts, neurons, neuronal intermediate progenitor cells (IPCs), oligos, radial glia, and vascular cells (Fig. 6e–g). Cells of unidentified subtypes were further annotated based on specific cell markers, utilizing *INA* (a neuronal marker gene) and *SLC32A1* (a GABAergic marker gene) (Supplementary Fig. 10i).

Our results unveiled significant differences in cell type proportions between the *CTCF*^+/+ and *CTCF*^+/R567W organoids at the 1 month mark. A notable rise in GABAergic neurons and a simultaneous decline in radial glial cells were observed (Fig. 6h). This trend was consistent with the visualization of the *CTCF*^+/+/*CTCF*^+/R567W cell proportion for each cell type in the 1 month organoids (Fig. 6i). Intriguingly, even at 1 month, *CTCF*^+/R567W organoids exhibited a substantial number of GABAergic cells, with proportions intermediate between the 1 month and 2 month *CTCF*^+/+ organoids. By 2 months, the cell type proportions

between the two groups began to converge (Fig. 6i). Our differential gene expression analysis between *CTCF*^+/+ and *CTCF*^+/R567W cell types revealed a decreased in the number of DEGs during differentiation from the 1 month to the 2 month, indicating that the gene expression differences are more pronounced in the early stages of organoid differentiation (Supplementary Fig. 10j). Enrichment analysis of the of 1 month data revealed that the *CTCF* heterozygous mutation affects numerous neural development pathways, including forebrain development, axonogenesis, and neuron projection guidance (Fig. 6j). We identified 77 genes that overlapped with risk genes associated with brain development disorders in GWAS, including the *RBFOX1* and *RUNX1T1* genes, which were differentially expressed in all cell types and associated with ASD (Supplementary Fig. 10k, l).

Our findings establish that the CTCF^R567W heterozygous mutation in hESC-derived cortical organoids accelerates the depletion of stem-like cells and triggers premature development of GABAergic neurons, mirroring the effects observed in the mouse cortex at the single-cell level. These perturbations in cellular composition and neural development-related pathways offer crucial insights for understanding specific clinical neurodevelopmental disorders.

## Discussion

In this study, we established a murine model to simulate clinical developmental disorders associated with the CTCF^R567W mutation, revealing the indispensability of CTCF-R567 in normal development. Given the presence of the CTCF^R567W mutation in clinical cases of neurodevelopmental disorders and the respiratory distress observed in our homozygous mice, our focus centered on investigating the effects of CTCF^R567W on brain and cardiopulmonary tissues. Our results suggest that the impact of CTCF^R567W on genes is cell-type specific, implying a broad influence on neurodevelopment and respiratory function, with potential implications for hematopoiesis, immunity, and other systems requiring further investigation.

The lack of homozygous individuals with *CTCF* mutations may be attributed to the dose-dependent function of CTCF, leading to the inability of homozygotes to survive. This supposition is congruent with the severe defects in hESC-mediated cortical organoids containing the homozygous mutation. Furthermore, the suboptimal performance of heterozygous mice on the rotarod hinted at minor abnormalities in cardiorespiratory function, aligning with the phenotype observed in some human patients with *CTCF* heterozygous mutations[17,18,20,21]. However, direct evidence linking the CTCF^R567W mutation to cardiac irregularities remains unclear and requires further investigation. Concurrently, despite our heterozygous mice could not fully phenocopy the clinical neurodevelopmental disorders, the observed cellular imbalance in heterozygous hESC-derived cortical organoids may offer essential insights into these specific disorders.

CTCF^R567W heterozygous organoids displayed an early depletion of stem-like cells and an increase in GABAergic neurons compared to wild-type organoids. This aligns with our findings in snRNA-seq data of

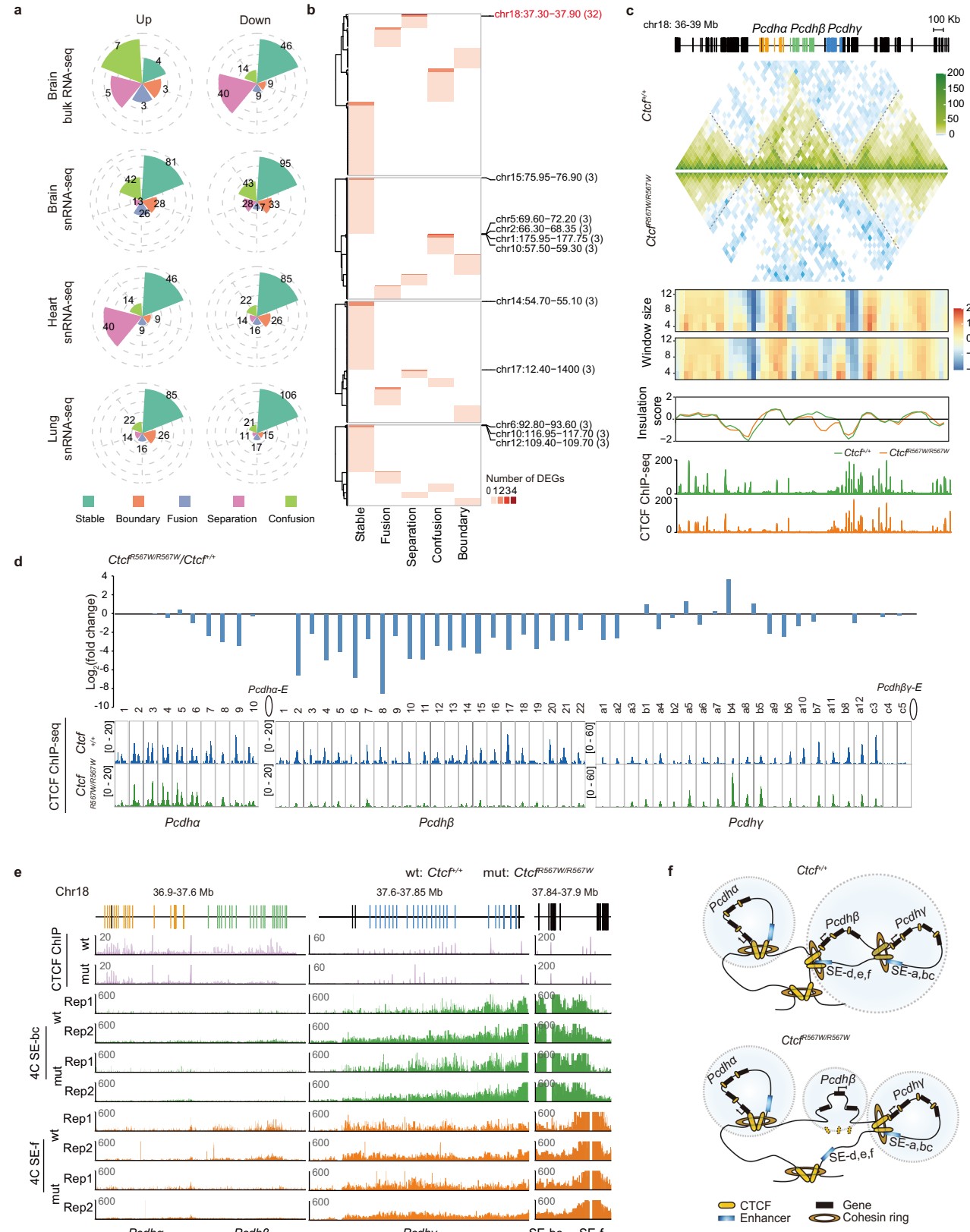

**Fig. 5 | CTCF^R567W mutation contributes to differential binding and disruption of EP interactions in the *Pcdhβ* locus. a** Correlation of DEGs positions with TAD boundaries and four types of TADs post CTCF^R567W mutation in different tissues. **b** Heatmap showing the number of DEGs located in one TAD or TAD boundary. The genomic location (Mb) of selected TADs or TAD boundaries are shown on the right with the associated number of DEGs. **c** Tracks showing Hi-C maps, insulation scores with different window sizes and normalized CTCF ChIP-seq signals in brain tissues of *Ctcf^+/+* and *Ctcf^R567W/R567W* mice at the cPcdh locus. **d** Fold change in cPcdh gene expression in brain tissue RNA-seq data (top) and corresponding CTCF binding strength (bottom) in the promoter of each gene in brain tissues of *Ctcf^+/+* and *Ctcf^R567W/R567W* mice. *Pcdhα*-E: enhancer for *Pcdhα*; *Pcdhβγ*-E: enhancer for *Pcdhβγ*. **e** 4C tracks showing the interactions between promoters and distal enhancers of cPcdh in brain tissues of *Ctcf^+/+* and *Ctcf^R567W/R567W* mice. SE: super enhancer. **f** Schematics summarizing the CTCF^R567W mutation in regulating higher-order chromatin structure at the cPcdh locus.

homozygous mutant mice, suggesting that CTCF[R567W] mutation may disrupt the differentiation process of stem-like cells into specific neurons in both human and mouse systems. This finding corresponds with previous reports indicating that CTCF is crucial for maintaining a balance between neural progenitor cell proliferation and differentiation during early forebrain development[16]. Recent scRNA-seq data on cortical organoids of autism risk genes also identified an increase in GABAergic neurons[48], suggesting that *CTCF* mutations, similar to autism risk factors, may induce neurodevelopmental disorders through shared cellular development pathways.

The single amino acid mutation, R567W within ZF11 of CTCF, significantly impacts CTCF binding to chromatin, particularly at sites containing the upstream motif. This finding is consistent with earlier findings indicating that mutations within ZF11 of CTCF primarily affect sites with this motif, yet over 60% of binding sites are maintained[39,49]. In our study, we took advantage of adjusted ChIP-seq methods for CTCF in mouse tissues and found that a single point mutation could significantly alter the chromatin functions of CTCF, which had the similar effect as observed after ZF11 deletion. It's important to note that individual heterogeneity and variations in methodologies and systems used across different studies could also lead to the certain biases. The altered binding specificity of CTCF[R567W] at sites with the U motif can be attributed to the essential functions of ZFs 9–11 in recognizing this motif and flanking sequences involved in boundary insulation[50]. This finding underscores the requirement of R567 for sequence-specific DNA-binding activity and the chromatin function of CTCF, which ultimately influences specific biological phenotypes. Similar to other CTCF ZF mutations, alterations in DNA binding at specific loci can potentially lead to a spectrum of phenotypes, ranging from loss to gain-of-function effects[51].

CTCF[R567W] exerts a broad and diverse impact on gene expression across distinct cell types, with DEGs potentially influenced by direct or indirect downstream effects of the CTCF[R567W] mutation. These effects may accumulate and become more evident at E18.5 during extended embryonic tissue development. Notably, among these DEGs, a substantial downregulation is observed in most *cPcdh* genes, both in human and mouse models, consistent with previous findings in *Ctcf*-conditional-knockout mice[23,24]. Dysregulation of *cPcdh* is strongly associated with abnormal neural functions and deletion of this cluster proves fatal at birth in mice[35,52,53].

In the mouse model, CTCF[R567W] leads to a global reduction in binding at the promoters of *Pcdhβ* cluster genes and induces changes in local 3D organization by dividing one subTAD covering the *Pcdhβ* and *Pcdhγ* regions into two subTADs. Transcriptome data from human organoids also reveal a significant downregulation of *PCDHB* genes. Given the highly conserved regulatory role of CTCF at this locus in humans and mice, we opted not to explore further the potential regulatory mechanisms in the complex organoid system. It is important to note that *cPcdh* encodes neuronal identity and exhibits single cell-specific expression pattern[34]. The congruence of regulation and chromatin conformational changes mediated by CTCF[R567W] at the individual neuron level with patterns found in bulk cells warrants further investigation. Our analysis primarily focused on the *cPcdh* locus in the nervous system at the regulatory mechanism level. However, the impact of the CTCF[R567W] mutation on the regulatory mechanisms of other tissue-specific differentially expressed genes, particularly those in the heart and lung tissues, remains an avenue for future study.

In conclusion, successful engagement of peripheral CTCF ZFs such as ZF11 containing DNA-binding residue R567 is essential for normal development across various tissue types. This study provides valuable disease models for further exploration into the pathogenesis of clinically relevant *CTCF* mutations. Furthermore, these models hold potential for aiding in the development of therapeutic interventions by manipulating the 3D chromatin structure.

# Methods

The animal experiments were performed in accordance with the Guidelines for the Care and Use of Laboratory Animals of the National Institutes of Health. The protocol was approved by the Committee on the Ethics of Animal Experiments of Guangzhou Institutes of Biomedicine and Health, Chinese Academy of Sciences (IACUC: 2023085). The use of human H1 ESCs (WiCell, WA01) in this study is compliant with the Guidance of the Ministry of Science and Technology for the Review and Approval of Human Genetic Resources and was approved by the Life Science and Medical Ethics Committee of the Guangzhou Institutes of Biomedicine and Health under license number GIBH-LMEC2023-024-01 (AL).

## Mice

Mice were accommodated in a specific pathogen-free (SPF) facility, maintained under controlled conditions of 20 °C temperature and 50% humidity, with a 12 hour (h) dark/light cycle. Food and water were provided ad libitum. C57BL/6 N mice (Cyagen Biosciences, Guangzhou, China) harboring the CTCF[R567W] mutation were generated via CRISPR/Cas9-mediated genome editing. The single guide RNA (sgRNA) targeting *Ctcf* was inserted into the gRNA-spCas9 cloning vector. Single-strand oligodeoxynucleotides (ssODNs) with targeting sequences and mutation sites were synthesized and flanked by 60–80 bp homologous sequences. *Ctcf* gRNA and Cas9 mRNA were produced by in vitro transcription and coinjected into fertilized eggs with ssODN donors to obtain mutant mice. The mice were identified by PCR and DNA sequencing for genotyping. Mice carrying the *Ctcf* mutation allele were mated with wild-type C57BL/6 N mice to obtain heterozygous offspring, which were then crossed to generate *Ctcf*[R567W/R567W] mice. Great efforts have been made to ensure that genetics and environment are comparable among groups in this study. When the number of mice needed for an experiment was small enough to be obtained from a single litter, littermate mice were used for comparisons between genotypes. For experiments requiring larger numbers of mice, pups were selected from two or more litters bred during the same period and housed them under identical conditions. The targeted sgRNA and ssODN sequences for genome editing and the genotyping primers are listed in Supplementary Tables 1 and 2.

## Plasmid construction

For construction of the sgRNA-Cas9 vector for gene editing, sgRNA targeting human *CTCF* was synthesized and constructed into the pX459 vector (pSpCas9(BB)-2A-Puro, Addgene Cat#62988). For construction of the prokaryotic expression vector, the sequence for CTCF ZF fragments was amplified from the cDNA of mESCs and then cloned into the pGEX-4T-2 vector, and the desired mutations were introduced with site-directed mutagenesis to construct the pGEX-ZF-wt, pGEX-ZF-mut (R567W) and pGEX-ZF-dmut (R566C, R567W) vectors.

## Tissue processing and cell culture

Pregnant mice were anesthetized with 1.2% v/v avertin (2,2,2-tribromoethanol) (Sigma, Cat# T48402) and sacrificed by cervical dislocation. The embryos were removed, and the desired fetal mouse tissues were taken immediately after decapitation. For brain tissue retrieval and neural cell culture, the whole brain of the fetal mouse was removed by dissection, the meninges were stripped, and the cerebral cortex and hippocampus were separated. Tissues were digested by using the papain dissociation system (Worthington, Cat# LK003160) according to the manufacturer's instructions. Cells were filtered through a 70 μm filter membrane, counted and inoculated in culture dishes coated with poly-D-lysine (PDL) (Sigma, Cat# P6407). Primary neurons were first cultured in medium containing DMEM/F12 (Gibco, Cat# 11130032) supplemented with 10% v/v fetal bovine serum (FBS) (Vistech, SE100-B) and 1% v/v penicillin/streptomycin (Gibco, Cat# 15140122) at 37 °C in a 5% $CO_2$ incubator. The medium was replaced

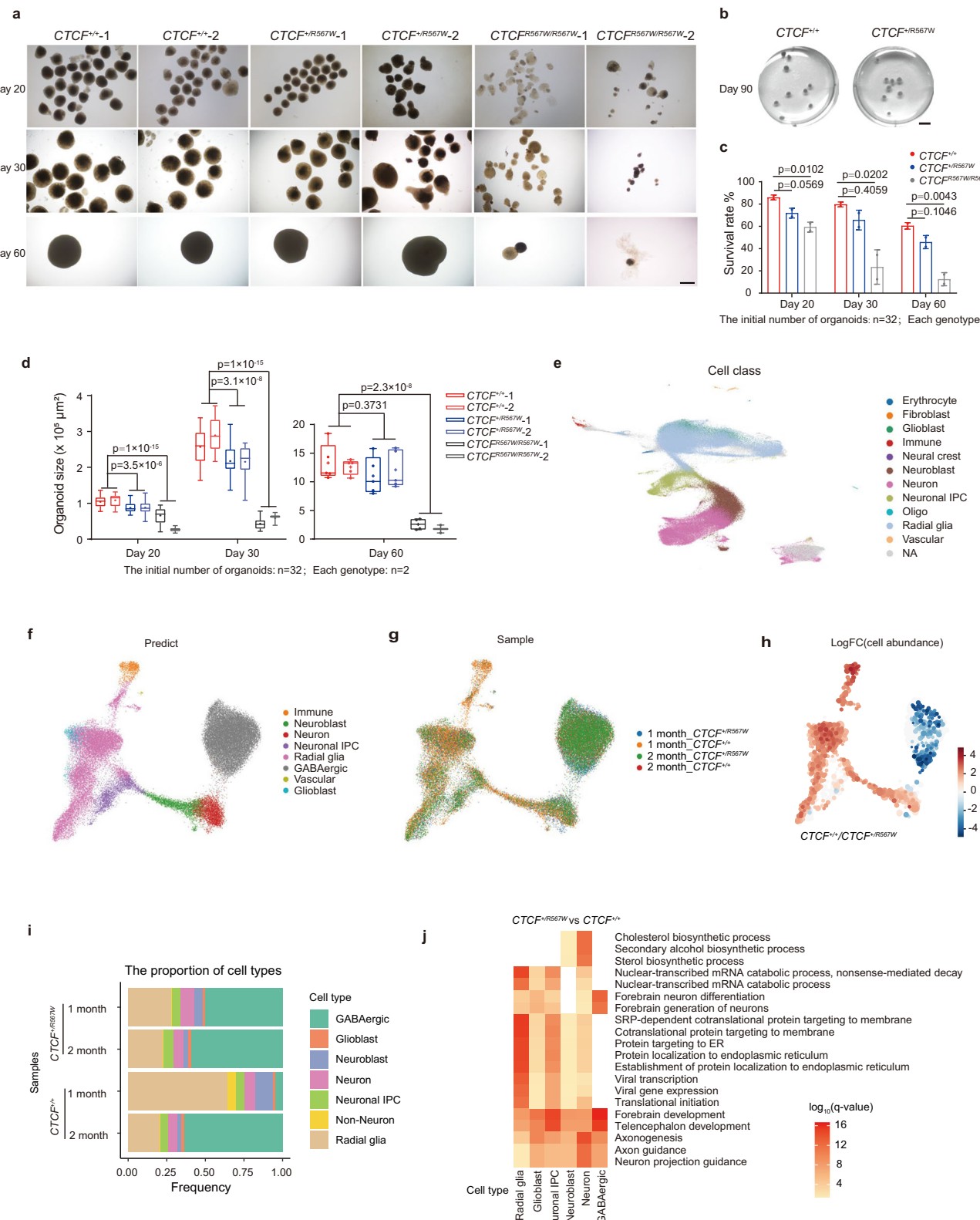

after 3–4 h with neurobasal medium (Gibco, Cat# 21103049) supplemented with 2% v/v B27 (Gibco, Cat# 17504044), 1% v/v GlutaMAX (Gibco, Cat# 35050061) and 1% v/v penicillin/streptomycin. The medium was changed every other day. HEK293T cells were cultured in high-glucose DMEM (HyClone, Cat# SH30022.01) supplemented with 10% v/v FBS. Feeder-free H1 hESCs (WA01) (WiCell) were cultured in PGM1 human pluripotent stem cell culture medium (Cellapy, Cat#

CA1007500) in Matrigel (Corning, Cat# 354277)-coated culture plates. Then, the cells were digested into clumps with 0.5 mM EDTA for normal culture or digested into single cells with Accutase for passaging (STEMCELL Technologies, Cat# 07920) and cultured with medium containing 10 μM ROCK inhibitor Y-27632 (Sigma, Cat# SCM075) for one day; then the inhibitor was withdrawn, and cells were cultured in normal medium to form individual clones.

**Fig. 6 | Effect of CTCF$^{R567W}$ mutation on cortical organoid development in hESC-derived models. a, b** Representative photographs (**a**: days 20, 30, 60; **b**: day 90) of the *CTCF$^{+/+}$*, *CTCF$^{+/R567W}$*, and *CTCF$^{R567W/R567W}$* organoids. Scale bars in (**a**): 500 μm, scale bars in (**b**): 5 mm. **c, d** Survival rate (**c**) and size quantification (**d**) (95% confidence interval) of *CTCF$^{+/+}$*, *CTCF$^{+/R567W}$* and *CTCF$^{R567W/R567W}$* organoids on days 20, 30 and 60, respectively. **e** UMAP visualization of scRNA-seq data integrated from cortical organoids and fetal brain cortices, colored by cell type. **f, g** UMAP visualization of cortical organoid scRNA-seq data after integration, colored by predicted cell type (**f**) and sample (**g**). **h** UMAP visualization of the log-fold change in cell proportions between *CTCF$^{+/+}$* and *CTCF$^{+/R567W}$* cells in 1 month cortical organoid scRNA-seq data. **i** Distribution of cell type proportions among cortical organoid scRNA-seq data from 1 month and 2 month *CTCF$^{+/+}$* and *CTCF$^{+/R567W}$* samples. **j** GO analysis of DEGs for each cell type from scRNA-seq in *CTCF$^{+/R567W}$* organoids compared to scRNA-seq in *CTCF$^{+/+}$* organoids. The starting number of organoids was 32 for each genotype (**a–d**), each genotype contains two clones. Quantitative data are presented as mean ± SD. *p*-values from one-way ANOVA with Dunnett's multiple comparisons test adjustment are indicated in the graphs. Experiments in (**a–d**) were repeated independently three times with similar results. All Box and whisker plots show lower and upper quartiles (box limits), median (center line) and minimum to maximum values (whiskers). Source data are provided as a Source Data file.

## Generation of CTCF$^{R567W}$ point mutation-containing hESC lines

Human H1 cells harboring the CTCF$^{R567W}$ mutation were constructed via CRISPR/Cas9. sgRNAs were designed with an online tool (http://benchling.com). The ssODN containing the desired mutation and enzymatic cleavage site was used as a template for homologous repair. The sgRNA primers were synthesized, annealed, cloned and inserted into the pX459 vector. The Cas9-sgRNA vector and ssODN were transfected into cells for gene editing using FuGENE HD (Promega, Cat# E2311) transfection reagent. Since Cas9-sgRNA vector contained puromycin resistant gene, the cells were screened with medium containing 1 μg/mL puromycin (Gibco, Cat# A11138-03). After 48 h of screening, the surviving cells were digested into single cells using Accutase for passaging into Matrigel-coated 6-well plates. After ~10 days of culture, the single cells were successfully expanded into individual clones. Over one hundred clones were observed under a microscope and manually picked out using a micropipette. The selected clones were transferred to 96-well plates and further cultured for ~3–5 days until they reached a sufficient size. Subsequently, the clones were digested using 0.5 mM EDTA in PBS to detach them from the culture plates for passaging and amplification. Once the clones reached a certain quantity, a portion of the cells was collected for subsequent genomic DNA extraction and PCR genotyping, followed by enzyme digestion and Sanger sequencing for identification of correctly edited clones.

Detailed information on clones and editing strategies: The first round of editing in wild-type cells using sgRNA1 (target: AACATTTA-CACGTCGGGTAA) and ssODN1 (introducing the EcoR I site) yielded several heterozygous clones, which were verified by genotyping to have one allele that was correctly mutated and the other allele with a 6 bp deletion. These clones were referred to as heterozygous clone 1. Several wild-type clones were also retained after transfection and were selected as editing control wild-type clones. In the second round of editing, sgRNA2 (target: TGGGAAAACATTTACACGTC) and ssODN2 (introducing the Mlu I site) or ssODN3 (introducing the EcoR I site) were used to edit heterozygous clone 1 and wild-type cells, respectively. This process yielded homozygous clones 1 and 2, and heterozygous clone 2. The clones were verified by genomic PCR, enzyme digestion, and Sanger sequencing. cDNA PCR (with primers for exons 8–10 [aa 469–644]) and sequencing also confirmed that the clones were correct (Supplementary Fig. 9a–d). The sequences of the targeted sgRNAs, ssODNs and genotyping primers used are listed in Supplementary Tables 1 and 2.

## RNA extraction and RT-qPCR

Total RNA was extracted from cells and tissues with an RaPure Total RNA Micro Kit (Magen, Cat# R4012-03) according to the manufacturer's instructions. For cells, $1 \times 10^5 - 1 \times 10^6$ cells were counted and collected. For tissues, the isolated mouse tissues were immediately crushed with a homogenizer. RNA quantification and reverse transcription were performed with HiScript III RT SuperMix for qPCR (+ gDNA wiper) (Vazyme Biotech, Cat# R323-01). Real-time PCR was performed with SYBR Green mix (Genstar, Cat# A301-01) on a CFX96 real-time PCR system (Bio-Rad) according to the manufacturer's instructions. *Gapdh* mRNA was used as an internal control for normalization of gene expression. The primers used in this study are listed in Supplementary Table 3.

## Protein extraction and immunoblotting

Isolated mouse tissues or cultured cells were lysed with RIPA buffer (1% v/v Triton X-100, 0.1% w/v sodium dodecyl sulfate (SDS), 150 mM KCl, 50 mM Tris-HCl (pH 7.4), 1 mM EDTA, 1% w/v deoxycholic acid (sodium salt)) supplemented with protease inhibitors (PIC) (Bimake, Cat# B14001) followed by sonication. The total protein supernatant was obtained by high-speed centrifugation, and the samples were separated on SDS-PAGE gels and transferred to polyvinylidene fluoride (PVDF) membranes (Millipore, Cat# IPVH00010). The membranes were blocked with 5% w/v skim milk and incubated with primary and secondary antibodies before being washed with TBS-T buffer (20 mM Tris-HCl (pH 7.5), 150 mM NaCl, 0.1% v/v Tween-20). Finally, the membranes were assessed with an enhanced chemiluminescence (ECL) Kit (NCM Biotech, Cat# P10300). The antibodies used in this study are listed in Supplementary Table 6.

## Histology, immunofluorescence, and image acquisition

Intact mouse tissues were fixed with 4% w/v paraformaldehyde (Beyotime Biotechnology, Cat# P0099) for 24 h and then made into 3 μm paraffin sections per standard protocols. Hematoxylin and eosin (H&E) staining was performed according to the manufacturer's instructions (Servicebio, Cat# G1005). For immunohistochemistry, sections were treated with 0.3% v/v H$_2$O$_2$ for 15 min, washed with PBS, blocked with 1% w/v BSA containing 0.3% v/v Triton X-100 for 1 h, and incubated with antibodies at 4 °C. After washing 5 times with PBS, the sections were incubated with secondary antibodies and then stained with DAB and hematoxylin. Images were acquired with a Pannoramic MIDI slice scanner (3D HISTECH).

For immunofluorescence, samples were rinsed with PBS for 3 min, and fixed in 4% w/v paraformaldehyde for 10 min. Following PBS rinses and permeabilization in 0.5% v/v Triton X-100 for 15 min, samples were blocked in 1% w/v BSA/PBS for 1 h. Next, primary antibodies diluted in 1% w/v BSA/PBS were added and incubated for 2 h at room temperature (RT). Following the removal of the primary antibody and subsequently wash with PBS, secondary antibodies were added at a 1:500 dilution in 1% BSA/PBS and incubated for 1 h at RT. After washing with PBS three times for 5 min each time, the samples were stained with DAPI/PBS (1:1000) for 10 min and washed with PBS. The antibodies used in this study are listed in Supplementary Table 6.

For image acquisition, the samples were analyzed by using a Pannoramic MIDI slice scanner (3D HISTECH, CaseViewer 2.4.0, 20x objective), a light microscope (OLYMPUS IX73, cellSens software, 40x objective) or a LSM 710 NLO confocal laser scanning microscope (Carl Zeiss Technology, ZEN 2012 software, 40x objective). Sholl analysis[54] was performed on individual neurons by using ImageJ software to quantify their morphological characteristics.

## Golgi staining

E18.5 mice were delivered by cesarean section as described above. Golgi staining experiments were conducted using an FD Rapid GolgiStain™ Kit (FD Neuro Technologies, Cat# PK401-50) according to the

kit instructions. Briefly, the brains of E18.5 mice were removed as quickly as possible, soaked in an equal-volume mixture of solutions A and B while protected from light at RT for 2 weeks and then transferred to solution C for another 3 days. The impregnated brains were sliced using a vibratome (Leica VT1200S) at a thickness of 100 μm. The sections were immersed in a mixture of solutions D and E for 10 min, dehydrated, cleaned with xylene and then sealed with resin sealing tablets. Images of brain regions of interest were taken by bright-field microscopy.

## Intracellular Ca²⁺ imaging

Neurons from the cortex or hippocampus were loaded with Fluo-4-AM (4 μM) (Yeasen, Cat# 40704ES72) in culture medium at 37 °C for 30 min. Afterward, the neurons were washed with PBS three times, and culture medium was added for the following measurements. The relative fluorescence intensity was measured with a fluorescence microscopic system equipped with a FITC filter. During the measurements, KCl was added to a final concentration of 30 mM to depolarize the neurons and evoke Ca²⁺ influx. The recorded images were quantified with ImageJ software.

## Multielectrode array (MEA) measurements

MEA experiments were recorded using the Maestro Pro system (Axion Biosystems). Briefly, brain tissues were isolated and digested as described above. Approximately $1 \times 10^5$ primary neurons were inoculated into PDL-coated 48-well MEA plates (Axion Biosystems, Cat# M768-tMEA-48W), and primary neural cells were cultured as described above. The spontaneous neural electrical activity of the cultured cells was recorded weekly, and the medium was changed 24 h prior to recording. The plates were loaded into the Maestro Pro instrument, and electrical activity was recorded by AxIS Navigator software for 15 min, measuring the number of active electrodes (electrodes with an average of 5 spikes per min), the number of spikes, the number of bursts (individual electrodes with > 5 spikes in 100 ms), the number of network bursts (35% of electrodes with > 50 spikes in 100 ms simultaneously), the synchronization index (with a synchrony window of 20 ms) and other metrics. The data were exported using the Neural Metric Tool and analyzed with GraphPad Prism 9 software.

## Mouse behavioral tests

Mice were group-housed (5–8 animals per cage) in standard filter-top cages with ready access to water and rodent food and maintained on a 12 h light/dark cycle with 50–60% relative humidity. Mouse behavioral experiments were performed using 8–12 week-old male mice as described previously[55–57]. During data collection, investigators were blinded to the group allocation of the mice to ensure the unbiased results. Each apparatus was cleaned with 75% v/v alcohol spray prior to testing each animal to prevent bias due to olfactory cues.

**Open-field test (OFT).** Each mouse was placed in an open-field arena (44.5 cm × 44.5 cm × 44.5 cm) and allowed to explore freely for 15 min. The activity of the mice in the box was automatically recorded with a camera and the animal behavior analysis software EthoVision XT 13 (Noldus), and each mouse was tested only once.

**Novel Object Recognition test (NOR).** NOR experiments were performed in the same open-field arena (44.5 cm × 44.5 cm × 44.5 cm) as the OFT. During the habituation phase, mice were allowed to freely explore the arena and two identical objects for 5 min. The mice were then placed back in the home cage for 1 h as a time interval sufficient for detection of memory in the next phase. During the testing phase, one of the objects was replaced with a new object, and mice were allowed to explore freely for 5 min. The exploration time and times as well as the distance from each object (touching the object with the mouth or nose and approaching the object within ~2–3 cm were considered as exploration of the object) were automatically recorded with a camera and the animal behavior analysis software EthoVision XT 13 (Noldus).

**Three-chamber test for social interaction and novelty behavior.** The test apparatus consisted of a rectangular three-chambered box. Each chamber measured 20 × 40 × 22 cm and was evenly separated by walls with two retractable doors (each door 3 × 5 cm), allowing mice access to each chamber. An unfamiliar male mouse (stranger 1), which had no prior contact with the subject mice, was placed in one of the side chambers. This unfamiliar mouse was enclosed in a small, round wire cage that enabled nose contact between the bars but prevented fighting. The wire cage was 11 cm high and 9 cm in diameter at the bottom, with vertical bars spaced 0.5 cm apart. Subject mice were first placed in the middle of the chamber and allowed to freely explore throughout the test box for 10 min. The time spent around the cage was measured with the aid of a camera mounted on top of the box to quantify of social preference for stranger 1. A second unfamiliar male mouse (stranger 2) was placed in an identical small wire cage on the other side of the chamber. Thus, the test mice could choose between the first unfamiliar mouse that had already been studied (stranger 1) and a new unfamiliar mouse (stranger 2). As before, the test mice were free to move around in the three chambers for 10 min. The animal behavior analysis software EthoVision XT 13 (Noldus) automatically recorded the time and number of mutual contacts between the test mice and Stranger 1 and Stranger 2.

**Elevated plus maze test (EPM).** The EPM apparatus consisted of two open arms and two closed arms (each 9.5 cm wide and 40 cm long) arranged in a cruciform pattern, with the intersection being the central zone, which was 55 cm above the floor. To assess anxiety, mice were placed in the central area, facing an open arm and allowed to explore freely for 5 min. The time and number of explorations in the open and closed arms were automatically recorded by EthoVision XT 13 (Noldus).

**Tail suspension test.** A cylindrical tube was placed over the tail of the mouse to prevent the mouse from holding its leg during suspension. One end of the tape was stuck to the tail of the mouse, and the other end was fixed to the hanging tail support. The distance from the head of each mouse to the platform was ensured to be the same. The suspension lasted for 6 min, and the immobility time during the last 4 min was recorded with a video camera and analyzed.

**Forced swim test.** Mice were placed in a 25 cm high, 10 cm diameter clear glass cylinder filled with water to a depth of 10 cm at 22 °C. A 6 min swimming test was performed on each mouse, and the last 4 min of immobility time was recorded for further processing.

**Morris water maze.** Mice were placed in a circular tank (137 cm in diameter) filled with water (22 ± 1 °C) and divided into quadrants with visual cues for spatial orientation. The circular platform (10 cm in diameter) was adjusted to be submerged 1 cm below the water surface, and the position of the platform was unchanged. Each mouse entered the water from one of the four quadrants at a time. Each mouse was gently placed into the water facing the wall of the tank and each trial lasted 1 min; the trial ended when the mouse climbed onto and stayed on the hidden platform for 10 s. The software automatically recorded the time taken by the mouse to reach the platform as the latency; if the mouse failed to find the hidden platform within 1 min, the mouse was guided to the platform and allowed to stay on the platform for 15 s. The mouse was removed and dried with a dry towel. Four trials were performed per mouse per day at intervals of 30 min or more for 6 consecutive days of training. On day 7, the platform was removed, and each mouse was placed in the water in the quadrant opposite the

target quadrant. The software was used to record the number of times each mouse crossed the platform as well as the time and distance it moved in the target quadrant within 1 min. The probe trial was performed twice.

**Rotarod test.** The rotarod test involved 3 training sessions per day for 4 days. The first day began with acclimatization of all experimental mice at a lower speed (4 r/min) to eliminate fear, with each mouse trained 3 times for 5 min each at intervals of 30 min or more. The next 3 days, the rotarod was set to accelerate from 4–25 rpm for 300 s and the time spent by each mouse on the rotarod was recorded.

## Protein purification

*BL21*(DE3) chemically competent cells (TransGen Biotech, Cat# CD601) were transformed with pGEX-4T-2, pGEX-ZF-wt, pGEX-ZF-mut (R567W) and pGEX-ZF-dmut (R566C, R567W) plasmids and grown to an OD600 of 0.6–0.8 at 37 °C in LB medium containing 100 µg/mL ampicillin. The cells were induced by adding 1 mM IPTG and 200 µM $ZnSO_4$ at 28 °C for 6–8 h. The cells were collected by centrifugation at 4 °C and resuspended in appropriate lysis buffer (PBS with 1 mM DTT, 0.5 mM EDTA) supplemented with 1 × PIC before being sonicated. The cell debris was removed by centrifugation at $20,000 \times g$. The supernatant was filtered through a 0.45 µm membrane and combined with prewashed glutathione agarose resin (Yeasen Biotech, Cat# 20507ES10) via incubation at 4 °C for 2–4 h. The GST fusion proteins were then washed with PBS-T buffer (PBS with 1% v/v Triton X-100 and 1 × PIC) and eluted with glutathione elution buffer (50 mM Tris-HCl (pH 8.0), 10 mM reduced glutathione, 2 mM DTT, 150 mM NaCl and 1% v/v Triton X-100). The eluted proteins were quantified by the Bradford method and concentrated for subsequent use.

## Electrophoretic mobility shift assay (EMSA)

EMSA experiments were performed using a chemiluminescent EMSA Kit (Beyotime Biotechnology, Cat# GS009) according to the manufacturer's instructions. Briefly, probes for the CTCF motif were synthesized and annealed. The probes were incubated with appropriate amounts of purified GST fusion proteins for 10–20 min at RT and then electrophoresed on a 5% v/v native polyacrylamide gel in ice-cold 0.5 × TBE buffer (45 mM Tris-borate, 1 mM EDTA (pH 8.0)). The gels were scanned in a gel imaging system (Azure Biosystems). The bound/free ratios of probes were quantified with ImageJ software. The sequences of EMSA probes of CTCF motifs are listed in Supplementary Table 4.

## GST pull-down assay

GST pull-down experiments were performed by incubating appropriate amounts of biotin-modified probes, purified GST fusion protein and glutathione agarose resin for 6–12 h. Then, glutathione agarose was washed with PBS-T buffer (PBS with 1% v/v Triton X-100) supplemented with 1 × PIC. After centrifugation, the precipitated complexes were digested with proteinase K (Tiangen Biotech, Cat# RT403). The enriched probes were extracted with phenol:chloroform:isoamyl alcohol (25:24:1, Sigma-Aldrich, Cat# P3803) followed by ethanol precipitation and purification. The enriched DNA was spotted onto an Amersham Hybond-N+ membrane (GE Healthcare, Cat# RPN203B), UV cross-linked for 10 min and incubated with streptavidin-HRP (Beyotime Biotechnology, Cat# A0303). Finally, the membranes were washed and detected with an ECL chemiluminescence kit. The sequences of GST pull-down probes of CTCF motifs are listed in Supplementary Table 4.

## RNA-seq and data analysis

The RNA-seq experiments followed as previously described[58]. Total RNA was extracted from cells and tissues with an RaPure Total RNA Micro Kit according to the manufacturer's instructions. The libraries were constructed using the VAHTS Universal V8 RNA-seq Library Prep Kit for Illumina (Vazyme Biotech, Cat# NR605) according to the

manufacturer's instructions. Briefly, total RNA was purified by two rounds of mRNA purification steps with mRNA capture beads to ensure removal of rRNA. The mRNAs were fragmented to 250–450 bp at 85 °C for 6 min. Immediately afterward, cDNA first- and second-strand synthesis, end-repair and adapter ligation reactions were performed. cDNA purification and size selection were performed with AMPure XP magnetic beads (Beckman Coulter, Cat# A63882). The product was used for library amplification and purification followed by sequencing on an Illumina NovaSeq platform (Annoroad Gene Technology Co., Ltd.). For RNA-seq data analysis, paired-end reads were handled with Trim Galore (version 0.6.5) and then mapped to the mouse (mm10) or human (hg38) genome using STAR aligner (version 2.7.0)[59]. Transcript abundance was quantified by RSEM (version 1.2.22)[60]. To remove the effect of sex on gene expression in mouse samples, genes originating from ChrX and ChrY were filtered out. DEGs were determined using DESeq2 (version 1.32.0)[61] with $q$-value < 0.05 and fold-change > 1.5. Gene Ontology (GO) analysis was performed using clusterProfiler (version 4.0.0)[62]. The DEGs of mouse brain tissues and neurons are listed in Supplementary Data 1. The DEGs of human cortical organoids are listed in Supplementary Data 2.

## SnRNA-seq and data analysis for mouse brain cortex

Nuclei of E18.5 mouse brain cortex tissues were extracted using lysis buffer (10 mM Tris-HCl (pH 7.4), 10 mM NaCl, 3 mM $MgCl_2$, 0.1% v/v NP 40, 0.01% w/v digitonin, 1% w/v BSA, 0.1% v/v Tween-20, 1 mM DTT, 1 U/µL RNase inhibitor). The suspended nuclei were filtered using a cell strainer and counted using Countess II FL Automated Cell Counter (Thermo Fisher Scientific) for subsequent single-cell capture. SnRNA-seq library was performed by SequMed Biotech Inc. (GuangZhou, China) following the manufacturer's instructions using BD Rhapsody WTA Amplification kit (BD Biosciences, Cat #633801). Briefly, the cells and barcoded magnetic beads were added to the microwell plate chip of the BD Rhapsody system. Each bead was linked with a cell in a microwell. The next steps involved cell lysis within the microwells, release of mRNA with polyA tails, and its capture by the oligo-dT of the magnetic beads. The captured single-cell mRNA transcriptome was then collected, and reverse transcription was performed to obtain cDNA sequences. The cDNA underwent second-generation sequencing library construction and was subjected to quality control before being sequenced on the Illumina NovaSeq platform in PE 150 mode.

For snRNA-seq data analysis, the BD Rhapsody whole transcriptome analysis (WTA) pipeline was utilized to process FASTQ files, converting raw sequencing reads into gene expression matrix with unique molecular identifier (UMI) counts for each gene in each cell. The Seurat (version 4.0.5) was utilized for further analysis of the read count matrix for each gene/sample. Quality control measures were applied to filter individual cells, retaining only those expressing genes detected in at least 3 cells and cells expressing a minimum of 200 genes. Additionally, cells with gene expression counts ranging from 500–6000, mitochondrial gene expression not exceeding 20%, and a minimum RNA count of 2000 per cell were retained. The analysis yielded 13137 single cells in wild-type sample and 15661 in mutant sample that were used for subsequent clustering and differential expression analyses.

For cell annotation, the ScType package[37], which employs machine learning algorithms and reference datasets, was used. This package annotated the cell types based on the gene expression profiles of the single cells, leveraging marker genes or gene expression patterns to assign cell types. To identify genes showing differential expression between *CTCF* mutated sample and wild type sample, the Seurat package's FindMarkers function was employed. Differential expression analysis utilized the Wilcox method, which applies the Wilcoxon rank-sum test to compare gene expression levels between groups of cells. Benjamini-Hochberg method was used for multiple testing correction of $p$-values to ensure the rigor of the statistical

results. A corrected *p*-value < 0.05 with log2 fold change above 0.5 was considered statistically significant. All significant DEGs in each cell type were selected for GO enrichment analysis using clusterProfiler (v3.18.1). The DEGs are listed in Supplementary Data 3.

### SnRNA-seq and data analysis for mouse heart and lung tissues

SnRNA-seq of E18.5 heart and lung tissues were performed by Jiayin Biotechnology Ltd. (Shanghai, China) according to the instruction manual of the Chromium Single Cell 3′ Reagent Kit v3 (10x Genomics, Cat# PN-1000075) with the 10x Genomics platform.

For snRNA-seq data analysis, paired-end reads were aligned to the mouse reference genome (mm10) using Cell Ranger (version 6.1.1). Raw count matrices were used to remove ambient RNA with SoupX (version 1.6.1)[63] and then analyzed using Seurat (version 4.1.1)[64]. Poor-quality cells (> 50% mitochondrial genes, < 500 genes and < 500 UMI per cell for heart samples; > 5% mitochondrial genes, < 500 genes and < 1000 UMI per cell for lung samples) were filtered out. Potential doublets were removed with DoubletFinder (version 2.0.3)[65]. The data were then normalized and integrated using the SCTransform integration workflow by regressing out the cell cycle effect and percentages of mitochondria-expressed genes per cell. PCA was performed (npcs = 50), and dimensionality reduction (uniform manifold approximation and projection, UMAP) and clustering were then carried out (dimensions = 30; cluster resolution = 0.2 for heart samples, 0.4 for lung samples). Cell markers for each cell cluster were determined with the FindAllMarkers function. For heart samples, cell types were annotated based on a reference paper[66]. For lung samples, cell types were manually verified based on canonical markers. DEGs were identified via FindMarkers with a min. pct = 0.25 and an adjusted *p*-value < 0.05, and are listed in Supplementary Data 4 and 5. GO networks for DEGs were constructed using BinGo (version 3.0.5)[67] and then visualized in Cytoscape (version 3.9.1)[68].

### ChIP-seq

ChIP experiments were performed as described previously with some modifications[69,70]. Briefly, an appropriate amount of mouse tissue was crushed with a homogenizer, cross-linked with 1% w/v formaldehyde for 10 min and quenched with 0.125 M glycine. The crosslinked tissues were lysed in ChIP lysis buffer (1% w/v SDS, 10 mM EDTA, 50 mM Tris-HCl (pH 8.0) and 1 × PIC) and then sonicated to obtain chromatin fragments 200–400 bp in size. For normalization, an equal amount of HEK293T chromatin was added to the sample chromatin solution of different groups for subsequent ChIP experiments. Immunoprecipitation was carried out using 5 μg of antibodies for a duration of 12 h incubation at 4 °C, followed by a 2 h incubation with Dynabeads™ protein G (Thermo Fisher Scientific, Cat# 10003D). The immune complexes were subsequently washed with low salt buffer, high salt buffer, LiCl wash buffer and TE buffer. The enriched chromatin DNA was reverse-crosslinked and then purified using the MinElute PCR Purification Kit (Qiagen Cat# 28006) for ChIP-qPCR or ChIP-seq library construction. ChIP-seq libraries were constructed using the VAHTS Universal DNA Library Prep Kit for Illumina® V3 (Vazyme Biotech, Cat# ND607) according to the manufacturer's instructions. Briefly, 1–5 ng of enriched DNA was end-repaired and adapter-ligated. PCR library amplification was performed using VAHTS Multiplex Oligos Set 4 for Illumina (Vazyme Biotech, Cat# N321). DNA size selection and library purification were performed with AMPure XP Beads followed by sequencing on the Illumina NovaSeq platform (Annoroad Gene Technology Co., Ltd.). The primers of ChIP-qPCR used in this study are listed in Supplementary Table 5.

### ChIP-seq data analysis

Raw reads were trimmed to remove adapters using Trim Galore and then mapped to the mouse (mm10) and human (hg38) mixed genomes using Bowtie2 (version 2.2.5)[71] with the following parameters: --very-

sensitive --end-to-end --no-unal --no-mixed --no-discordant. Only uniquely mapped reads with MAPQ > 30 were retained. Reads mapped to the mouse or human genome were separated and then subjected to peak calling using macs2 (version 2.2.7.1)[72] with the default options. The CTCF peaks are listed in Supplementary Data 6.

To quantitatively compare the CTCF binding strength between wild-type and *CTCF*-mutated samples, CTCF ChIP signals were normalized according to the method in a previously published paper with minor changes[73]. Normalization was performed based on the assumption that the average CTCF binding strength from HEK293T cells mixed in different samples should be the same. The peak summit value of the averaged CTCF binding strength from HEK293T cells for each sample was extracted to calculate the scale factor. The scale factor for wild-type samples was considered to be 1. For corresponding *CTCF*-mutated samples, the scale factor was calculated by summit (wild-type)/summit (mutant). The scale factors for brain, heart and lung samples are shown in Supplementary Fig. 7a. Normalized bigwig files were generated with the bamCoverage tool in deepTools2 (version 3.5.1)[74] using RPGC normalization and scale factors.

### CTCF upstream motif analysis

The CTCF core motif was first identified for each CTCF site with FIMO (version 5.5.3)[75] using the parameter "--thresh 1e-4" with the CTCF position frequency matrix obtained from the HOCOMOCO database, and then the upstream 20 bp sequence for each CTCF site was extracted. The upstream sequences were clustered using hamming distance. Heatmaps were generated with ggmsa (version 1.3.4)[76], and sequence logos were generated with ggseqlogo (version 0.1)[77].

The CTCF motif in the promoters of *Pcdh* genes was analyzed as follows. As the CTCF binding strength in most *Pcdh* promoters was weak (Supplementary Fig. 8h), CTCF binding sites were hardly identified using the default parameters of macs2. Instead, CTCF binding regions in *Pcdh* promoters were manually selected based on the CTCF ChIP signal. Then, the CTCF core motif sequences and ± 20 bp flanking sequences were extracted, and sequence logos were generated with ggseqlogo.

### Bridge Linker-Hi-C (BL-Hi-C)

The BL-Hi-C experiments were conducted following previously established procedures[78]. In summary, suitable quantities of mouse tissues were crushed using a homogenizer, cross-linked with 1% w/v formaldehyde for 10 min, and subsequently neutralized with 0.2 M glycine. The nuclei were extracted with SDS lysis buffer (1% w/v SDS, 50 mM HEPES-KOH (pH 7.5), 150 mM NaCl, 1 mM EDTA, 1% v/v Triton X-100, 0.1% w/v Sodium Deoxycholate) and then utilized for subsequent procedures: digestion with HaeIII (NEB, Cat# R0108L) at 37 °C for 12 h, end-plus-A treatment with 10 mM dATP (NEB, Cat# N0440S) and Klenow Fragment (exo -) (NEB, Cat# M0212L) at 37 °C for 30 min, along with proximity ligation with biotin-labeled BL-Linker (oligo sequences were listed in Supplementary Table 2.) at 16 °C for a duration of 4 h. The unligated linker was eliminated using Lambda Exonuclease (NEB, Cat# M0262L) and Exonuclease I (NEB, Cat# M0293L) at 37 °C for 1 h. The DNA was then reverse-crosslinked and purified using a phenol:chloroform:isoamyl alcohol (25:24:1) extraction. Subsequently, the DNA was fragmented into an average size of 300 bp through sonication, and the biotin-labeled DNA was enriched using Dynabeads™ M-280 streptavidin (Thermo Fisher Scientific, Cat# 11205D). The enriched bead-bound DNA underwent end repair, adapter ligation, PCR amplification, and DNA library construction, followed by sequencing on the Illumina NovaSeq platform (Annoroad Gene Technology Co., Ltd.).

### BL-Hi-C data analysis

The adapters were trimmed from raw paired-end reads with Trim Galore, and the linker sequences were removed with the trimlinker

tool from the ChIA-PET2 package (version 0.9.3)[79]. The trimmed sequences were then mapped to the mouse (mm10) genome and processed using HiC-Pro (version 2.11.1)[80]. The reproducibility of the Hi-C replicates was assessed with HiCRep (version 0.2.6)[81]. The valid pair files generated by HiC-Pro were transformed into hic files with KR normalization using juicer tools (version 1.122.01)[82].

A/B compartments were analyzed by juicer tools with 100 kb resolution. Insulation scores (ISs) were calculated with 50 kb resolution using FAN-C (version 0.9.20)[83]. ISs were calculated with a window size of 400 kb for brain samples and a window size of 300 kb for heart and lung samples. TAD boundaries were determined by the "fanc boundaries" function with the parameter -s 0.7. The genomic regions that had no mapped reads and the lowest 5% covered reads were combined as low-coverage regions, which were excluded from the insulation score and TAD boundary analysis. The region between two nearby TAD boundaries was defined as the TAD. The statistics from HiC-Pro processes and TAD information are summarized in Supplementary Data 7.

### Quantitative high-resolution chromosome conformation capture copy (QHR-4C)

QHR-4C experiments were performed as previously described[46]. Briefly, digested suspensions of $1 \times 10^5 – 1 \times 10^6$ cells from tissues were cross-linked with 2% w/v formaldehyde for 10 min, and crosslinking was terminated with 0.2 M glycine. The cell pellet was permeabilized by lysis buffer (50 mM Tris-HCl (pH 7.5), 150 mM NaCl, 5 mM EDTA, 0.5% v/v NP-40, 1% v/v Triton X-100), digested with DpnII overnight (NEB, Cat# R0543S) and subjected to proximity ligation by T4 DNA Ligase (NEB, Cat# M0202S). Then, chromatin was reverse-crosslinked and DNA was purified by using MinElute PCR Purification Kit, and was subjected to sonication, resulting in fragmentation into pieces <1000 bp. To enrich ligation events associated with a specific viewpoint, an appropriate amount of sonicated DNA was taken as a template for linear amplification. This process was conducted for 100 cycles using a 5' biotin-labeled probe of the viewpoint of interest by using Phanta Max Super-Fidelity DNA Polymerase (Vazyme Biotech, Cat# P505). The amplified products were incubated at 95 °C for 5 min, immediately cooled on ice to obtain ssDNA and then enriched with Dynabeads™ M-280 streptavidin. The bead-bound ssDNA was then ligated with adapters. Finally, QHR-4C libraries were constructed with specific primer pairs (forward primers containing Illumina P5 with sequences near a specific viewpoint and reverse primers containing Illumina P7 with an index and sequences matching the adapter) within the system of Phanta Max Super-Fidelity DNA Polymerase and then sequenced on the Illumina NovaSeq platform (Annoroad Gene Technology Co., Ltd.). The primers for QHR-4C used in this study are listed in Supplementary Table 2.

### QHR-4C data analysis

Adapter sequences in raw paired-end reads were removed with Trim Galore. The primer sequence at the 5' end of read 1 was then trimmed with cutadapt (version 3.4)[84]. Reads that did not contain primer sequences were discarded. Reads were mapped to the mm10 genome using Bowtie2 with the following parameters: --very-sensitive --end-to-end --no-unal -X 2000. Bam files were imported into the r3Cseq package (version 1.38.0)[85]. Normalized bedgraph files were thus generated and then transformed into bigwig files using the bedGraphToBigWig tool.

### Generation of cortical organoids

Cortical organoids differentiated by different genotypes of hESCs were generated as previously described[47,48]. hESC clones of different genotypes were digested into single cells for subsequent organoid differentiation. Briefly, 9000 cells per well were reaggregated in ultra-low-cell-adhesion 96-well plates with V-bottomed conical wells (Sbio Japan, Cat# MS9096VZ) in cortical differentiation medium (CDM) I. This

medium contained Glasgow-MEM (Gibco, Cat# 11710035), 20% v/v Knockout Serum Replacement (KSR) (Gibco, Cat# 10828028), 0.1 mM Minimum Essential Medium non-essential amino acids (MEM-NEAA) (Gibco, Cat# 11140050), 1 mM sodium pyruvate (Gibco, Cat# 11360070), 0.1 mM 2-mercaptoethanol (Life Technologies, Cat# 21985023), and 1% v/v penicillin/streptomycin. From day 0 to day 6, ROCK inhibitor Y-27632 was added to the medium at a final concentration of 20 µM. From day 0–day 18, WNT inhibitor IWR-1 (STEMCELL Technologies, Cat# 72564) and TGF-β inhibitor SB431542 (STEMCELL Technologies, Cat# 72232) were added at concentrations of 3 µM and 5 µM, respectively. From day 18, the floating aggregates were cultured in 6-well ultra-low-attachment culture dishes (Corning, Cat# 3741) on an orbital shaker at 70 rpm in CDM II medium, containing DMEM/F12, 2 mM Glutamax, 1% v/v N2 Supplement (Gibco, Cat# A1370701), 1% v/v Chemically Defined Lipid Concentrate (Gibco, Cat# 11905031), 0.25 µg/mL Amphotericin B (Gibco, Cat#15290026), and 1% v/v penicillin/streptomycin. On day 35, cell aggregates were transferred to the CDM III medium, consisting of CDM II supplemented with 10% v/v FBS, 5 µg/mL heparin (STEMCELL Technologies, Cat# 07980) and 1% v/v Matrigel (Corning, Cat# 356234). From day 70, organoids were cultured on an orbital shaker at 70 rpm in CDM IV medium, consisting of CDM III supplemented with B27 supplement and 2% v/v Matrigel. The cultures were incubated at 37 °C and 5% CO₂. And the medium was replaced every 3–5 days. The cortical organoids at different time points of differentiation, were imaged and collected for size quantification, frozen section preparation and RNA-seq experiments, respectively. The organoids were fixed with 4% w/v paraformaldehyde, immersed in 30% w/v sucrose solution, encapsulated in OCT Tissue Freezing Medium, and snap-frozen at -80 °C. Ten-micrometer sections were made using a cryostat microtome (Leica, Germany) at -20 °C for subsequent immunofluorescence staining. Images and area values were obtained with an Olympus inverted microscope. The measured data are provided in the Source Data table. Bulk RNA-seq experiments and data analysis have been described in the previous methods section.

### Organoid dissociation and scRNA-seq

A combination of 3–4 similarly sized cortical organoid spheres of the same genotype were subjected to digestion with papain at a concentration of 1 mg/mL for 10 min at 37 °C on a shaker. The digests were gently pipetted and subjected to another 10 min digestion. The resulting cell suspension was filtered through a 40 µm filter and resuspended in a solution of 1% w/v BSA in PBS for cell counting. The scRNA-seq library for cortical organoids was constructed by SequMed Biotech Inc. (GuangZhou, China). Briefly, single-cell suspensions were processed using the DNBelab C Series Single-Cell Library Prep set (MGI, 940-000047-00). Droplet encapsulation, emulsion breakage, mRNA capture, reverse transcription, cDNA amplification, and purification was used to convert the suspensions into barcoded scRNA-seq libraries. After confirmation of the quality, the libraries were sequenced on the DNBSEQ-T7RS platform.

### Computational analysis of organoid scRNA-seq data

Cell Ranger (version 6.1.1) with the human reference genome (hg38) was used to generate the output count matrix. Cells with > 20% mitochondrial content, fewer than 300 features, and features expressed in fewer than three cells were excluded from the analysis. We used Scrublet (version 0.2.3)[86] and the gene expression patterns (total counts > 40,000) to determine the doublets. The doublet score was calculated independently for each sample with the default settings and with the expected doublet score set at a rate at 10%.

Four organoid scRNA-seq datasets were generated for this study. The cortex region data were extracted from the Allen Brain's human fetal developmental brain scRNA data, which were used as reference data for cell type annotation. SCANPY (version 1.9.1)[87] and Seurat (version 4.3.0)[64] were used for major data analysis and visualization.

The following steps were performed in order: data normalization, log-transformation, highly variable gene selection, and PCA. The expression levels were calculated as counts per 10,000 (NC); briefly, the total mapped read counts of a gene within one cell were scaled by the number of total mapped reads of that cell and times multiplied by 10,000. The $\log_2(NC + 1)$-transformed values were used for further analysis unless indicated otherwise. The highly variable genes were selected based on the raw count data with flavor as "Seurat_v3". The top 1000 were selected. PCA was run with the selected highly variable genes.

Data integration was performed on the top highly variable genes via Harmony (version 0.0.6)[88] with the default parameters. UMAP was performed on the Harmony-corrected PCA embedding. The neighborhood graphs were calculated using scanpy.pp.neighbors, with 15 local neighborhoods and 50 Harmony-corrected PCAs using the mKNN graph. The connectivities were computed using the UMAP method with Euclidean distance. Then, UMAP embedding was performed using scanpy.tl.umap with a minimal effective distance of 0.5, a spread of 1.0, an initial learning rate of 1.0, a negative sample weighting of 1.0 and a negative edge sample rate of 5. Louvain clustering was performed using sc.tl.louvain on the neighborhood calculated in the previous steps, with a resolution of 1, with the vtraag package and with no weights from the mKNN graph. For each cluster, a logistic regression model was trained on the reference dataset and used to annotate organoid cell types. After cell type annotation, the proportions of various cells in each sample were calculated. Milopy (version 0.1.1) was used to calculate and visualize the changes in cell proportions between wild-type and *CTCF*-mutated cells in 1-month samples.

DEGs for each cell type population between wild-type and *CTCF*-mutated samples (see Supplementary Data 8) were selected by Seurat via FindMarkers with min. pct = 0.25 and an adjusted *p*-value < 0.05. GO enrichment analysis was performed on the sets of differentially expressed genes for each cell type using the enrichGO function in the clusterProfiler (version 3.18.1) package.

### Statistical analysis

All statistical analyses were performed using GraphPad Prism software. Statistical parameters are reported as sample quantity (n represents the number of animals, cells or organoids per group), and the statistical details of experiments can be found in the figure legends. All data and graphs are presented as mean ± SD, unless stated otherwise. The n-values of statistical samples and the exact *p*-values are indicated in the figures. The statistical data are provided in the Source Data file.

### Reporting summary

Further information on research design is available in the Nature Portfolio Reporting Summary linked to this article.

## Data availability

The ChIP-seq, RNA-seq, BL-Hi-C, QHR-4C, snRNA-seq and scRNA-seq data reported in this paper have been deposited in the Gene Expression Omnibus (GEO) database under accession code GSE214692 and in the Genome Sequence Archive database[89] in the National Genomics Data Center[90] (GSA) under accession code CRA008223 and HRA003128. The dataset of this paper has been submitted to the figshare repository (https://doi.org/10.6084/m9.figshare.26010637). The referenced Allen fetal brain data is available from the European Genome Phenome Archive under accession code EGAS00001004107 (https://ega-archive.org/datasets/EGAD00001006049). The GWAS table source of autism spectrum disorder used in this study is available in the GWAS Catalog under accession code EFO_0003756 (https://www.ebi.ac.uk/gwas/efotraits/EFO_0003756). Source data are provided with this paper.

## Code availability

All code used in this study was previously published and no customized code was used in this paper.

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

## Acknowledgements

We thank Professor Qiang Wu for sharing the protocol of the QHR-4C experiment. The authors also thank the Guangzhou Branch of the Supercomputing Center of the Chinese Academy of Sciences for support. This work was supported by the National Natural Science Foundation of China (31925009), National Key R&D Program of China (2021YFA1100300), the National Natural Science Foundation of China (U21A20195, 32300477), Independent Fund of the State Key Laboratory of Respiratory Diseases (SKLRD-Z-202102), Major Project of Guangzhou National Laboratory (GZNL2023A02010, GZNL2023A02008), and Guangzhou Key R&D Program (2023B03J1230).

## Author contributions

H.Y. initiated the study and designed the experiments. J.Z., Y.L., and H.R. conducted most of the experiments. G.H. performed the bioinformatic analysis for RNA-seq, ChIP-seq and Hi-C experiments. Y.H. and Z.M. helped with scRNA-seq analysis. W.Z. helped with Golgi and Ca$^{2+}$ imaging. D.W. and Y.Q. helped with histology. N.M. helped with MEA measurements. Y.W., B.J., Haidong.W., H.L., L.Z., Hua W., and G.P. contributed to the work. H.Y., J.Z., G.H., Y.L., and H.R. wrote the manuscript. H.Y. conceived and supervised the entire study.

## Competing interests

The authors declare no competing interests.

## Additional information

[1]State Key Laboratory of Respiratory Disease, Guangzhou Institutes of Biomedicine and Health, Chinese Academy of Sciences, Guangzhou, China. [2]The First Affiliated Hospital of Guangzhou Medical University, Guangzhou, China. [3]Department of Basic Research, Guangzhou National Laboratory, Guangzhou, China. [4]University of Chinese Academy of Sciences, Beijing, China. [5]College of Veterinary Medicine, Shanxi Agricultural University, Jinzhong, China. [6]Division of Life Sciences and Medicine, University of Science and Technology of China, Hefei, China. [7]Key Laboratory of CNS Regeneration (Ministry of Education), Guangdong-Hong Kong-Macau Institute of CNS Regeneration, Jinan University, Guangzhou, China. [8]Institute of Clinical Pharmacology, Key Laboratory of Anti-Inflammatory and Immune Medicine (Ministry of Education), Anhui Medical University, Hefei, China. [9]These authors contributed equally: Jie Zhang, Gongcheng Hu, Yuli Lu, and Huawei Ren. ✉e-mail: yao_hongjie@gzlab.ac.cn

