## [Peer Review File · Nature Communications]

CTCF mutation at R567 causes developmental disorders via 3D genome rearrangement and abnormal neurodevelopmentReviewers' Comments:

Reviewer #1:

Remarks to the Author:

This is a very solid and well done study modeling the phenotype of a single CTCF mutation which has been described in humans and causes a severe phenotype. The authors model this in mice and characterize the different developmental phenotypes occurring as well as the molecular changes accompanying this mutation, and how the 3D landscape is affected and how that changes gene expression. The authors extend their findings to a human iPSC model and show robust phenotypes in neuronal organoids and phenotypes obtained.

The results shown are very convincing and solid, and the conclusions are supported by the results and methods used to obtain these results. The paper is clear and the figures are informative and easy to comprehend.

I support publishing this work and I have a minor comment:

1) Seems IPSC results are based on a single IPSC clone. I recommend validating this in an independent clone or independent line to exclude off-target effects of CRISPR.

Reviewer #2:

Remarks to the Author:

Zhang et al. established the first mouse model to study CTCFR567W, a mutant has been reported in several intellectual impairment patients. Their study focuses on characterizing the phenotypes associated with this mutation and aims to uncover the molecular mechanisms that may explain these phenotypes. The work is intriguing and presents a rich dataset that holds great value for the scientific community. However, I found it's overwhelming to read the manuscript. The logical flow between paragraphs is lacking, which makes it challenging to follow the study's progression. Additionally, there are several flaws in the experimental design, analysis methods, and interpretation of results, which undermine the overall quality of the research. It is crucial for the authors to address these issues and make significant efforts to rectify them before the manuscript can be well received by its intended audience.

Major:

1. Fig 1f, 1g, 1h. Figure legend mentioned "(n=3)" so I suppose there are values and t-test results somewhere, but I could not find them? While Fig 1g is convincing but I find P0 results in Fig 1h might be misleading, because it's not clear whether the strong difference in CTCFR567W/R567W is simply a result of mouse death and stopped breathing. I would suggest moving to Sup or making it clear in the text. In "thick interstitia and denser alveoli, followed by alveolar inflation failure, resulting in respiratory pulmonary distress", I don't think these evidences could support the "resulting" here.
2. Page 6: "cardiorespiratory deficits, mimicking clinical case-related phenotypes to some extent". I tried going through references 12-16 but had trouble concluding patients with R567W have heart problems. It would be nice if they could list all the reported phenotypes for R567W. Given Konrad et al. also suggested the "Clinical spectrum is highly variable", it's worth dissecting the details.
3. Fig 2: why 2a, 2e-k doesn't have CTCF+/R567W results?
4. Fig 3d/Extended Fig 4c: which section of brain are these data? How do you know the difference was not because of their different location of brain? The DEG of which cluster were most overlapping with DEGs from Fig. 3a/3b? Does Pcdh genes appear in any of these DEGs, if yes, which cluster?
5. Extended Fig 4g: dCM1 cell number reduced the most. If you detected DEGs compare dCM1 vs dCM2/3/4, do the top genes also differ between wt and mutant? Does any of these genes have an implication in the phenotype reported in Fig 1g?
6. Fig 3k: the DEGs detected are from different technologies thus I am not sure this is meaningful

comparison. Also, not surprising if they are indeed cell type specific. I would be more attracted to dissecting some details. For example, Extended Fig4j shown Lung cell does not change much that are consistent with Fig 1h. While Extended Fig4g shown dCM1 reduced almost 2-fold that might explain Fig 1g. Make me believed CTCF mutant have strong effect in heart than lung. It might be worth to extend analysis to either support or disapprove these observations.

7. Fig 4/Extend Fig5d,5e: all the motif analysis involved (U) motif were not legitimate. The (U) motif could have different space to CTCF core motif thus conventional motif analysis would not work. You need employ sequence clustering method like Nakahashi et.al (Fig4, doi: 10.1016/j.celrep.2013.04.024) and Hyle et.al (Fig 6, doi: <https://doi.org/10.1186/s13059-022-02843-3>). This would provide better visualization and more insights. Another interesting analysis could be conduct is scanning the (U) motif and examining whether there is a significant difference in the proportion of this motif between the groups depicted in Fig4c. Also, what's the dash line in middle of Fig4c?

8. Fig 4a and Page 10: It's confusing that the mutant is "reduced CTCF binding genome-wide" and "by weakening its ability to bind the U motif". These together does the authors suggested the U motif is genome-wide? Which is not true, for example, Nakahashi et.al detected about 18000 lost peaks in dZF11 and 10000 significant decreased peaks(out of 43000) if we re-analysis their mice data. Half the peaks decreased 2 fold have (U) motif(~5000), 60% peaks decreased 4 fold have (U) motif(~2000). I could reason why single mutant at ZF11 could have stronger effect than dZF11 from Nakahashi. Besides, I could not find replicates for the ChIP-seq they provided at GEO and there is no reviewers' token for GSA that I could check any processed data.

9. Fig 4g: does the TAD boundary affected have disrupted CTCF binding? Does the 4 groups have any bias on proportion of overlapping disrupted binding? Does the Disrupted CTCF binding at boundaries have (U) motif?

10. Fig 5c: I downloaded their .hic data on GEO but I could not reproduce the top panel regardless which normalization method used(see below). I could not convince myself there is a difference at this locus, and their values were huge compared to the value in the file. Please elaborate.

11. Page 12: They previously mentioned detecting "genome-wide" decreased CTCF sites with (U) motif. It's against their claim here "selectively reduced CTCF binding in the promoters of Pcdh β genes". It would be interesting to check how many of the gene promoters have "decreased CTCF sites with (U) motif"? How many of these gene have decreased expression? Does log2fc from Pcdh β significantly lower than the log2fc of these genes? And careful revise the conclusion here accordingly.

12. Please provide side by side expression(KO vs WT) for PCDH genes in Extend Fig7f for six clusters of Extend Fig7k. Seurat Dotplot would be good choice. Would they only see difference only in "Radial glia" and "GABAergic" like their Fig 6f suggested? If not, why?

Minor:

Page 4: "Based on the number of different genotypes at E18.5, most Ctcfr567W/R567W mice could fully develop, but they were lighter in weight and smaller than wild-type mice (Fig. 1c)." What does "Based on the number of different genotypes at E18.5" meant?

Page 4: "exhibited no coloration or respiratory distress and perished within 30 min" confusion. "no respiratory distress" or "have respiratory distress and perished"?

Page 4: "clinical phenotypes of short stature and developmental delay" first mentioned, reference? Fig 2h: why the black line out of blue box is missing? Looks suspicious, please double check.

Extend Fig5n: why the bottom panel have empty sequences?

Fig 6g/Extend Fig7k: why there are a lot missing values?

Reviewer #3:

Remarks to the Author:

The manuscript by Zhang et al establishes a mouse gene mutation knock-in model and human cortical organoid model from gene-edited human embryonic stem cells to investigate the impact of a pathogenic CTCF mutations that has been implicated in a human neurodevelopmental disorder. The breadth of innovative techniques probing chromatin architecture and gene expression that are impacted by disrupted CTCF binding often at a locus- or cell/tissue-specific manner is very insightful. The R567W knock-in mouse model is very informative and is the first model examining CTCF ZF mutation in the context of whole organismal development.

A number of issues have come up in my review, which I believe are problematic and need to be addressed by the authors. One major factor is the veracity of the human embryonic stem cell model, as how these cells were isolated and having appropriate controls could have a major bearing on the results. This matter clouds the clear phenotypic differences observed between heterozygous CTCF^{+/R567W} mice and the heterozygous organoids.

Major Comments

1. Title: It needs to mention that the chromatin rearrangement and abnormal neurodevelopment only occurs with the homozygous mutation. Abstract: Whenever CTCF^{R567W} is mentioned it needs to be qualified whether it is homozygous or heterozygous. It should also be stated that this mutation is within the DNA binding ZF region. 'Developmental disorders' (mentioned twice) – strictly speaking there is only one linked to CTCF, MRD21. Mention OMIM descriptor at appropriate place on Page 3.
2. Page 3: it should be clearly stated that human mutations in clinical case reports only occur as heterozygous mutations.
3. Ext Data Figure 1b: For each developmental time point in this table a Chi-square analysis should be performed and presented in a new column. By my reckoning at E18.5 there is a sub-Mendelian ratio of R567W homozygous pups
4. Page 2: CTCF cellular level changes and cell death & Page 3: lean phenotype during juvenile stage in mice have been reported before. Adding PMID 30513694 would be appropriate here as supportive evidence (also at Page 5 Fig 1e).
5. Page 4: it should be stated that the R567W mutation occurs in ZF 11 of the 11-ZF DNA binding domain of CTCF, and also occurs adjacent to the exon 9 5' splice site (this is important, see later questions). Marked onto Figure 1 should be those CTCF ZF residues depicted as DNA binding (from PMID 29076501). Also Zn²⁺ co-ordinating residues should be marked with a shaded band or in different colour.
6. Further to Q4: the CGG to TGG mutation occurs in the last codon before the 5'SS. This could affect normal splicing of Ctf in mice by modifying the splice site or ESEs. The authors have only used an antibody (#07-729) that was raised with an immunogenic peptide representing in the C-terminus (aa 659-675). Therefore the authors must use a CTCF antibody recognising the N-terminus that will detect any potentially truncated proteins.
7. Clarification: Didn't CTCF homo mice exhibit respiratory distress (Page 4?)
8. Figure 1c and for any data in this manuscript that reports statistical significance – the numerical data should be reported in the text of the Results, ie mean +/- SD and give p-value.
9. Page 5: 'half dose of CTCF at R567' is a misleading or inaccurate statement. Please revise.
10. It is not explicitly mentioned, but were all mouse experiments performed with littermates? i.e. WT, het and KO mice from the same litters. This is an important distinction and is needed to support the rigour of any findings.
11. Figure 4: as CTCF binding due to R567W mutation is reduced, the U motif is being enriched. This suggests that those sites that are lost do not contain the U motif?? Unless, I am not understanding the Figure correctly (or the authors are not explaining it well). Does Figure 4b depict the sequence logo of those CTCF sites remaining or those that are lost due to R567W?
12. What percentage of all CTCF sites contain the U-motif? What is the nature and distribution of those sites lost with R567W? Are they inter- or intragenic? Intronic? Proximal to promoter/enhancers? Can they be matched with any public Hi-C data to determine if they are convergent or divergent sites?
13. As the crystal structures for overlapping portions of the CTCF ZF domain have been solved eg PDB 5YEL (ZFs 6-11). It would be useful to model the R567W mutation to visualise the loss of DNA binding

and informative to examine the impact on DNA binding to the U motif. A similar approach has been used in PMID 34657170 & 37047368

14. Page 10: Reference 12 is not an appropriate reference here, remove.

15. GST pulldowns: it is not clear why R566C was being examined here in the context of a R567W? Was it to get a more effective loss of DNA binding ZF? Why is only the gel shift using the U-motif M2 oligo shown (& demonstrating weak binding), when the M1 oligo (mentioned in Supp Table 2) is clearly of a core consensus CTCF site, but no data is shown? The authors must show that the 11-ZF recombinant CTCF protein binds a core consensus site. I suspect that with a longer oligo (C + U motif), there will be no loss of binding, but this still needs to be tested, as this speaks to the fact that only some CTCF binding is lost due to R567W mutation, but these sites are critical for tissue-specific development. Also, a (C + mutant U motif) oligo should be used as well). Also it is not clear how the M2 oligo sequence here relates to the U motif? It was hard to do complementary alignments based on the very small text provided for the U motif sequence logo in the Figures.

16. Figure 4g: four TAD types are defined here, but 5 TAD descriptors are then examined in Figure 5. What happened to 'boundary' in the 4g analysis?

17. I suspect loss of potential RNA interactions due to R567W mutation are playing a bigger role in both models than what has been stated (page 17). This is a bit of a shortcoming as no declarative statement about RNA binding can really be made, other than speculation.

18. Figure 6: how the R567W-mutant H1 human embryonic cell lines were established is problematic and a major flaw of this paper. The authors only describe one het clone and one KO clone (from the data I can see), but then base a whole range of scRNAseq and bulk RNAseq analyses on these 2 clones. Furthermore, no 'scrambled' or non-targeting gRNA control was used for wildtype cells. Effectively the authors are comparing cells subjected to the transfection, single-cell selection process and expansion process to cells (WT) that have not gone through this process. As it stands there are quite considerable differences between WT, het and KO (Ext Fig 7a). Without this control, the human cortical organoid study is invalid. Furthermore, only one clone was selected each for het and KO. This does not control for clonal variation which can emerge during the gene editing and screening process. What is more, the het clone that is used (Ext data 6b) has one mutant allele (W567) but has a 6 bp deletion in the 5'SS of the second 'normal' allele. This 6 bp deletion actually weakens the splice donor site (maximum entropy score: 10.17 to 4.20 (MaxEntScore)). This could result in alternative splicing at this allele, resulting in a truncated CTCF protein, which cannot be detected with the C-terminal specific antibody used here. The authors need to validate that these cells express full-length (130 kDa) CTCF using an antibody recognising the N-terminus. Also they should perform RT-PCR across exon 9 to 10 to show that no alternative splicing is occurring (and truncated CTCF eventuating). Furthermore, additional clones of WT, het and KO need to be examined. The variation introduced by the method that clones were isolated may be sufficient enough to explain the discrepancy between human and mouse heterozygous results. Also, Please update methods with length of time it took to isolate clones.

19. Figure 6a, b: is this depicting survival rates beyond two months (Page 13)? As only data up to day 31 are shown.

20. Page 17: 'The DEGs caused by CTCF^{R567W} might be affected by direct or indirect mutation cascades, or these DEGS might aggregate...' – cascades and aggregate are inaccurate terms here. Page 17: imitate=phenocopy.

21. Page 17: Its worth adding a comment in here after discussion of R567 mutation: 'As with other CTCF ZF mutations, locus-specific effects on DNA binding may result, leading to a spectrum of loss or even gain-of-function phenotypes (PMID 34657170). Note also for Discussion, genome-wide analysis of CTCF binding using a mutant CTCF (in the 11th ZF) still resulted in over 60% sites being maintained (PMID 23707059).

22. Figure 1e-k, why was CTCF^{+/R567W} mouse data not acquired here?

23. Summary Figure (Fig 5f) for the Pcdh locus: cohesin rings are depicted here in the chromatin architecture of the Pcdh locus, but are not examined in the manuscript. Whilst CTCF R567W mutation may not impact cohesin binding genome wide, RAD21 ChIP datasets are not examined. This is an important aspect of chromatin looping dynamics. Please comment.

Minor Comments

1. Abstract: damnification is not useful scientific term here.
2. Page 3: code shifts is not an appropriate scientific term, use 'frameshift'.
3. Page 5: 'thicary interstitia'?
4. Page 5: Revise: 'homozygous CTCF^{R567W} mutation causes lethality at birth, developmental delay and cardiorespiratory deficits, overlapping clinical case-related phenotypes to some extent.'
5. CM is not defined.
6. Page 10: 'To determine whether CTCF^{R567W} disrupts chromatin organization'
7. Page 21: 'RNApure'?
8. Figure 1c: Figure key not needed if you use x-axis labels. Please check you are using the right stats here? A one way ANOVA and multiple testing between all 3 genotypes may be more appropriate.
- Figure 1e: make gender symbols more obvious (underlined and labelled under both genotypes)
9. Ext data 3j & k: what do the units pfu represent?
10. The sequence logos in Figure 4 and Ext data Fig 5 are too small too be useful.
11. Ext Data Fig 6: 'hete' is not standard usage
12. Western blot size markers should be kDa not KD

Point-by-point Responses to Reviewers

Reviewer #1 (Remarks to the Author):

This is a very solid and well done study modeling the phenotype of a single CTCF mutation which has been described in humans and causes a severe phenotype. The authors model this in mice and characterize the different developmental phenotypes occurring as well as the molecular changes accompanying this mutation, and how the 3D landscape is affected and how that changes gene expression. The authors extend their findings to a human iPSC model and show robust phenotypes in neuronal organoids and phenotypes obtained.

The results shown are very convincing and solid, and the conclusions are supported by the results and methods used to obtain these results. The paper is clear and the figures are informative and easy to comprehend.

I support publishing this work and I have a minor comment:

1) It seems the results are based on a single iPSC clone. I recommend validating this in an independent clone or independent line to exclude off-target effects of CRISPR.

Response: We apologize for not describing this clearly. Two independent hESC clones were used for the initial phenotypic studies in the previous manuscript. Due to the difficulties of editing the *CTCF* gene in hESCs, nearly 2 years and multiple editing strategies were needed to obtain 2 homozygous and several heterozygous edited clones. We selected two clones of each genotype for the organoid experiments and bulk RNA-seq. We have updated the figures and methods with more details on the editing process and the selected clones (**Fig. 6** and **Supplementary Fig. 9** in the revised version).

In our hESC-derived cortical organoids, we interestingly found that compared to wild-type organoids, CTCF R567W heterozygous organoids exhibited early exhaustion of stem-like cells and an increase in GABAergic neurons. This observation is consistent

with our new snRNA-seq data from homozygous mutant mice (**Fig. 3d-g** in the revised version), suggesting that the CTCF R567W mutation might disrupt the differentiation of stem-like cells into specific neurons in both human and mouse systems. This finding implies that CTCF mutations, similar to autism risk genes, may induce neurodevelopmental disorders through similar cellular development pathways.

Reviewer #2 (Remarks to the Author):

Zhang et.al established the first mouse model to study CTCFR567W, a mutant has been reported in several intellectual impairment patients. Their study focuses on characterizing the phenotypes associated with this mutation and aims to uncover the molecular mechanisms that may explain these phenotypes. The work is intriguing and presents a rich dataset that holds great value for the scientific community. However, I found it's overwhelming to read the manuscript. The logical flow between paragraphs is lacking, which makes it challenging to follow the study's progression. Additionally, there are several flaws in the experimental design, analysis methods, and interpretation of results, which undermine the overall quality of the research. It is crucial for the authors to address these issues and make significant efforts to rectify them before the manuscript can be well received by its intended audience.

Major:

1. Fig 1f, 1g,1h. Figure legend mentioned "(n=3)" so I suppose there are values and t-test results somewhere, but I could not find them? While Fig 1g is convincing but I find P0 results in Fig 1h might be misleading, because It's not clear whether the strong difference in CTCFR567W/R567W is simple a result of mouse dead and stopped breathing. I would suggest move to Sup or make it clear in the text. In "thicary interstitia and denser alveoli, followed by alveolar inflation failure, resulting in respiratory pulmonary distress", I don't think these evidences could support the "resulting" here.

Response: We appreciate Reviewer #2's careful review. We apologize for the confusion in Fig. 1f, 1g and 1h. The "n=3" in the figure legend merely indicates that histological sections from 3 mice were examined, and the results were consistent with the representative images shown.

We agree with the reviewer that the phenotypes shown in Fig. 1h (**Supplementary Fig. 1h** in the revised version) for P0 *Ctcf*^{R567W/R567W} mice could be misleading. The observed changes in lung morphology could be secondary effects due to perinatal death

rather than direct consequences of the mutation itself. To avoid overstating the results, we have moved Fig. 1h to **Supplementary Fig. 1h**. In addition, we agree that the data does not strongly support the statement of thicary interstitia and alveolar inflation failure "resulting in" respiratory distress. We have clarified these points in the revised manuscript and modified the original text as follows: "*Additionally, we observed a thickened alveolar interstitium and dense, noninflated alveoli in the lungs of E18.5 and postnatal day 0 (P0) Ctcf^{R567W/R567W} mice. While these pathological changes could potentially contribute to respiratory distress, they might also represent secondary effects of the animals' demise rather than direct consequences of homozygous Ctcf mutation*" (Page 6).

2. Page 6: "cardiorespiratory deficits, mimicking clinical case-related phenotypes to some extent". I tried went through references 12-16 but hard to conclude patients with R567W have heart problems. It would be nice if they could list all the reported phenotypes for R567W. Given Konrad et.al also suggested the "Clinical spectrum is highly variable", it worth dissect the details.

Response: We agree with Reviewer #2 that there are no definitive reports of cardiac phenotypes in patients with the specific CTCF R567W mutation. Our intention was to point out that the cardiac defects observed in our CTCF mutant mouse model resemble congenital heart defects that have been reported in a subset of patients with CTCF mutations in general, not specifically in those with R567W mutations.

To provide more details on the clinical phenotypes of the R567W mutation, we looked at the ClinVar records (<https://www.ncbi.nlm.nih.gov/clinvar/variation/88638/>) that have documented several reported cases of the R567W mutation. Of these, two cases have been described in more detail in the literature. One article reported three *CTCF* mutation cases, including one patient with R567W (patient 3), who was described as having severe intellectual disability with autistic features, microcephaly, and severe feeding difficulties requiring tube feeding at 4 years of age. The shared features among the three patients were intellectual disability, microcephaly, feeding difficulties, and a

lack of major malformations, except for a single patient who exhibited congenital heart defects and cleft palate (PMID: 23746550). Another article described three patients, including patient 2 with R567W, who exhibited hypotonia, delayed motor milestones, short stature, and delayed bone age (PMID: 30893510). A third article analyzed *CTCF* mutations in a larger cohort but did not provide detailed phenotypes for patients with R567W specifically. The main clinical features summarized for the overall *CTCF* cohort included intellectual disability in all patients, with cardiac defects in approximately 1/3 of patients (PMID: 31239556). Additionally, two other reports identified R567W in broader cohorts of patients with neurodevelopmental disorders and intellectual disability (PMID: 33644862, 34374989).

We agree with the reviewer that the clinical spectrum of *CTCF* mutations is highly variable. The lack of clearly defined cardiac phenotypes in reported R567W patients could be due to variability in manifestation. In addition, the primary focus on studies of these mutations has been intellectual disability, and cardiac findings may be less consistently evaluated or reported. In summary, while our mouse model showed cardiac defects, these defects have not been definitively described in R567W patients. We have modified the conclusion and discussion to clarify this and added a list of reported phenotypes for R567W patients. In the revised version, we have included the following statements:

1. *"Building on the observation that a clinically heterozygous CTCF mutation at c.1699C>T (p.Arg567>Trp) induces severe phenotypes, including intellectual disability, microcephaly, hypotonia, growth deficiency, delayed development, short stature, delayed bone age, and feeding difficulties"* (Page 3)

2. *"Furthermore, the suboptimal performance of heterozygous mice on the rotarod hinted at minor abnormalities in cardiorespiratory function, aligning with the phenotype observed in some human patients with CTCF heterozygous mutations. However, direct evidence linking the CTCF^{R567W} mutation to cardiac irregularities remains unclear and requires further investigation"* (Page 20).

3. Fig 2: why 2a, 2e-k doesn't have CTCF+/R567W results?

Response: We appreciate Reviewer #2 for bringing up this inquiry. Based on our observations and phenotype examinations, the *Ctcf*^{f+/R567W} mice did not exhibit significant divergence from the wild-type mice. Subsequently, our primary attention was directed toward the implications of homozygous mutations in subsequent experiments, for instance, Golgi staining, as presented in **Fig. 2i**, and calcium imaging, as demonstrated in **Fig. 2j and 2k**. For the indicators in the MEA experiment, the heterozygotes did not display any measurable deviation when compared to the wild type, so we included these data in **Supplementary Fig. 3f-i**.

We have clarified this point in the revised manuscript as follows: "*As the phenotypes of Ctcf^{f+/R567W} mice were generally consistent with those of Ctcf^{f+/+} mice according to preliminary observations and tests, we focused on examining the effects of the homozygous mutation on neural development and activity in subsequent studies*" (Page 7).

4. Fig 3d/Supplementary Fig.4c: which section of brain are these data? How do you know the difference were not because of they are different location of brain?

Response: We thank Reviewer #2 for this very valuable comment. The spatial RNA-seq data in the original manuscript were obtained from sagittal sections of the mouse brain. Due to the complexity of the experiments, the sections captured from wild-type and mutant mice were not perfectly matched, which is a significant limitation of these data. However, to mitigate the differences caused by different locations, we selected three comparable regions—the diencephalon and hindbrain (DH), ventricular zone of the dorsal pallium (Dpallv), and cartilage—for further DEG analysis based on the number of cells captured in each cluster.

In the revised version of our paper, we conducted single-nucleus RNA-seq (snRNA-seq) on the cortices of the brains of E18.5 *Ctcf*^{f+/+} and *Ctcf*^{R567W/R567W} mice to replace the previous spatial RNA-seq data.

Our snRNA-seq data produced the following results: "*cluster analysis identified nine cell populations, including radial glial cells, neural progenitor cells, neuroblasts, immature neurons, glutamatergic neurons, GABAergic neurons, oligodendrocyte precursor cells, microglia, and endothelial cells (Fig. 3d and Supplementary Fig. 4c). Our data revealed that the CTCF^{R567W} mutation had no significant effect on the cellular phenotype or landscape of the mouse cortex but led to noticeable shifts in specific cohorts. Specifically, the mutant mice exhibited a reduction in the proportion of stem-like neural progenitor cells and radial glial cells, as well as immature neurons, coupled with an increase in postmitotic neuroblasts and inhibitory GABAergic neurons (Fig. 3e). Additionally, a marginal decrease in excitatory glutamatergic neurons was observed (Fig. 3e)*" (Page 9). This dataset offers enhanced dependability in comparison to the original data. When combined with subsequent analysis, this approach provides a more comprehensive and coherent overall picture.

The DEG of which cluster were most overlap with DEGs from Fig.3a/3b? Does *Pcdh* genes appeared in any of these DEGs, if yes, which cluster?

Response: The DEGs from bulk RNA-seq in Fig. 3a/3b cover a much wider range of genes due to the utilization of total RNA from the samples, primarily revealing the average expression of the cell population. In comparison, the variability of gene expression in single cells is substantial, and certain genes may be robustly expressed in some cells but not in others. This variability can influence the results of differential analysis, potentially leading to the filtering out of possible DEGs. As a result, the DEGs identified from our previous spatial RNA-seq data and our new snRNA-seq data showed less overlap with the bulk DEGs (**Figure R1a, b**). The cluster of *cPcdh* genes, which exhibited relatively weak average expression according to the single-cell data, was also not detected among the DEGs. However, we separately analyzed the *cPcdh* genes in our new snRNA-seq data from the cortex. We observed downregulation of the *Pcdha* and *Pcdhβ* genes and upregulation of the *Pcdhy* genes, which is consistent with the findings of the bulk RNA-seq data. These changes in expression were consistent

across all subgroups (**Supplementary Fig. 4e**). The *cPcdh* genes were not identified among the DEGs in the single-cell analysis due to the high variability and relatively weak average expression of this cluster in single cells.

Figure R1. a. Venn plots showing the overlap of DEGs detected by bulk RNA-seq of neuron samples and DEGs identified by spatial RNA-seq in three comparable regions. b. UpSet plots showing the overlap of DEGs detected by bulk RNA-seq of neuron samples and DEGs of the main neuronal clusters identified by cortex snRNA-seq.

5. Supplementary Fig.4g: dCM1 cell number reduced the most lot. If you detected DEGs compare dCM1 vs dCM2/3/4, does the top genes also different between wt and mutant? Does any of these genes have implication in phenotype reported in Fig 1g?

Response: We performed overlap analysis of DEGs from dCM1, dCM2, dCM3, and dCM4. The results showed that dCM1/2/3 had relatively more DEGs, most of which were shared among these three cell types (**Figure R2a**).

We further performed Gene Ontology (GO) analysis using common DEGs (52 genes differentially expressed in dCM1/2/3), and the results showed that these genes were mainly related to ATP metabolic processes and oxidative phosphorylation (**Figure R2b**). The expression of these genes was altered in all three cell types but did not result

in a change in dCM2/3 cell numbers. Therefore, dCM1-specific DEGs (34 genes) may be responsible for the observed change in cell number. GO analysis of these genes revealed that they were related to cell communication and muscle cell development (**Figure R2c**), which might be the cause of the phenotype of dCM1 cells after CTCF mutation.

Our data offer some insights into the genotype-phenotype correlation. The downregulation of genes related to energy metabolism and myocardial contraction, coupled with the upregulation of cardiac development genes, suggests that the hearts of homozygous CTCF-mutant mice may be developmentally less mature than their wild-type counterparts at E18.5. This finding likely explains the observed ventricular hypoplasia phenotype. We have incorporated this information into the relevant discussion section in the manuscript, emphasizing that "*These gene pathways might correlate with the previously detected abnormal heart and lung morphology phenotypes and could contribute to potential respiratory failure at birth in these mutant mice (Fig. 1i and Supplementary Fig. 1h)*". However, we have yet to conduct an in-depth examination of the influence of any specific gene on the phenotype. Considering this intriguing aspect for future analysis, our research on heart-related insights continues in another project.

Figure R2. DEG analysis of dCMs between wild-type and homozygous CTCF-mutant heart tissues. a. Venn diagram showing the DEG overlap among dCM1, dCM2, dCM3 and dCM4. b. Gene Ontology results illustrating the biological processes associated with DEGs shared among dCM1, dCM2 and dCM3. c. Gene Ontology results illustrating the biological processes associated with DEGs specific to dCM1.

6. Fig 3k: the DEGs detected are from different technology thus I am not sure this is meaningful comparison. Also, not surprising if they are indeed cell type specific. I would be more attracted to dissecting some details. For example, Supplementary Fig.4j shown Lung cell does not change much that are consistent with Fig 1h. While Supplementary Fig.4g shown dCM1 reduced almost 2-fold that might explain Fig 1g. Make me believed CTCF mutant have strong effect in heart than lung. It might be worth to extend analysis to either support or disapprove these observations.

Response: According to our results, the CTCF mutation affects both heart and lung tissues. In terms of the number of DEGs (**Supplementary Fig. 5f, g**), changes in cell proportions (**Fig. 3h and Supplementary Fig. 5b**) and histological analyses (**Fig. 1g and Supplementary Fig. 1h**), CTCF mutations had a greater impact on heart tissue than on lung tissue. Our DEG data from heart snRNA-seq showed that the DEGs were mainly related to metabolic processes, muscle contraction and heart development (**Fig 3i**). These changes could cause abnormal development and impaired functionality of the heart. We theorize that the CTCF mutation could impact the regulation of essential genes associated with heart function. Presently, we are working on a project designed to delve deeper into the implications of CTCF R567W mutation on the heart.

7. Fig 4/Extend Fig5d,5e: all the motif analysis involved (U) motif were not legitimate. The (U) motif could have different space to CTCF core motif thus conventional motif analysis would not work. You need employ sequence clustering method like Nakahashi et.al (Fig4, doi: 10.1016/j.celrep.2013.04.024) and Hyle et.al (Fig 6, doi: <https://doi.org/10.1186/s13059-022-02843-3>). This would provide better

visualization and more insights. Another interesting analysis could be conducted by scanning the (U) motif and examining whether there is a significant difference in the proportion of this motif between the groups depicted in Fig4c. Also, what's the dash line in middle of Fig4c?

Response: We appreciate Reviewer #2's valuable suggestion. By incorporating the methodology outlined in the second reference, we undertook an analysis of the CTCF U motif. First, CTCF sites in brain tissues were categorized into four groups based on alterations in CTCF binding strength after CTCF mutation (**Supplementary Fig. 7d**). We identified the core motif for each CTCF site and subsequently extracted the upstream 20 bp sequence. Second, we divided the upstream sequences for each group into 10 classes; these are displayed on a heatmap from top to bottom in order from 1 to 10 (**Supplementary Fig. 7d**). Finally, we generated and reordered the sequence logo of each class according to the similarity. The results showed that CTCF binding sites with a greater decrease in binding after CTCF mutation tended to have a greater proportion of the U motif (Group 1: 3.81%, group 2, 8.57%, group 3:27.79%, group 4: 31.69%) (**Supplementary Fig. 7d**).

In addition to the canonical CTCF U motif, we also found some noncanonical U motifs at the downregulated CTCF sites (**Supplementary Fig. 7d**). To conduct a rigorous evaluation of the CTCF U motif, we performed clustering analysis using the upstream sequences of all downregulated CTCF sites in the three tissues. Consequently, we obtained 7 types of upstream motifs, which we labeled U1 through U7, with U1 corresponding to the canonical CTCF U motif (**Supplementary Fig. 7e**). Together, CTCF sites containing U motifs accounted for approximately 60% of the downregulated CTCF sites (**Supplementary Fig. 7f**) and approximately 20-30% of the total CTCF sites (**Supplementary Fig. 7g**) in each tissue.

Another interesting analysis could be conducted by scanning the (U) motif and examining whether there is a significant difference in the proportion of this motif between the groups depicted in Fig4c.

Response: Using the U motifs identified above, we found that the proportions of different U motifs at different types of CTCF sites were similar (**Fig. 4f** in the revised version).

Also, what's the dash line in middle of Fig4c?

Response: The dashed line in Fig. 4c (**Fig. 4f** in the revised version) represents the average CTCF binding strength in different groups of downregulated CTCF binding sites.

8. Fig 4a and Page 10: It's confusing that the mutant is "reduced CTCF binding genome-wide" and "by weakening its ability to bind the U motif". These together does the authors suggested the U motif is genome-wide?

Response: We apologize for the confusion caused by using "reduced CTCF binding genome-wide", which meant that the reduced CTCF sites were distributed throughout the genome, not that CTCF was reduced genome-wide. We have changed this text to "*These results suggest that CTCF^{R567W} partially reduces CTCF binding, potentially achieved by weakening its ability to bind the U motif*" (Page13). In the mouse brain, there were 14645 (14645/45021, 32.53%) CTCF sites that were downregulated twofold after CTCF homozygous mutation, and the remaining sites were not strongly affected.

Which is not true, for example, Nakahashi et.al detected about 18000 lost peaks in dZF11 and 10000 significant decreased peaks (out of 43000) if we re-analysis their mice data. Half the peaks decreased 2 fold have (U) motif(~5000), 60% peaks decreased 4 fold have (U) motif (~2000). I could reason why single mutant at ZF11 could have stronger effect than dZF11 from Nakahashi.

Response: We performed adjusted ChIP-seq experiments in endogenous CTCF-mutated tissues, which revealed the true change in CTCF binding. According to our data, the change in *Pcdh* gene expression was consistent with the binding changes in

CTCF and cohesin (verified by Rad21 ChIP–qPCR) (**Supplementary Fig. 8f, g**). These results indicate the reliability of our data.

The data in Nakahashi’s paper were obtained through CTCF overexpression via mutation of the histidine residue in the ZF11 domain, which may reveal trend of CTCF binding after mutation rather than reflecting the precise change. In contrast, our results were obtained from mouse tissues, and the R567 residue likely made a greater structural contribution to DNA binding than the H residue in the ZF11 domain (PDB: 5YEL). We reanalyzed the motif changes in brain tissue after mutation, as shown in **Supplementary Fig. 7d**. In fact, the proportion of disrupted canonical U motifs (highlighted in red) in our data closely resembles what was observed in dZF11 by Nakahashi et al., with no substantial differences. This similarity further substantiates the reliability of our data. In addition, as CTCF binding can affect DNA methylation and chromatin accessibility, we propose that CTCF mutation might induce a cumulative effect on CTCF binding during development. This could result in a more pronounced effect than what is typically observed in cell lines.

Besides, I could not find replicates for the ChIP-seq they provided at GEO and there is no reviewers’ token for GSA that I could check any processed data.

Response: We performed adjusted ChIP-seq in different tissues without replicates, and the data from three tissues showed similar results. The processed ChIP-seq data were uploaded to the GEO database (GEO number: GSE214692, access token: apgpqgoodbwrnop).

9. Fig 4g: does the TAD boundary affected have disrupted CTCF binding? Does the 4 groups have any bias on proportion of overlapping disrupted binding? Does the Disrupted CTCF binding at boundaries have (U) motif?

Response: We divided TAD boundaries into two classes (common (unchanged) and varied (enhanced or weakened)) depending upon whether the TAD boundary significantly changed after CTCF mutation and analyzed CTCF binding to these TAD

boundaries. The results showed that both common and varied TAD boundaries had downregulated CTCF binding sites (**Figure R3a**). The decrease in CTCF binding strength at varied boundaries was more pronounced than that at common boundaries (**Figure R3b**). The proportions of downregulated CTCF sites on the boundaries of the four types of TADs were comparable in the same type of tissue (**Figure R3c**), and the proportions of downregulated CTCF sites on TAD boundaries were similar for the different types of CTCF U motifs among the three tissues (**Figure R3d**).

Figure R3. Relationships between downregulated CTCF binding and different types of TAD boundaries. a. Number of different types of TAD boundaries. b. Boxplots showing the log_2 (fold change) values of CTCF binding at common and variant TAD boundaries. c. Percentages of the boundaries of different types of TADs with downregulated CTCF sites. d. Percentages of downregulated CTCF sites on TAD boundaries with different types of CTCF U motifs or without CTCF U motifs.

10. Fig 5c: I downloaded their .hic data on GEO but I could not reproduce the top panel regardless which normalization method used(see below). I could not convince myself there is a difference at this locus, and their values were huge compared to the value in the file. Please elaborate.

Response: We loaded wild-type and homozygous CTCF mutant Hi-C data of brain tissues into the WashU epigenome browser and adjusted the Hi-C data using 50 kb resolution, KR normalization and 50 as the maximum score. The figure below is the original screenshot (**Figure R4**). This TAD was clearly separated. The Hi-C heatmap in Fig. 5c was drawn with a nonlinear color scale to facilitate the visualization of TAD changes.

Figure R4. Original screenshot of brain Hi-C data in the WashU epigenome browser. Upper heatmap: wild-type brain; lower heatmap: homozygous CTCF mutant brain.

11. Page 12: They previously mentioned detecting "genome-wide" decreased CTCF sites with (U) motif. It's against their claim here "selectively reduced CTCF binding in the promoters of *Pcdhβ* genes". It would be interesting to check how many of the gene promoters have "decreased CTCF sites with (U) motif"? How many of these gene have decreased expression? Does log₂fc from *Pcdhβ* significantly lower than the log₂fc of these genes? And careful revise the conclusion here accordingly.

Response: As noted in our response to Question 8, the term "genome-wide" was not appropriately used here, we have revised the sentence with "*These results suggest that CTCF^{R567W} partially reduces CTCF binding, potentially achieved by weakening its ability to bind the U motif*" (Page13).

According to Reviewer #2's suggestion, our analysis revealed that 294 gene promoters harbored decreased CTCF sites with U motifs (including canonical and

noncanonical U motifs). However, only 4 genes were significantly downregulated after CTCF homozygous mutation (**Figure R5**). CTCF binding strength in the *Pcdhb17* promoter showed a greater decrease compared with others after CTCF mutation.

Regarding why only one *Pcdh* gene with a U motif in the promoter was identified, the CTCF binding strength in most *Pcdh* promoters was weak, and CTCF signal regions could not be identified as peaks using the default parameters of macs2. For the CTCF signal in the *Pcdh* promoter, we manually selected the CTCF signal region for further analysis. We have added this explanation to the Methods section as follows: "*The CTCF motif in the promoters of Pcdh genes was analyzed as follows. As the CTCF binding strength in most Pcdh promoters was weak, CTCF binding regions were rarely identified using the default parameters of macs2. Instead, CTCF binding regions in Pcdh promoters were manually selected based on the CTCF ChIP signal. Then, the CTCF core motif sequences and ± 20 bp flanking sequences were extracted, and sequence logos were generated with ggseqlogo*".

Figure R5. Bar plot showing the changes in CTCF binding strength in the promoters of downregulated genes after CTCF mutation in brain tissue.

12. Please provide side by side expression (KO vs WT) for PCDH genes in Extend Fig7f for six clusters of Extend Fig7k. Seurat Dotplot would be good choice. Would they only see difference only in "Radial glia" and "GABAergic" like their Fig 6f suggested? If not, why?

Response: We have included a side-by-side dot plot showing *PCDH* gene expression across cell types in the scRNA-seq data from one-month-old *CTCF^{+/+}* and *CTCF^{+/R567W}* cortical organoids (**Figure R6**). Our results demonstrated that *PCDH* genes exhibit heterogeneous single-cell expression, with only a subset of cells within each population expressing a given *PCDH* gene at appreciable levels. This finding is consistent with the known single-cell variability of *PCDH* expression in the nervous system. Consistent with the bulk RNA-seq trends, we observed reduced *PCDH* expression and fewer *PCDH*-expressing cells across all cell populations in the *CTCF^{+/R567W}* mutant organoids compared to those in the *CTCF^{+/+}* controls. However, we did not observe differences restricted only to radial glia and GABAergic neurons. Rather, this reduction was evident across all the cell types profiled. We propose that the *CTCF* mutation causes global dysregulation of the *cPCDH* locus, a phenomenon conserved across cell types. Over time, the accumulation of *PCDH* regulatory defects or other irregularities in neural development pathways caused by *CTCF* mutation could lead to widespread neural developmental defects and anomalies in cell differentiation and development processes. These abnormalities may eventually manifest at the whole-organoid level.

Figure R6. Dot plot showing *PCDH* gene expression across cell types according to scRNA-seq data from one-month-old *CTCF*^{+/+} and *CTCF*^{+/R567W} cortical organoids. The size of each circle reflects the percentage of cells within a specific cell type that expressed the given gene. The color indicates the average expression level of that gene within the specific cell population.

Minor:

Page 4: "Based on the number of different genotypes at E18.5, most *Ctcf*^{R567W/R567W} mice could fully develop, but they were lighter in weight and smaller than wild-type mice (Fig. 1c)." What does "Based on the number of different genotypes at E18.5" meant?

Response: We apologize for this confusion. We have modified this sentence as follows: "*Upon analyzing the genotyping results of numerous embryos at E18.5, we noted that most Ctcf^{R567W/R567W} mice developed into intact individuals, albeit considerably smaller than their wild-type counterparts*" (Page 5).

Page 4: "exhibited no coloration or respiratory distress and perished within 30 min" confusion. "no respiratory distress" or "have respiratory distress and perished"?

Response: We thank Reviewer #2 for pointing out this confusion. We have clarified this sentence as follows: "*...homozygous mutant mice (Ctcf^{R567W/R567W}) experienced respiratory distress and succumbed within 30 minutes (min) after birth*" (Page 5).

Page 4: "clinical phenotypes of short stature and developmental delay" first mentioned, reference?

Response: In the revised manuscript, we have described the clinical phenotypes of short stature and developmental delay associated with *CTCF* mutations earlier in the text, with references cited (Page 3). Therefore, we did not provide additional references for this statement.

Fig 2h: why the black line out of blue box is missing? Looks suspicious, please double check.

Response: We appreciate the Reviewer#2 for pointing out this mistake. This was an inadvertent error during figure formatting. We have corrected this issue in the revised figure.

Extend Fig5n: why the bottom panel have empty sequences?

Response: The empty sequences in the previous results were caused by space alignment. We have reanalyzed the motif and replaced it with **Supplementary Fig. 8i**. In this figure, we directly used sequence logos without alignment, and it is clearly

showed that the downregulated CTCF sites in the *cPcdh* locus contained the CTCF U motif.

Fig 6g/Extend Fig7k: why there are a lot missing value?

Response: We appreciate the reviewer for raising this point about the missing values. The gaps in **Fig. 6j** and **Supplementary Fig. 10k** in the revised version represent pathways or genes that were not significantly enriched or differentially expressed in certain cell populations according to the scRNA-seq analysis.

Reviewer #3 (Remarks to the Author):

The manuscript by Zhang et al establishes a mouse gene mutation knock-in model and human cortical organoid model from gene-edited human embryonic stem cells to investigate the impact of a pathogenic CTCF mutations that has been implicated in a human neurodevelopmental disorder. The breadth of innovative techniques probing chromatin architecture and gene expression that are impacted by disrupted CTCF binding often at a locus- or cell/tissue-specific manner is very insightful. The R567W knock-in mouse model is very informative and is the first model examining CTCF ZF mutation in the context of whole organismal development.

A number of issues have come up in my review, which I believe are problematic and need to be addressed by the authors. One major factor is the veracity of the human embryonic stem cell model, as how these cells were isolated and having appropriate controls could have a major bearing on the results. This matter clouds the clear phenotypic differences observed between heterozygous CTCF^{+/R567W} mice and the heterozygous organoids.

Major Comments

1. Title: It needs to mention that the chromatin rearrangement and abnormal neurodevelopment only occurs with the homozygous mutation. Abstract: Whenever

CTCF^{R567W} is mentioned, it needs to be qualified whether it is homozygous or heterozygous. It should also be stated that this mutation is within the DNA binding ZF region. ‘Developmental disorders’ (mentioned twice) – strictly speaking there is only one linked to CTCF, MRD21. Mention OMIM descriptor at appropriate place on Page 3.

Response: Thank the Reviewer#3 for the helpful comments. We have addressed these questions as follows:

For the title, we have respectfully maintained the current wording, as we believe it aptly reflects that, considering both mouse models and human cortical organoid models, both homozygous and heterozygous CTCF R567 mutations can lead to abnormalities, albeit to varying degrees. Although our primary focus was on chromatin rearrangement and abnormal neurodevelopment in the homozygous mutant, we do not believe it is necessary to emphasize this point in the title.

In the abstract, we have clarified whether the CTCF^{R567W} mutation is homozygous or heterozygous. Due to length constraints and the necessity for revisions, we have rewritten the abstract portion of the text. Additionally, we mentioned that this mutation resides within the 11th zinc finger DNA binding domain of CTCF.

Regarding your comment on developmental disorders, we have added the OMIM descriptor after the following sentence: "*These variations are highly correlated with potential symptoms subsumed under the classification of intellectual developmental disorder, autosomal dominant 21 (OMIM 615502), which can range from global developmental delay, intellectual disability, and short stature to autistic behaviors and symptoms resembling congenital heart disease*" (Page 3).

2. Page 3: it should be clearly stated that human mutations in clinical case reports only occur as heterozygous mutations.

Response: Thank the Reviewer#3 for this valuable suggestion. We have revised the manuscript to clearly state that the CTCF mutations reported in human clinical case studies to date have been observed only in the heterozygous state. Specifically, the

manuscript now includes the following statement: "*Recent discoveries in clinical and targeted sequencing studies have unveiled cases of heterozygous CTCF mutations, encompassing deletions, frameshifts, and missense mutations*" (Page 3).

3. Ext Data Figure 1b: For each developmental time point in this table a Chi-square analysis should be performed and presented in a new column. By my reckoning at E18.5 there is a sub-Mendelian ratio of R567W homozygous pups.

Response: We appreciate the Reviewer#3 for this valuable suggestion. We have updated the figure to include Chi-square *p* values in a new column (**Table S1**). Indeed, we observed a sub-Mendelian ratio of R567W homozygous pups at E18.5. Although we occasionally observed stillborn homozygous mutant pups during breeding (approximately 1 in every 4-5 litters), the majority of E18.5 homozygous embryos we collected for analysis were alive, although they were smaller in size and could not survive after birth, as described. Thus, we chose to perform most experiments on E18.5 embryos to reflect this viable prenatal stage. Statistically, we reflect the true ratios in our data. The possibility of embryonic lethality occurring before the stages examined here cannot be excluded. The sub-Mendelian ratio likely reflects sporadic cases of more severely affected mutant embryos that could result in earlier lethality and resorption.

Genotypes of Ctcf mutant mice						
stage	+/+	+/R567W	R567W/R567W	Total	Litters	P
E14.5	8	22	8	38	4	0.6227
E18.5	67	142	46	255	29	0.0341
P0	40	50	*32	122	14	0.0814
P1	36	43	0	79	14	<0.0001

* Died within 30 minutes after birth

Table S1. Genotype frequencies of progenies from the mating of *Ctcf* mutant mice. Chi-square analysis was performed to analyze the genotype distributions. The *p* values are presented.

4. Page 2: CTCF cellular level changes and cell death & Page 3: lean phenotype during juvenile stage in mice have been reported before. Adding PMID 30513694 would be appropriate here as supportive evidence (also at Page 5 Fig 1e).

Response: We agree that the previous study (PMID: 30513694) provides relevant supportive evidence for our current findings. We have cited this reference in the relevant sentences following Reviewer #3's suggestion.

5. Page 4: it should be stated that the R567W mutation occurs in ZF 11 of the 11-ZF DNA binding domain of CTCF, and also occurs adjacent to the exon 9 5' splice site (this is important, see later questions). Marked onto Figure 1 should be those CTCF ZF residues depicted as DNA binding (from PMID 29076501). Also Zn²⁺ co-ordinating residues should be marked with a shaded band or in different colour.

Response: We appreciate the Reviewer#3's valuable feedback. The location details of the R567W mutation have been further elucidated in the text as well as the accompanying figure. We have added that the R567W mutation exists within the 11th zinc finger domain of the CTCF protein and is proximal to the 5' splice site of exon 9 (**Fig. 1b** in the revised version). Furthermore, we highlighted the DNA-binding residues in CTCF zinc fingers, which are expected to interact with DNA bases or the phosphate backbone (residues 565-568), using a blue line; the Zn²⁺ coordinating residues are highlighted by a yellow band; and the mutation site is marked in red (**Fig. 1a** in the revised version). These modifications make the functional implications of the R567W mutation more explicit and enrich the information in the figure.

6. Further to Q4: the CGG to TGG mutation occurs in the last codon before the 5'SS. This could affect normal splicing of Cctf in mice by modifying the splice site or ESEs. The authors have only used an antibody (#07-729) that was raised with an immunogenic peptide representing in the C-terminus (aa 659-675). Therefore the authors must use a CTCF antibody recognising the N-terminus that will detect any potentially truncated proteins.

Response: The mutation occurs just before the 5' splice site, which may influence *Ctcf* splicing, but previous expression data suggest that this is not the case. By following the Reviewer#3's suggestion, we have probed mouse brain tissues with a CTCF antibody that identifies the N-terminus (CTCF: ab37477; the immunogen is a recombinant fragment corresponding to human CTCF aa 1-300). Western blotting revealed no difference in CTCF expression or splicing between the wild-type and mutant samples, and we detected no potentially truncated proteins (**Figure R7**).

Figure R7. Western blotting analysis of CTCF protein levels in brain tissues isolated from E18.5 embryos of different genotypes by using two different anti-CTCF antibodies.

7. Clarification: Didn't CTCF homo mice exhibit respiratory distress (Page 4)?

Response: The Reviewer#2 made a similar comment. We have corrected this sentence in the revised manuscript to "*...we observed that heterozygous mutant mice (*Ctcf*^{f⁺/R567W}) exhibited a birth state similar to that of wild-type mice (*Ctcf*^{+/+}), while homozygous mutant mice (*Ctcf*^{R567W/R567W}) experienced respiratory distress and succumbed within 30 minutes (min) after birth"* (Page 4).

8. Figure 1c and for any data in this manuscript that reports statistical significance – the numerical data should be reported in the text of the Results, ie mean +/- SD and give p-value.

Response: By following the Reviewer#3' suggestion, we have updated the figures, legends, and numerical data in the revised manuscript. For statistical data, the *p*-values have been added to the graphs.

9. Page 5: 'half dose of CTCF at R567' is a misleading or inaccurate statement. Please revise.

Response: We sincerely thank the Reviewer#3's valuable suggestion. We now feel that the text "half dose of CTCF at R567" could be misleading. We have revised this sentence as follows: "*These findings suggest that the half amount of wild-type CTCF is adequate to maintain normal functions in mice.*" (Page 6).

10. It is not explicitly mentioned, but were all mouse experiments performed with littermates? i.e. WT, het and KO mice from the same litters. This is an important distinction and is needed to support the rigour of any findings.

Response: Substantial efforts were made to ensure that genetics and the environment were comparable among groups. When the number of mice needed for an experiment was small enough to be obtained from a single litter, littermate mice were used for comparisons between genotypes. For experiments requiring larger numbers of mice, pups were selected from two or more litters bred during the same period and housed under identical conditions. We have added this information to the Mice section of the Methods in the revised version.

11. Figure 4: as CTCF binding due to R567W mutation is reduced, the U motif is being enriched. This suggests that those sites that are lost do not contain the U motif?? Unless, I am not understanding the Figure correctly (or the authors are not explaining it well). Does Figure 4b depict the sequence logo of those CTCF sites remaining or those that are lost due to R567W?

Response: We apologize that our diagram caused confusion. The original Figure 4b showed that the downregulated CTCF sites were enriched in U motifs. Based on the

suggestions from reviewer #2, we have conducted an additional analysis on the motif (Supplementary Fig. 7d, e). The overall conclusion remains unchanged; that is, sites with a greater decrease in CTCF binding upon mutation tended to contain a greater proportion of the U motif.

12. What percentage of all CTCF sites contain the U-motif? What is the nature and distribution of those sites lost with R567W?

Response: By using downregulated CTCF sites, we identified 7 types of U motifs. CTCF sites containing U motifs were extracted, which accounted for approximately 60% of the downregulated CTCF sites (Supplementary Fig. 7f) and approximately 20-30% of the total CTCF sites (Supplementary Fig. 7g) in each tissue.

Are they inter- or intragenic? Intronic? Proximal to promoter/enhancers?

Response: We annotated the downregulated CTCF sites and found that, compared with all CTCF sites, the proportions of downregulated CTCF sites decreased in the promoter regions and increased in the intergenic regions (Figure R8a). In addition, we analyzed the distribution of the distances between downregulated CTCF sites and their nearest promoters or enhancers, and results showed that the most of the downregulated CTCF sites were far from both promoters and enhancers (Figure R8b).

Figure R8. Features of downregulated CTCF sites after CTCF homozygous mutation. a. Genomic distributions of genome-wide CTCF binding sites and downregulated CTCF binding sites in the indicated tissues. The genomic features are color-coded in

the legend bar. The x-axis shows the cumulative percentage of genomic occupancy of each feature. b. Cumulative plots showing the distribution of distances between downregulated CTCF sites and their nearest promoters or enhancers.

Can they be matched with any public Hi-C data to determine if they are convergent or divergent sites?

Response: We used loops identified from a previously published paper (Enhancer–promoter interactions and transcription are largely maintained upon acute loss of CTCF, cohesin, WAPL or YY1, PMID: 36471071). We extracted loops with both anchors occupied with CTCF, and only kept the loops that one anchor possesses one single CTCF site. The number and direction of CTCF loops were collected, and the results showed that approximately 75% of CTCF loops had convergent CTCF sites, regardless of whether CTCF was downregulated (Figure R9).

Figure R9. Bar plot showing the percentage of CTCF loops with convergent or divergent CTCF sites. All represents all CTCF loops, one_dn represents CTCF loops with downregulated CTCF at one of the anchor sites, and all_dn represents CTCF loops with downregulated CTCF at both anchor sites.

13. As the crystal structures for overlapping portions of the CTCF ZF domain have been solved eg PDB 5YEL (ZFs 6-11). It would be useful to model the R567W mutation to

visualise the loss of DNA binding and informative to examine the impact on DNA binding to the U motif. A similar approach has been used in PMID 34657170 & 37047368

Response: Thank you for raising this thoughtful point. By utilizing PDB 5YEL (ZFs 6-11), it has been suggested that amino acids 565-568 in the CTCF zinc finger 11 form hydrogen bonds with the nucleotide bases and phosphate backbone of DNA (PMID 29076501). It has been theorized that the R567W mutation could perturb these interactions, which was addressed in previous study (PMID 23746550). Thus, the findings in our last version of manuscript did not include the results of structural modeling. In this updated manuscript, we have modeled the R567W mutation to visualize the potential loss of DNA binding to the CTCF U motif DNA by following the Reviewer#3's suggestion. The findings, detailed in our revised manuscript, show that the R567W mutation is positioned too far from C21's phosphoric acid group to form a pivotal hydrogen bond. This weakened binding could also impact the formation of the hydrogen bond in the adjacent residue R566, which is considered more critical for ZF11 binding (**Fig. 4a-c** in the revised version).

14. Page 10: Reference 12 is not an appropriate reference here, remove.

Response: We have removed this reference from our revised manuscript by following the Reviewer#3's suggestion.

15. GST pulldowns: it is not clear why R566C was being examined here in the context of a R567W? Was it to get a more effective loss of DNA binding ZF? Why is only the gel shift using the U-motif M2 oligo shown (& demonstrating weak binding), when the M1 oligo (mentioned in Supp Table 2) is clearly of a core consensus CTCF site, but no data is shown? The authors must show that the 11-ZF recombinant CTCF protein binds a core consensus site. I suspect that with a longer oligo (C + U motif), there will be no loss of binding, but this still needs to be tested, as this speaks to the fact that only some CTCF binding is lost due to R567W mutation, but these sites are critical for tissue-

specific development. Also, a (C + mutant U motif) oligo should be used as well). Also it is not clear how the M2 oligo sequence here relates to the U motif? It was hard to do complementary alignments based on the very small text provided for the U motif sequence logo in the Figures.

Response: We agree that our description for this part was too simplified in the previous version. In the revised manuscript, we have expanded this section to provide better explanations.

Based on prior structural analyses, R567 was shown to form a hydrogen bond with the phosphate backbone of DNA at C21, while R566 was shown to form hydrogen bonds with the G4 and G23 bases, suggesting a higher influence by R566 on DNA binding (**Fig. 4a-c**). Therefore, we also introduced the R566C mutation in our *in vitro* EMSA and GST pulldown assays to evaluate the combined effect of R567W and R566C on the affinity of the CTCF zinc finger for DNA probes.

The M1 and M2 oligo sequences were designed based on previous studies (PMID: 28315159). M2 represents the CTCF U motif. We apologize for exclusively presenting the M2 data in the initial submission and overlooking the inclusion of M1 data. As requested, we have incorporated EMSA and GST pulldown data showing the binding of wild-type, R567W-mutant, and R567W/R566C double-mutant CTCF zinc fingers to probes representing the CTCF U, C, U+C and mutant U+ C motifs, respectively. Our updated results revealed that compared to wild-type zinc fingers, both the R567W single-mutant and R567W/R566C double-mutant zinc fingers exhibited reduced binding to all motifs, albeit to varying extents. This was quantified by the EMSA-bound/free ratios (**Figure R10b-i**) and identified by dot blot (**Figure R10j, k**). These results are consistent with our *in vivo* ChIP-seq data, which show that the R567W mutation results in reduced binding at numerous sites, particularly those containing the U motif. We extend our gratitude for your constructive input, enhancing the completion of our *in vitro* binding experiments.

Figure R10. *In vitro* binding analysis of CTCF ZF, ZF with R567W-mutant, and ZF with R567W/R566C double-mutants to CTCF motifs. **a.** Coomassie brilliant blue staining showing the purified GST, GST-CTCF-ZF-wt, GST-CTCF-ZF-mut (R567W) and GST-CTCF-ZF-dmut (R566C+R567W) proteins electrophoresed on SDS-PAGE gels. **b-e.** EMSA experiments showing the migration of Cy5-modified motif probes (M2, U motif in **b**; M1, C motif in **c**; M1+M2, U+C motif in **d**; M1+M2(mut), mutant U+C motif in **e**) on native polyacrylamide gels after coincubation with GST fusion proteins. **f-i.** The EMSA results for **b-e** were quantified, and a curve of the ratio of bound to free probe for the motif probe with increasing protein concentration was generated. **j.** Flow chart of the GST pulldown assay (top). Dot blot assay showing the enrichment of the biotin-labeled U motif probe by purified GST fusion proteins (bottom left). Quantification of the dot blot results (bottom right). **k.** A GST pulldown assay combined with a dot blot assay showing the enrichment of biotin-labeled C, U+C, and mutant U+C motif probes by purified GST fusion proteins.

16. Figure 4g: four TAD types are defined here, but 5 TAD descriptors are then examined in Figure 5. What happened to ‘boundary’ in the 4g analysis?

Response: We defined the four TAD types in **Fig. 4h** (revised version). The 5 descriptors in **Fig. 5a** include the TAD boundaries and four types of TADs. We have revised the legend of **Fig. 5a** to "*Correlation of DEG positions with TAD boundaries and four types of TADs post CTCF^{R567W} mutation in different tissues*" for clarification.

17. I suspect loss of potential RNA interactions due to R567W mutation are playing a bigger role in both models than what has been stated (page 17). This is a bit of a shortcoming as no declarative statement about RNA binding can really be made, other than speculation.

Response: We agree that our statement about the potential role of RNA interactions in the R567W mutation was speculative and lacked experimental evidence. Thus, we have removed the discussion about RNA binding effects from current version of manuscript.

18. Figure 6: how the R567W-mutant H1 human embryonic cell lines were established is problematic and a major flaw of this paper. The authors only describe one het clone and one KO clone (from the data I can see), but then base a whole range of scRNAseq and bulk RNAseq analyses on these 2 clones. Furthermore, additional clones of WT, het and KO need to be examined. The variation introduced by the method that clones were isolated may be sufficient enough to explain the discrepancy between human and mouse heterozygous results. Also, Please update methods with length of time it took to isolate clones.

Response: We apologize for the insufficient details on the generation of the edited cell lines in the original manuscript. After nearly 2 years of gene editing, we successfully generated two homozygous and several heterozygous edited clones. For each genotype, we selected two clones for each phenotype to conduct organoid experiments and bulk

RNA-seq analysis. We have provided detailed information in the **Methods** section as follows:

“Generation of CTCF^{R567W} point mutation-containing hESC lines

Human H1 cells harboring the CTCF^{R567W} mutation were constructed via CRISPR/Cas9. sgRNAs were designed with an online tool (<http://benchling.com>). The ssODN containing the desired mutation and enzymatic cleavage site was used as a template for homologous repair. The sgRNA primers were synthesized, annealed, cloned and inserted into the pX459 vector (Addgene Cat #62988). The Cas9-sgRNA vector and ssODN were transfected into cells for gene editing using FuGENE HD (Promega, Cat# E2311) transfection reagent. The cells were screened with medium containing 1 µg/mL puromycin (Gibco, Cat# A11138-03). After 48 h of screening, the surviving cells were digested into single cells using Accutase for passaging into Matrigel-coated 6-well plates. After approximately 10 days of culture, the single cells were successfully expanded into individual clones. Over one hundred clones were observed under a microscope and manually picked out using a micropipette. The selected clones were transferred to 96-well plates and further cultured for approximately 3-5 days until they reached a sufficient size. Subsequently, the clones were digested using 0.5 mM EDTA in PBS to detach them from the culture plates for passaging and amplification. Once the clones reached a certain quantity, a portion of the cells was collected for subsequent genomic DNA extraction and PCR genotyping, followed by enzyme digestion and Sanger sequencing for identification of correctly edited clones.

Detailed information on clones and editing strategies: The first round of editing in wild-type cells using sgRNA1 (target: AACATTTACACGTCGGGTAA) and ssODN1 (introducing the EcoR I site) yielded several heterozygous clones, which were verified by genotyping to have one allele that was correctly mutated and the other allele with a 6 bp deletion. These clones were referred to as heterozygous clone 1. Several wild-type clones were also retained after transfection and were selected as editing control wild-type clones. In the second round of editing, sgRNA2 (target:

TGGGAAAACATTTACACGTC) and ssODN2 (introducing the Mlu I site) or ssODN3 (introducing the EcoR I site) were used to edit heterozygous clone 1 and wild-type cells, respectively. This process yielded homozygous clones 1 and 2 and heterozygous clone 2. The clones were verified by genomic PCR, enzyme digestion, and Sanger sequencing. cDNA PCR (with primers for exons 8-10 [aa 469-644]) and sequencing also confirmed that the clones were correct (Supplementary Fig. 9a-d). The sequences of the targeted sgRNAs, ssODNs and genotyping primers used are listed in Supplementary Tables 1 and 2.”

With respect to the organoid experiments, we performed multiple preliminary differentiations to characterize clone phenotypes (e.g., smaller sizes, altered neural marker expression). We then utilized two clones of each genotype that displayed representative phenotypes for quantitative organoid size and differentiation experiments, pooling data from biological replicates (previously, we combined data from both clones). Bulk RNA-seq revealed the influence on neurodevelopment, notably the effects on *cPCDH* genes, corroborating the observations from mouse models. To further understand the impact of heterozygous mutations at the single-cell level, we further performed scRNA-seq experiments as a supplement, based on the consistency of the two clones in terms of organoid size, survival rate, and trends in changes in the expression of some neuronal markers. Each scRNA-seq dataset was derived from a mixture of 3-4 organoid spheres, and associated results, such as alterations in cell populations, including faster depletion of stem-like cells and premature development of GABAergic neurons, were identified. This phenomenon was consistent with our new cortical snRNA-seq data of homozygous mutant mice.

We concur that data from multiple clones would be more robust and acknowledge this limitation in our current study. Consolidating scRNA-seq across various lines, clones, and differentiation batches could provide mechanistic insights, as demonstrated by Paulsen et al. in their Nature paper, where they analyzed large sets of single-cell cortical organoid data to reveal how the autism risk genes *SUV420H1*, *ARID1B* and *CHD8* converge to disrupt neuron development. However, such an extensive scRNA-

seq analysis exceeds the scope of the current study but merits further investigation in a follow-up study. Interestingly, we observed that CTCF R567W heterozygous organoids led to the premature development of GABAergic neurons. This finding is consistent with our new snRNA-seq data for homozygous mutant mice (**Fig. 3d-g** in the revised version). These findings are also in agreement with those of Paulsen et al., as reported in their Nature paper. Therefore, it is plausible that CTCF mutations, similar to autism risk factors, may trigger neurodevelopmental disorders through similar cellular development pathways.

Furthermore, no ‘scrambled’ or non-targeting gRNA control was used for wildtype cells. Effectively the authors are comparing cells subjected to the transfection, single-cell selection process and expansion process to cells (WT) that have not gone through this process. As it stands there are quite considerable differences between WT, het and KO (Ext Fig 7a). Without this control, the human cortical organoid study is invalid. Furthermore, only one clone was selected each for het and KO. This does not control for clonal variation which can emerge during the gene editing and screening process.

Response: We greatly appreciate Reviewer#3 for raising these valuable suggestions. We have provided more detailed information on the cloning strategies in the revised Methods. In summary, to facilitate legitimate comparisons and offset consequences stemming from the editing procedure itself, we adopted wild-type clones that underwent an identical transfection, single clone selection, and amplification process as the edited clones. These clones were subsequently genotyped and confirmed to retain their wild-type genetic configuration. Therefore, any effects from the editing process itself would be accounted for when comparing these wild-type controls to mutant clones. Furthermore, we conducted functional experiments by using two clones for each phenotype in this new manuscript.

What is more, the het clone that is used (Ext data 6b) has one mutant allele (W567) but has a 6 bp deletion in the 5’SS of the second ‘normal’ allele. This 6 bp deletion actually

weakens the splice donor site (maximum entropy score: 10.17 to 4.20 (MaxEntScore)). This could result in alternative splicing at this allele, resulting in a truncated CTCF protein, which cannot be detected with the C-terminal specific antibody used here. The authors need to validate that these cells express full-length (130 kDa) CTCF using an antibody recognising the N-terminus. Also they should perform RT-PCR across exon 9 to 10 to show that no alternative splicing is occurring (and truncated CTCF eventuating).

Response: We sincerely thank the Reviewer#3 for the valuable comments. Following Reviewer#3's suggestion, we probed edited hES cells with anti-CTCF antibody recognizing the N-terminus (CTCF: ab37477; the immunogen is a recombinant fragment corresponding to human CTCF aa 1-300). Western blot analysis revealed no difference in CTCF expression or splicing between wild-type and mutant cells, and could not detect any potentially truncated proteins (Figure R11). Additionally, RT-PCR and genomic Sanger sequencing across aa 469-664 (primers in exons 8-10) also demonstrated no splicing alterations (**Supplementary Fig. 9b, d**).

Figure R11. Western blot analysis of CTCF protein levels using two different anti-CTCF antibodies.

19. Figure 6a, b: is this depicting survival rates beyond two months (Page 13)? As only data up to day 31 are shown.

Response: We have updated Figure 6 in the revised manuscript to include organoid images at Days 20, 30, and 60, respectively, along with quantitation of organoid survival rates throughout the differentiation process. As described in the revised manuscript, the homozygous organoids displayed significantly developmental defects,

manifesting as poor self-organization, reduced survival rates beyond 2 months, and substantially decreased sizes during differentiation, as evidenced by images captured at Days 20, 30, and 60 (**Fig. 6a, c, d**). Conversely, the heterozygous organoids initially exhibited smaller sizes but reached dimensions comparable to those of the wild-type organoids by Days 60 and 90 (**Fig. 6a-d**).

20. Page 17: ‘The DEGs caused by CTCF^{R567W} might be affected by direct or indirect mutation cascades, or these DEGS might aggregate...’ – cascades and aggregate are inaccurate terms here. Page 17: imitate=phenocopy.

Response: We apologize that the terms "cascades", "aggregate" and "imitate" were inaccurately used in the original manuscript. We have revised the relevant sentences as follows: "*CTCF^{R567W} exerts a broad and diverse impact on gene expression across distinct cell types, with DEGs potentially influenced by direct or indirect downstream effects of the CTCF^{R567W} mutation. These effects may accumulate and become more evident at E18.5 during extended embryonic tissue development*" (Page 21). Additionally, we have replaced "imitate" with "phenocopy" in our new version of the manuscript according to the Reviewer#3’s suggestion.

21. Page 17: Its worth adding a comment in here after discussion of R567 mutation: ‘As with other CTCF ZF mutations, locus-specific effects on DNA binding may result, leading to a spectrum of loss or even gain-of-function phenotypes (PMID 34657170). Note also for Discussion, genome-wide analysis of CTCF binding using a mutant CTCF (in the 11th ZF) still resulted in over 60% sites being maintained (PMID 23707059).

Response: By following the Reviewer#3’s suggestion, we have added relevant statements to the corresponding discussion section as follows: "*The single amino acid mutation R567W within ZF11 of CTCF significantly impacts CTCF binding to chromatin, particularly at sites containing the upstream motif. This finding is consistent with earlier findings indicating that mutations within ZF11 of CTCF primarily affect sites with this motif, yet over 60% of binding sites are maintained (PMID 23707059).*"

The altered binding specificity of CTCF^{R567W} at sites with the U motif can be attributed to the essential functions of ZFs 9-11 in recognizing this motif and flanking sequences involved in boundary insulation. This finding underscores the requirement of R567 for sequence-specific DNA-binding activity and the chromatin function of CTCF, which ultimately influences specific biological phenotypes. Similar to other CTCF ZF mutations, alterations in DNA binding at specific loci can potentially lead to a spectrum of phenotypes, ranging from loss to gain-of-function effects (PMID 34657170)" (Page 20).

22. Figure 1e-k, why was CTCF^{+/+}/R567W mouse data not acquired here?

Response: We guess that the Rviewer might refer to Figure 2e-k, not Figure 1e-k. This question has also been raised by the Reviewer #2. In numerous observational and phenotypic experiments, we discerned that *Ctcf*^{+/+/R567W} mice exhibited no notable differences from wild-type mice. Our primary focus and investigative emphasis were centered on the effects of homozygous mutations in certain experimental setups, such as Golgi staining, as shown in **Fig. 2i**, and calcium imaging, as shown in **Fig. 2j and 2k**. For the indicators in the MEA experiment, the heterozygotes did not display any measurable deviation when compared to the wild type, therefore we included these data in **Supplementary Fig. 3f-i**.

23. Summary Figure (Fig 5f) for the *Pcdh* locus: cohesin rings are depicted here in the chromatin architecture of the *Pcdh* locus, but are not examined in the manuscript. Whilst CTCF R567W mutation may not impact cohesin binding genome wide, RAD21 ChIP datasets are not examined. This is an important aspect of chromatin looping dynamics. Please comment.

Response: We agree with the Reviewer#3 that it is important to examine cohesin (RAD21) occupancy after the CTCF R567W mutation. We performed RAD21 ChIP-qPCR to measure RAD21 binding at promoters and enhancers across the *cPcdh* locus in brain samples in both *Ctcf*^{+/+} and *Ctcf*^{R567W/R567W} mice (**Supplementary Fig. 8f, g**).

Consistent with the changes in CTCF binding, we detected decreased RAD21 binding at the promoters of *Pcdhβ* genes and increased binding at *Pcdhγa5*, *b4*, and *b5* after CTCF^{R567W} mutation. RAD21 occupancy was also reduced at the enhancers. These results support our model that CTCF^{R567W} disrupts chromatin interactions at the *cPcdh* locus through coordinated effects on CTCF and cohesin binding.

Minor Comments

1. Abstract: damnification is not useful scientific term here.

Response: We apologize for the inappropriate use of "damnification" in the abstract. We have revised the original text to "*Mice with a homozygous CTCF^{R567W} mutation exhibited growth impediments, resulting in postnatal mortality, and deviations in brain, heart, and lung development at the pathological and single-cell transcriptome levels*" (Page 2).

2. Page 3: code shifts is not an appropriate scientific term, use 'frameshift'.

Response: We have replaced the term "code shifts " with "frameshift" in our revised manuscript.

3. Page 5: 'thicary interstitia'?

Response: We have changed the corresponding description in the new version as follows: "*Additionally, we observed a thickened alveolar interstitium and dense, noninflated alveoli in the lungs of E18.5 and postnatal day 0 (P0) Ctcf^{R567W/R567W} mice*" (Page 6).

4. Page 5: Revise: 'homozygous CTCF^{R567W} mutation causes lethality at birth, developmental delay and cardiorespiratory deficits, overlapping clinical case-related phenotypes to some extent.'

Response: By following the Reviewer#3's suggestion, we have rephrased these sentences as follows: "*In summary, our results indicate that the homozygous*

CTCF^{R567W} mutation leads to birth lethality and developmental delay in mice, somewhat mirroring known clinical phenotypes" (Page 6). "Furthermore, the suboptimal performance of heterozygous mice on the rotarod hinted at minor abnormalities in cardiorespiratory function, aligning with the phenotype observed in some human patients with CTCF heterozygous mutations. However, direct evidence linking the CTCF^{R567W} mutation to cardiac irregularities remains unclear and requires further investigation" (Page 19).

5. CM is not defined.

Response: We thank the reviewer for bringing this up. We have added a definition to clarify that "CM" stands for "cardiomyocyte" in our new manuscript.

6. Page 10: 'To determine whether CTCF^{R567W} disrupts chromatin organization'

Response: We have revised the original sentence "To determine whether CTCF^{R567W} regulates chromatin organization" to "*To investigate whether CTCF^{R567W} disrupts chromatin organization, Bridge Linker-Hi-C (BL-Hi-C) was performed...*" (Page 13).

7. Page 21: 'RNApure'?

Response: This is an accurate term. The RNA extraction and purification kit that we used was the RaPure Total RNA Micro Kit (Magen, Cat# R4012-03).

8. Figure 1c: Figure key not needed if you use x-axis labels. Please check you are using the right stats here? A one way ANOVA and multiple testing between all 3 genotypes may be more appropriate. Figure 1e: make gender symbols more obvious (underlined and labelled under both genotypes).

Response: We thank the Reviewer#3 for the valuable suggestions. We have revised the figures and performed the statistical analysis using one-way ANOVA with multiple comparisons among all 3 genotypes in the new manuscript.

9. Ext data 3j & k: what do the units pfu represent?

Response: We apologize for the incorrect use of "pfu" in the previous version, which should be "rfu", referring to relative fluorescence units. We have corrected this in the new version of the manuscript.

10. The sequence logos in Figure 4 and Ext data Fig 5 are too small too be useful.

Response: We have reanalyzed the CTCF motif utilizing the method recommended by the Reviewer#2. All motif results in the original Fig. 4 and Supplementary Fig. 7 have been updated with new results, which all showed good sequence logos.

11. Ext Data Fig 6: 'hete' is not standard usage

Response: Thank you for catching the nonstandard abbreviation "hete" in our figures. We have revised the figure legends to indicate the following: Het: heterozygous; Hom: homozygous.

12. Western blot size markers should be kDa not KD

Response: We have carefully checked all the Western blot figures and corrected the size markers to the units "kDa".

Reviewers' Comments:

Reviewer #1:

Remarks to the Author:

The authors have properly addressed my minor comment.

Reviewer #2:

Remarks to the Author:

The authors properly addressed all my questions. The only puzzle I have is how a single mutant could have stronger effect on CTCF binding than mutant of whole ZF11(Nakahashi). One possibility is lacking of biological replicates for ChIP-seq might lead to calling many false positives. I would suggest discuss these to avoid misinterpretation.

Minor:

line 252 and 255: I believed the figures referred are not right. Please carefully review the manuscript to avoid typos like these.

Sup 5A: the P from Pecam1 position moved?

Should be described in legend: "The dashed line in Fig. 4c (Fig. 4f in the revised version)"

line 308: If I look into their Sup 7d. Many clusters looks also like U motif such as Group2:Clusters 3/6/8/9/5; Group3: 1/2/4/6/7/9/10; Group4: 7/8/10/1/2/6/4. It's not clear how they selected those highlighted clusters. Software like tomtom from meme suite could be used to check which of those motifs match U motif.

Reviewer #3:

Remarks to the Author:

The authors have done a good job in addressing most comments by the reviewers. This has greatly improved the strengths of the manuscript. A number of major and minor comments now follow from my review which should be addressed to fix some errors and some erroneous statements.

Major Comments:

Abstract: the first line is awkward, and there is too much specific detail in this first line for a potential Nature Comms article. It puts too much emphasis on one point mutation in one gene. Note this is not TP53 R172H or KRAS G12V. The importance of this study is that this is the first report of the functional impacts of a clinically-relevant mutation in CTCF, and the effects this has on chromatin organisation and development. The last sentence of the abstract might be suitable for a specialised human genetics journal but not a journal of broad readership. Readers will say 'so-what'?

Abstract: 'it hindered CTCF binding to its upstream motifs' – this will make no sense to a general readership. What does this really mean? How about 'Peripheral motifs upstream to the core consensus site'

Line 97: 'CTCF R567 a conserved residue....' It would be stronger if you could state 'a conserved residue in all known or most orthologues. '...is located proximal to the splice donor site in exon 9.

Line 100: The whole 11 ZFs are in the DNA interacting region of CTCF. '....specifically contacts DNA'.

Line 113: 'Ctcf hemizygous mice'

Line 121: RT-qPCR shows a significant increase in CTCF? Therefore this statement is not correct.

Line 146: 'These findings suggest that half the amount of wild-type CTCF....' This is very clunky and inaccurate. Total CTCF dosage has not changed as demonstrated by your careful WB analysis, despite the presence of mutant CTCF. I would suggest something along the lines of: 'Unlike Ctcf hemizygous mice which exhibit loss of CTCF expression in the brain (Ref PMID 32816606), a mostly normal brain function can be observed in Ctcf+/R567W mice where CTCF expression is preserved'.

Line 151: what known 'clinical phenotypes'? Please specify.

Lines 189+: The bulk RNA-Seq of brain shows a lot of heterogeneity between WT and R567W mice

(Supplementary Figure 4), whilst the human cortical organoid data is very consistent. I would suggest that the fact that E18.5 embryos were collected with no regard for sex/gender of the embryos accounts for this. Can the authors comment whether sexes are mixed here and what samples were actually use for the RNAseq? This will definitely affect DEGs. See also line 415. Is this this limited due to heterogeneity introduced through gender?

Line 242: Section heading does not make sense.

Lines 269-270: Rephrase. CTCF ZFs engage DNA not vice versa.

Lines 274-5: R567W does not shift the tryptophan, it shifts the ZF. 'Phosphoric acid' is not correct here, 'phosphate backbone' or 'phosphate ion' is more accurate here.

Line 305: Is this the core consensus? If so, please state.

Line 330: I can't see how the insulation score is noticeably altered? Please make colour schema in Fig 4g more obvious here. Also Supp Fig 8d, are in a different but more effective colour scheme.

Line 384: I don't see these as repercussions? I would say functional consequences.

Lines 546-7: This sounds too narrow. 'Successful engagement of peripheral CTCF ZFs such as ZF11 containing DNA-binding residue R567 is essential for'

Box and whiskers plots for many data visualisations but it is not described what is being depicted ie 95% confidence intervals? Also for additional data in Supp Fig3 f to I – bar plots are used to depict similar data? Keep consistent.

Fig.1a: The ordering of the species is puzzling. It would look better and make better sense if they were ordered from higher-order to lower-order species. Also please indicate those residues which make up ZF11 based on UniProt annotation. Bovine is misspelt.

Fig.2a: what comparisons are the p-values generated from? It is unclear.

Fig 1g, Fig2 and others: It would be more helpful if all graphs doing measurement of multiple samples eg mice are labelled on the x-axis with 'n'. This would save having to find this information in the legend.

Fig 4d: The core consensus sequence logo must be shown here for both genotypes to show that it is invariant, while there are changes in U motif binding due to the presence of R567W

Minor Comments:

Line 67: 'to emulate relevant phenotype' – this sounds awkward, please revise.

Line 76: CTCF should be Ctcf; italics for all usage of CTCF/Ctcf gene and mRNA.

Line 103: What is 'expanded reproduction'?

Line 339: EP abbreviation is not necessary.

Line 395: 'manifesting as poor organisation'.

Line 406: spelling of VGLUAT1.

Line 481: 'its effects of CTCFR567W'?

Lines 495-6: Rephrase for clarity.

Line 518: Influence x2

Materials & Methods: all % should be qualified with w/v or v/v.

Line 619: Addgene cat no was stated earlier.

Line 621: What is the purpose of the puromycin?

Line 713: 'with'

BL-Hi-C and QHR-4C are not defined in Materials and Methods.

Line 1066: check 'frozen a chryotome' phrasing.

Ref 14 is not in title case.

Line 1495: M & F would be more effective here in legend and graph, not symbols.

Line 1503: i not I

Line 1527: indicated

Line 1567: gb7CSE is not described as to what it is?

Fig. 2b: y-axis 'area'

Fig. 3c: label 'involved'

Fig 3e: Colours are too hard to distinguish here.

Fig 3f: Colours for 'down' & 'up' are switched/different to Supp 4B which is confusing.

Fig 4a: PDB ID is not needed; label DNA strands with 5' and 3' ends.

Fig 4d: The core consensus sequence logo must be shown here for both genotypes to show that it is invariant, while there are changes in U motif binding due to the presence of R567W

Fig 6a: labelling size should increase.

Supplementary Information

1a: 'CTCF heterozygous mice'

2k: y-axis 'Interaction'

5a: Pecam1 has become separated

5h: 'transporter'

9a: 'editing' x2

9c: a short vertical line should be added to mark exon/intron boundaries.

Point-by-point Responses to The Reviewers

Reviewer #1 (Remarks to the Author):

The authors have properly addressed my minor comment.

Response: We are delighted that the Reviewer #1 was satisfied with our responses to his/her comments.

Reviewer #2 (Remarks to the Author):

The authors properly addressed all my questions. The only puzzle I have is how a single mutant could have stronger effect on CTCF binding than mutant of whole ZF11(Nakahashi). One possibility is lacking of biological replicates for ChIP-seq might lead to calling many false positives. I would suggest discuss these to avoid misinterpretation.

Response: Although our findings were supported by multiple lines of evidences, we acknowledge that the lack of biological replicates for adjusted ChIP-seq experiments might lead to the identification of false positives, and we agree that individual variability and batch effects could also contribute to heterogeneity.

Regarding the comparison with the data from Nakahashi's paper, we noted that their data were obtained through overexpression of biotinylated CTCF with the mutation of the second histidines within ZF11 in cell lines. In contrast, our results were obtained from mouse tissues and suggest that the R567 residue might contribute more significant to DNA binding than the histidine residue within zinc finger domain, as supported by our structural modeling data. From Nakahashi's data, there were about 10,000 significantly decreased peaks out of 43,000, while in this study, there were 14,645 significantly decreased peaks out of 45,021 by CTCF-R567W mutation. Based on the above analysis, we conclude that the mutations for both CTCF-R567W and the second histidines within ZF11 on CTCF binding have the similar effect. It could be within a tolerable range and might be due to differences in the systems and methods used. Actually, similar results that we have obtained from mutated mES cells (data not shown) also supported our findings, although these data were not included in the manuscript due to relevance and space considerations.

By following the Reviewer's suggestion, we have added the following sentences into the Discussion section "*In our study, we took advantage of adjusted ChIP-seq methods for CTCF in mouse tissues and found that a single point mutation could significantly alter the chromatin functions of CTCF, which had the similar effect as observed after ZF11 deletion. It's important to note that individual heterogeneity and variations in methodologies and systems used across different studies could also lead to the certain biases*".

Minor:

line 252 and 255: I believed the figures referred are not right. Please carefully review the manuscript to avoid typos like these.

Sup 5A: the P from Pecam1 position moved?

Response: Thank the Reviewer for careful reading. We have carefully revised the manuscript and corrected these errors.

Should be described in legend: "The dashed line in Fig. 4c (Fig. 4f in the revised version)"

Response: We have added the description in the legend of Fig. 4c to explain the dashed line as follows: "*Hydrogen bonds are shown as dashed lines*".

line 308: If I look into their Sup 7d. Many clusters looks also like U motif such as Group2: Clusters 3/6/8/9/5; Group3: 1/2/4/6/7/9/10; Group4: 7/8/10/1/2/6/4. It's not clear how they selected those highlighted clusters. Software like tomtom from meme suite could be used to check which of those motifs match U motif.

Response: The canonical CTCF U motif is characterized by the core sequence TGCA~~X~~TXCC, as identified in the paper "A Genome-wide Map of CTCF Multivalency Redefines the CTCF Code" (PMID: 23707059, 2013) and the paper "CTCF chromatin residence time controls three-dimensional genome organization, gene expression and DNA methylation in pluripotent cells" (PMID: 34326481, 2021). We did not use tomtom to evaluate the similarity between the newly generated U motif with the

canonical U motif. Instead, the motifs that we chose clearly exhibited the feature of canonical U motif, illustrating that the downregulated CTCF sites contain more canonical U motifs. Other motifs that only partially resemble the canonical U motif were not highlighted.

Reviewer #3 (Remarks to the Author):

The authors have done a good job in addressing most comments by the reviewers. This has greatly improved the strengths of the manuscript. A number of major and minor comments now follow from my review which should be addressed to fix some errors and some erroneous statements.

Response: We are grateful for Reviewer #3's meticulous review and valuable corrections on our manuscript and figures. We believe that the Reviewer #3's insights could indeed improve the accuracy and clarity of our work.

Major Comments:

Abstract: the first line is awkward, and there is too much specific detail in this first line for a potential Nature Comms article. It puts too much emphasis on one point mutation in one gene. Note this is not TP53 R172H or KRAS G12V. The importance of this study is that this is the first report of the functional impacts of a clinically-relevant mutation in CTCF, and the effects this has on chromatin organisation and development. The last sentence of the abstract might be suitable for a specialised human genetics journal but not a journal of broad readership. Readers will say 'so-what' ?

Response: We are appreciated for the Reviewer #3' insightful suggestions on refining the abstract. By following the Reviewer #3' suggestion, we have made the following revisions:

We have added one sentence as the first sentence in the abstract: "*The three-dimensional genome structure organized by CTCF is required for development*", simplified the mutation information to: "*Clinically identified mutations in CTCF have been linked to adverse developmental outcomes. Nevertheless, the underlying mechanism remains elusive*", and revised the final sentence to: "*In summary, this study*

elucidates the influence of the CTCF^{R567W} mutation on human neurodevelopmental disorders, paving the way for potential therapeutic interventions".

Abstract: ‘it hindered CTCF binding to its upstream motifs’ – this will make no sense to a general readership. What does this really mean? How about ‘Peripheral motifs upstream to the core consensus site’

Response: Thank the Reviewer for the helpful suggestion. We have corrected this sentence in the revised manuscript as "*it specifically hindered CTCF binding to peripheral motifs upstream to the core consensus site*".

Line 97: ‘CTCF R567 a conserved residue….’ It would be stronger if you could state ‘a conserved residue in all known or most orthologues. ‘…is located proximal to the splice donor site in exon 9.

Line 100: The whole 11 ZFs are in the DNA interacting region of CTCF. ‘…specifically contacts DNA’ .

Response: We have corrected this sentence in the revised manuscript as "*The R567 residue of CTCF, a conserved residue across most known orthologues, is located proximal to the splice donor site in exon 9. Notably, this residue, situated within the 11th zinc finger (ZF) domain of the CTCF protein, was predicted to specifically contact DNA*".

Line 113: ‘Ctcf hemizygous mice’

Response: We have corrected this term by following the Reviewer’s suggestion.

Line 121: RT-qPCR shows a significant increase in CTCF? Therefore this statement is not correct.

Response: We are appreciated for the Reviewer to point out this error. We have corrected this sentence in the revised manuscript as "*we noted a statistically significant increase in Ctcf expression in neuron and lung tissues, as revealed by RT-qPCR, due to the CTCF^{R567W} mutation. However, this mutation did not significantly alter CTCF*

splicing or distribution across tissues, as assessed by Western blot and immunohistochemistry".

Line 146: ‘These findings suggest that half the amount of wild-type CTCF...’ This is very clunky and inaccurate. Total CTCF dosage has not changed as demonstrated by your careful WB analysis, despite the presence of mutant CTCF. I would suggest something along the lines of: ‘Unlike Ctf hemizygous mice which exhibit loss of CTCF expression in the brain (Ref PMID 32816606), a mostly normal brain function can be observed in Ctf^{+/R567W} mice where CTCF expression is preserved’ .

Response: We agreed that our initial phrasing might cause some confusions. We have revised the sentence as: *"These findings suggest that, unlike Ctf hemizygous mice which exhibit loss of Ctf expression in the brain (Ref PMID 32816606), Ctf^{+/R567W} mice maintain mostly normal brain function, given that Ctf expression is preserved"*.

Line 151: what known ‘clinical phenotypes’ ? Please specify.

Response: The term 'clinical phenotypes' in our study refers to the collection of developmental disorders that we have previously introduced in the introduction section. These traits include, but are not limited to, intellectual disability, growth deficiency, delayed development, short stature, delayed bone age, and feeding difficulties. Our study found that the mutation leads to birth lethality and severe developmental delay in mice, which, to some extent, mirrors these known clinical phenotype manifested as overall developmental disorders. To make this point clearer, we have revised the sentence as below: *"In summary, our results indicate that the homozygous CTCF^{R567W} mutation leads to birth lethality and developmental delay in mice, somewhat mirroring known clinical phenotypes manifested as overall developmental disorders"*.

Lines 189+: The bulk RNA-Seq of brain shows a lot of heterogeneity between WT and R567W mice (Supplementary Figure 4), whilst the human cortical organoid data is very consistent. I would suggest that the fact that E18.5 embryos were collected with no regard for sex/gender of the embryos accounts for this. Can the authors comment

whether sexes are mixed here and what samples were actually use for the RNAseq? This will definitely affect DEGs. See also line 415. Is this this limited due to heterogeneity introduced through gender?

Response: In our initial investigations, we found no significant sex-based differences in our tested phenotypes. Given the rarity of obtaining homozygous fetal mouse samples, we did not standardize the sex of the samples used for the bulk RNA-Seq, but we have had this potential confounding factor in consideration. To mitigate the effect of sex on gene expression in mouse samples, we filtered out genes originating from ChrX and ChrY during the following analysis.

Regarding the limited number of differentially expressed genes and the high level of heterogeneity, we believe that the impact of individual heterogeneity at the overall gene expression level is likely greater than the influence of sex. We also analyzed the expression of Y chromosome genes from RNA-seq which revealed that there was one female and three males in both the wild-type and homozygous mutant groups (**Figure R1**). However, significant heterogeneity was also observed among the three male mice (Supplementary Fig.4a). As for the human cortical organoid data, it was originated from a specific cell line, which inherently has less heterogeneity compared to mouse brain tissue.

Figure R1. RNA-seq signal tracks for the chrY region in wild-type and mutant mice.

Line 242: Section heading does not make sense.

Response: We have revised the section heading to "*CTCF^{R567W} mutation has little effect on cell type proportions but leads to noticeable DEGs in E18.5 heart and lung tissues*".

Lines 269-270: Rephrase. CTCF ZFs engage DNA not vice versa.

Response: Thanks for pointing this out. We have revised this sentence as follows:
"Specifically, ZFs 4-7 engage the M1/core (C) motif, which represents the core consensus, and ZFs 9-11 engage the M2/upstream (U) motif, corresponding to peripheral motifs upstream to the core consensus site".

Lines 274-5: R567W does not shift the tryptophan, it shifts the ZF. ‘Phosphoric acid’ is not correct here, ‘phosphate backbone’ or ‘phosphate ion’ is more accurate here.

Response: We are appreciated for the Reviewer #3’s insightful feedback. We have revised the sentence as *"Structural modeling indicates that the R567W mutation causes a shift of the ZF away from the DNA's phosphate backbone, thereby disrupting the critical hydrogen bond between R567 and the DNA backbone".*

Line 305: Is this the core consensus? If so, please state.

Response: Yes, it is the core consensus. We have revised this sentence as: *"For each group, we extracted 20 bp upstream sequences of CTCF core consensus binding motif, which were then divided into 10 subclusters".*

Line 330: I can’t see how the insulation score is noticeably altered? Please make colour schema in Fig 4g more obvious here. Also Supp Fig 8d, are in a different but more effective colour scheme.

Response: In the scatter plot for insulation score, points located along the diagonal indicate no change in the insulation score, while points further from the diagonal indicate a larger change. For example, points in the second and fourth quadrants represent cases where the insulation scores of the two genotypes have opposite signs, indicating a noticeable change. Given the large number of genome-wide insulation score values, we used a density scatter plot for comprehensive visualization. The colors in the plot only represent the density of points and are not related to changes in the

insulation score. We have explained the meaning of the color in the legend to avoid misunderstanding.

Line 384: I don't see these as repercussions? I would say functional consequences.

Response: We have revised the sentence as "*To evaluate the functional consequences of the CTCF^{R567W} mutation within the human cell system, we employed CRISPR/Cas9 technology to introduce the mutation into hESCs*".

Lines 546-7: This sounds too narrow. 'Successful engagement of peripheral CTCF ZFs such as ZF11 containing DNA-binding residue R567 is essential for

Response: We have revised the sentence as "*successful engagement of peripheral CTCF ZFs such as ZF11 containing DNA-binding residue R567 is essential for normal development across various tissue types*".

Box and whiskers plots for many data visualisations but it is not described what is being depicted ie 95% confidence intervals? Also for additional data in Supp Fig3 f to I - bar plots are used to depict similar data? Keep consistent.

Response: We thank the Reviewer's feedback on the visualizations of our data. We have added descriptions to all box and whiskers plots to indicate that they depict 95% confidence intervals. Furthermore, to maintain consistency, we have revised Supplementary Fig 3 (f to i) by using box and whiskers plots.

Fig.1a: The ordering of the species is puzzling. It would look better and make better sense if they were ordered from higher-order to lower-order species. Also please indicate those residues which make up ZF11 based on UniProt annotation. Bovine is misspelt.

Response: We have rearranged the species in descending order from higher to lower. Additionally, we have marked the residues of CTCF ZF11 based on UniProt annotation with a green line. We apologize for the misspelling and have corrected 'bovine' in the revised manuscript.

Fig.2a: what comparisons are the p-values generated from? It is unclear.

Response: We apologize for the lack of clarity in our original manuscript. The *p*-values in Fig.2a were generated from two-way ANOVA with repeated measures. We have now added this information to the figure legend.

Fig 1g, Fig2 and others: It would be more helpful if all graphs doing measurement of multiple samples eg mice are labelled on the x-axis with 'n'. This would save having to find this information in the legend.

Response: We have followed the Reviewer's suggestion and labeled the number of samples directly on the x-axis to enhance the clarity of the figures.

Fig 4d: The core consensus sequence logo must be shown here for both genotypes to show that it is invariant, while there are changes in U motif binding due to the presence of R567W

Response: We apologize for not displaying the core consensus sequence logo. Due to changes in our analysis method, we primarily focused on the characteristics of the U motif. In the initial version of our manuscript, we presented a sequence logo that included both the core consensus sequence and the upstream sequence. Now we have included this in our revised manuscript as Supplementary Fig. 7d. We have also added a description: "*Additionally, a greater reduction in CTCF binding corresponded to an increased enrichment of U CTCF motifs, specifically the ZFs 9-11 binding motifs, while the core consensus sequence remained invariant (Supplementary Fig. 7d)*".

Minor Comments:

Line 67: 'to emulate relevant phenotype' – this sounds awkward, please revise.

Response: We have revised the sentence as "*to mirror the pertinent phenotypic traits*".

Line 76: CTCF should be Cctf; italics for all usage of CTCF/Cctf gene and mRNA.

Response: We used "CTCF" and "CTCF^{R567W}" to represent the protein, while "*CTCF*" in italics represents the human gene and mRNA, and "*Cctf*" in italics represents the mouse gene and mRNA.

Line 103: What is 'expanded reproduction' ?

Response: We have revised the sentence to "Through multiple rounds of breeding and genotyping"

Line 339: EP abbreviation is not necessary.

Response: We have removed the abbreviation "EP".

Line 395: 'manifesting as poor organisation' .

Response: We have revised the phrase as "manifesting as poor organization."

Line 406: spelling of VGLUAT1.

Line 713: 'with'

Line 1503: i not I

Line 1527: indicated

Response: We have corrected all these errors.

Line 518: Influence x2

Response: We thank the Reviewer's#3 for point this out. We have deleted one "Influence".

Line 619: Addgene cat no was stated earlier.

Response: We have removed the repeated Addgene catalog here.

Line 481: 'its effects of CTCFR567W' ?

Response: We have corrected as "our focus centered on investigating the effects of CTCF^{R567W}"

Lines 495-6: Rephrase for clarity.

Response: We have rephrased it to "*Concurrently, despite our heterozygous mice did not fully phenocopy the clinical neurodevelopmental disorders, the observed cellular imbalance in heterozygous hESC-derived cortical organoids may offer essential insights into these specific disorders*".

Materials & Methods: all % should be qualified with w/v or v/v.

Response: We have revised our Materials & Methods section to specify whether the percentages refer to weight/volume (w/v) or volume/volume (v/v).

Line 621: What is the purpose of the puromycin?

Response: The purpose by adding puromycin is to select positive cells that have successfully incorporated the px459 (pSpCas9(BB)-2A-Puro) vector, which confers puromycin resistance. We have rephrased this sentence as "*Since Cas9-sgRNA vector contained puromycin resistant gene, the cells were screened with medium containing 1 µg/mL puromycin*".

BL-Hi-C and QHR-4C are not defined in Materials and Methods.

Response: "BL-Hi-C" refers to Bridge Linker-Hi-C and "QHR-4C" refers to Quantitative High-Resolution Chromosome Conformation Capture Copy. We have now added these definitions to the Materials and Methods section of our manuscript.

Line 1066: check 'frozen a chryotome' phrasing.

Response: We have revised "frozen a chryotome" to "a cryostat microtome".

Line 1093: Ref 14 is not in title case.

Response: We believe that the title case of Reference 14 was shown in the correct

format: 14. Gregor A, et al. De novo mutations in the genome organizer CTCF cause intellectual disability. American journal of human genetics 93, 124-131 (2013).

Line 1495: M & F would be more effective here in legend and graph, not symbols.

Response: We agree that using "M" for male and "F" for female would be clearer than using symbols. We have now revised our manuscript to replace the symbols with these abbreviations in both figure legend and the graph.

Line 1567: gb7CSE is not described as to what it is?

Response: "gb7CSE" refers to the DNA sequence used in the PDB data of the zf6-11-DNA binding complex, specifically the CTCF-binding conserved sequence element of the *Pcdhyb7* promoter. We have now added this definition to the figure legend: "*Crystal structure of the CTCF ZFs6-11-gb7CSE (CTCF-binding conserved sequence element within Pcdhyb7 promoter) complex obtained from PDB: 5YEL*".

Fig. 2b: y-axis 'area'

Fig. 3c: label 'involved'

Response: We thank the Reviewer #3 for his/her meticulous examination. We have corrected these spellings in these figures.

Fig 3e: Colours are too hard to distinguish here.

Response: We have revised the figure by using a more distinguishable color scheme.

Fig 3f: Colours for 'down' & 'up' are switched/different to Supp 4B which is confusing.

Response: We have now revised the figures to use a consistent color scheme.

Fig 4a: PDB ID is not needed; label DNA strands with 5' and 3' ends.

Response: We have removed the PDB ID and added labels to indicate the 5' and 3' ends of the DNA strands.

Fig 4d: The core consensus sequence logo must be shown here for both genotypes to show that it is invariant, while there are changes in U motif binding due to the presence of R567W

Response: we have now included the invariant core consensus sequence logo in our revised manuscript as Supplementary Fig. 7d. We have added one panel and following sentence: *"Additionally, a greater reduction in CTCF binding corresponded to an increased enrichment of U CTCF motifs, specifically the ZFs 9-11 binding motifs, while the core consensus sequence remained invariant (Supplementary Fig. 7d)".*

Fig 6a: labelling size should increase.

Response: We have increased the labelling size in this figure.

Supplementary Information

1a: 'CTCF heterozygous mice'

Response: We thought that the term "Genotypes of *Ctcf* mutant mice" has no problem, as the data includes both heterozygous and homozygous mutants.

2k: y-axis 'Interaction'

5h: 'transporter'

9a: 'editing' x2

5a: *Pecam1* has become separated

Response: We have corrected these mistakes by following the Reviewer's suggestions.

9c: a short vertical line should be added to mark exon/intron boundaries.

Response: We have now added black dashed lines to represent exon/intron boundaries in Supplementary Fig. 9c.

Reviewers' Comments:

Reviewer #2:

Remarks to the Author:

Thanks for elaboration. I don't have further questions.

Reviewer #3:

Remarks to the Author:

The authors have addressed all concerns, suggestions and errors pointed out by the Reviewers. The abstract is vastly improved, the primary data is well laid out and the manuscript is easy to follow. I commend the authors on their willingness to adopt critical feedback. Congratulations on a mature and interesting manuscript.